# Mechanical disengagement of the cohesin ring

**Martina Richeldi** [1,2,3], **Georgii Pobegalov** [1,3], **Torahiko L. Higashi**[2,4], **Karolina Gmurczyk**[2], **Frank Uhlmann** [2] ✉ **& Maxim I. Molodtsov** [1,3] ✉

Cohesin forms a proteinaceous ring that is thought to link sister chromatids by entrapping DNA and counteracting the forces generated by the mitotic spindle. Whether individual cohesins encircle both sister DNAs and how cohesin opposes spindle-generated forces remains unknown. Here we perform force measurements on individual yeast cohesin complexes either bound to DNA or holding together two DNAs. By covalently closing the hinge and Smc3[Psm3]–kleisin interfaces we find that the mechanical stability of the cohesin ring entrapping DNA is determined by the hinge domain. Forces of ~20 pN disengage cohesin at the hinge and release DNA, indicating that ~40 cohesin molecules are sufficient to counteract known spindle forces. Our findings provide a mechanical framework for understanding how cohesin interacts with sister chromatids and opposes the spindle-generated tension during mitosis, with implications for other force-generating chromosomal processes including transcription and DNA replication.

The accuracy of genome inheritance depends on the faithful segregation of sister chromatids. Segregation without errors requires correct biorientation of chromosomes, achieved when microtubules from opposite spindle poles attach to kinetochores on sister chromatids until all of them come under tension[1,2]. Sister chromatids are physically linked by the chromosomal complex cohesin, which counteracts the pole-directed, spindle-generated forces required for biorientation and chromosome segregation[3,4]. Once biorientation is established, the cohesin complex is cleaved by separase and individual chromatids symmetrically separate towards opposite poles[5].

The cohesin complex is composed of four core subunits, arranged to form a distinct ring-like architecture critical to its capacity to embrace DNA and establish sister chromatid cohesion[6–8]. The flexible Smc1[Psm1] and Smc3[Psm3] subunits (budding yeast nomenclature with fission yeast proteins used in this study in superscript) are connected at one end via the hinge domain, while at the other end lie the ATP-binding heads[5] (Fig. 1a). The kleisin subunit Scc1[Rad21] completes the cohesin ring by connecting the ATPase heads. Scc1[Rad21] also mediates interactions with Scc3[Psc3], key to recruiting and maintaining

cohesin's association with chromosomes, as well as with the cohesin loader Scc2[Mis4]–Scc4[Ssl3] and regulatory proteins Pds5[Pds5] and Wpl1[Wapl] (refs. 9–11).

The cohesin ring physically entraps DNA[6,12]. This activity is consistent with its function in mitosis, where cohesin must possess remarkable mechanical stability to hold sister chromatids against tensions of up to several hundreds of piconewtons[13] to allow correct chromosome biorientation. However, whether a single cohesin complex can hold both sister DNAs is unknown, as is the force that one cohesin might be able to withstand. During interphase, cohesin is thought to translocate along DNA and extrude DNA loops[14–17]. This might entail interactions between cohesin and mechanical barriers such as RNA polymerases and the replication machinery, which can generate tens of piconewtons of force[18,19]. How cohesin behaves upon encountering force-generating molecular complexes is also unknown. In this article, we measured mechanical forces that disengage a single cohesin ring complex and showed that the disengagement leads to the dissolution of both cohesin–DNA and cohesin-mediated DNA–DNA interactions.

[1]Biophysics and Mechanobiology Laboratory, The Francis Crick Institute, London, UK. [2]Chromosome Segregation Laboratory, The Francis Crick Institute, London, UK. [3]Department of Physics and Astronomy, University College London, London, UK. [4]Present address: Kamakura Research Laboratories, Chugai Pharmaceutical Co., Kamakura City, Japan. ✉e-mail: frank.uhlmann@crick.ac.uk; m.molodtsov@ucl.ac.uk

## Results

### Individual cohesins load topologically on DNA

To address how cohesin mechanically interacts with DNA, we devised an in vitro system in which we could monitor the response of individual DNA-bound cohesin complexes to external force, while visualizing the same cohesin molecules and DNA by total internal reflection fluorescence (TIRF) microscopy (Fig. 1b). We purified active fission yeast cohesin tetramers labeled with both a tetramethylrhodamine (TMR) fluorophore for visualization, and a biotin tag for binding to streptavidin-coated beads and force application, fused to the Smc1[Psm1] and Smc3[Psm3] head domains, respectively (Extended Data Fig. 1a–d)[20].

In the presence of ATP and the cohesin loader, we loaded individual cohesin complexes onto the tethered λ-DNA such that ~80% of all loaded cohesin molecules were single cohesin complexes, as indicated from fluorescence intensity and single-step photobleaching (Fig. 1c,d and Extended Data Fig. 1e). To ensure that only those cohesins that topologically entrapped DNA remained bound, we performed washes with increasing NaCl concentrations. After washes containing 130 mM NaCl, ~30% of all initially loaded cohesin persisted on DNA. This fraction remained unchanged after further increasing the salt concentration up to 2 M (Fig. 1e). When, in a separate experiment, we added cohesin without the loader Scc2[Mis4]–Scc4[Ssl3] and ATP, required for topological loading, almost no cohesin remained on DNA following the salt washes (Fig. 1e), supporting the idea that salt-resistant cohesin topologically interacts with DNA. To test the topological nature of cohesin loading further, we cleaved the λ-DNA at a single site using the restriction enzyme XhoI. Upon cleavage, all the examined cohesin molecules slid off and left DNA ($n = 25$), which confirmed the topological interaction between DNA and salt-resistant cohesin (Supplementary Video 1).

Finally, we asked whether, following loading, DNA was entrapped inside cohesin's main ring[6]. To test this, we employed the SpyTag–SpyCatcher covalent crosslinking system[21]. We purified cohesin bearing two SpyTags, one attached to the Smc3[Psm3] C-terminus and one to the kleisin N-terminus ('Smc3[Psm3]–kleisin' cohesin). This allowed for covalent closure of the Smc3[Psm3]–kleisin interface using a crosslinker consisting of two SpyCatcher modules, connected by a long and flexible unstructured polypeptide linker (Methods and Extended Data Fig. 1f). Before crosslinker addition, Smc3[Psm3]–kleisin cohesin loaded onto DNA in a salt-resistant manner, similarly to the wild-type complex (Fig. 1f). After loading onto DNA, addition of the crosslinker efficiently prevented spontaneous cohesin release from DNA in the presence of ATP, consistent with covalent closure of the Smc3[Psm3]–kleisin interface through which DNA is thought to unload[22,23]. Agreeing with the efficiency of spontaneous release determined earlier[22], ~52% of non-crosslinked cohesin complexes were released from the DNA in the presence of ATP after 60 minutes of incubation. In comparison, only 3% of the Smc3[Psm3]–kleisin crosslinked cohesin were removed in similar conditions. This indicates crosslinking efficiency of over 90% (Fig. 1g) and shows that closure of the Smc3[Psm3]–kleisin interface of cohesin on DNA prevents it from unloading. Hence, we confirmed that DNA was topologically entrapped inside the cohesin ring after it was loaded onto DNA.

### Cohesin disengages from DNA under force

Having identified conditions for the topological loading of individual cohesin complexes onto DNA, we investigated the mechanical stability of the cohesin ring entrapping the DNA. To this end, we attached streptavidin-coated beads to cohesins on λ-DNA and used optical tweezers to apply force by displacing the bead.

To ensure that force was applied to a single cohesin only, we optimized the number of cohesin molecules per DNA. The experimentally determined probability distribution of the number of cohesins per DNA and the distribution of the number of beads per DNA allowed us to calculate the distribution of cohesin complexes per individual bead on DNA (Methods and Extended Data Fig. 2a). In optimized conditions,

this resulted in approximately 70% of all beads attached to DNA via a single cohesin molecule and the rest by two or more cohesin molecules.

After the bead attachment, we probed mechanical stability of cohesin by displacing the bead with respect to DNA. When force was applied along the DNA, cohesin slid on DNA almost freely, with little to no resistance (Fig. 2a and Supplementary Video 2). In contrast, when force was applied perpendicularly to DNA, after the initial DNA stretching, cohesin strongly resisted movement and detachment. Free movement along the DNA, but resistance to perpendicular removal from DNA, is consistent with the topological interaction between cohesin and DNA.

We next collected force–distance (FD) curves by moving the bead at a constant velocity perpendicularly away from the DNA, which first stretched the DNA and then led to an abrupt detachment event (Fig. 2b and Supplementary Video 3). Imaging cohesin attached to the bead before force application revealed that cohesin bleached in a single-step manner, consistent with the single-step detachment signature observed in the FD curves and demonstrating that the measured detachment force was that of a single cohesin molecule bound to DNA (Fig. 2b and Extended Data Fig. 2b). Of the total detachment events observed, 70% (total $n = 92$) of the FD curves showed a single rupture event at an average force of ~20 pN (Fig. 2c), and the remaining FD curves showed multiple detachment peaks (Extended Data Fig. 2c). Thus, the observed ratio of the single-step detachments was in excellent agreement with the expected fraction of the beads bound to single cohesin molecules, which further supported our conclusion that the single-step detachment events corresponded to single cohesin ruptures.

To verify that the observed detachments were indeed due to the rupture of the cohesin ring and release of its DNA interaction, we visualized single cohesins on DNA before and after the force-induced rupture. The disappearance of the cohesin signal from DNA after the mechanical rupture event confirmed cohesin detachment (Fig. 2d). As a control, we applied 5 pN of force to cohesin for the same duration of the experiment, which was not sufficient for detachment. After releasing the bead without observing ruptures in the FD curves, cohesin remained visible on the DNA and attached to the bead in all cases ($n = 12$).

To further confirm that our assay reported on cohesin rupture, and not detachment of the biotin–avidin interaction or DNA unwinding, we replaced cohesin with a biotin that was directly covalently coupled to DNA and to which we attached an avidin-coated bead. We could not detach the bead from DNA, even with forces exceeding 80 pN, ultimately leading to the bead escaping from the trap but remaining bound to DNA (Extended Data Fig. 2d). We also recorded FD curves of the DNA by attaching its one end to the flow cell and the other end to the bead. This resulted in the typical DNA overstretching transitions at ~65 pN (Extended Data Fig. 2e), as would be expected for the double-stranded DNA (dsDNA)[24], and very different from our observed discrete cohesin–DNA rupture events. These controls further bolstered our conclusion that the bead detachment from DNA in our experiments was indeed due to the rupture of the cohesin ring.

### The cohesin ring ruptures at a specific interface

To determine whether the direction of force applied to cohesin influenced the rupture, we tested a recombinant cohesin complex bearing the biotin tag inserted at the Smc3[Psm3] hinge domain, instead of the head (Extended Data Fig. 3). Combined data from single-step FD curves revealed a small but statistically significant decrease in the rupture force for complexes pulled via the hinge domain (median 18 pN) compared to those pulled via the head domain (median 24 pN, $P = 0.0083$−Kolmogorov–Smirnov test) (Fig. 2e).

To explore possible reasons for this difference, we took a modeling approach. Using Monte-Carlo simulations, we tested whether our experimental observations could be explained by a model in which the rupture occurs due to the disengagement of a cohesin ring entrapping DNA. We described the cohesin ring using two parameters: $k_0$, the rate

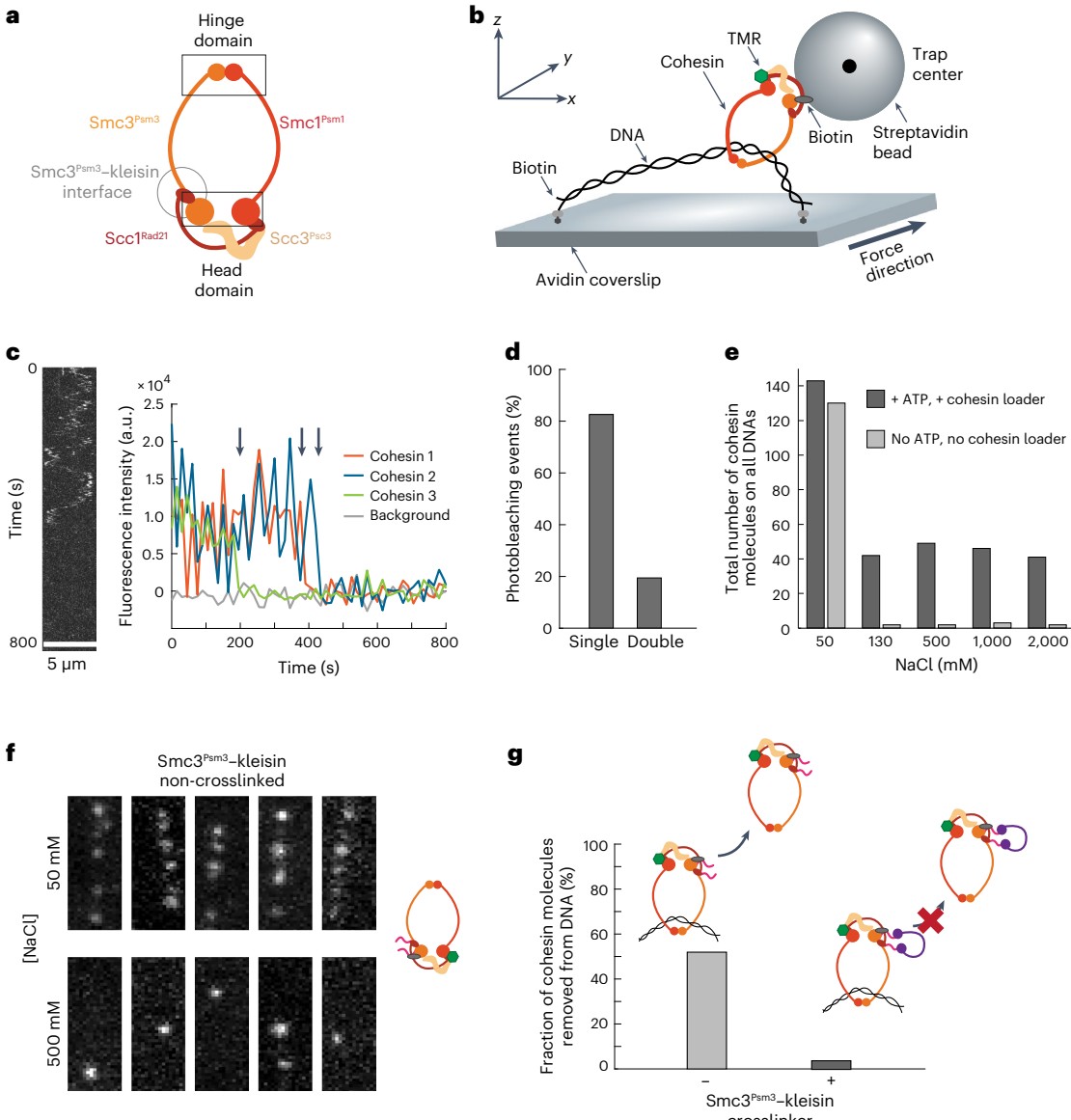

**Fig. 1 | Topological loading of individual cohesins on DNA. a**, Schematic illustration of the cohesin tetramer. **b**, Graphical representation of the assay for visualizing and applying force to the head domain of cohesin bound to λ-DNA. Loading of the cohesin tetramer onto DNA is performed in the presence of the cohesin loader Scc2$^{Mis4}$–Scc4$^{Ssl3}$ and ATP (not depicted). **c**, Example of a kymograph (left) and three examples of single-step photobleaching traces (right) of TMR-labeled single cohesin molecules on λ-DNA. Arrows point to bleaching events of three independent cohesins. **d**, Distribution showing the fraction of single salt-resistant monomeric cohesins on DNA (*n* = 112). **e**, Total number of cohesin molecules bound to all λ-DNAs in the presence or absence

of the cohesin loader and ATP after washes at increasing NaCl concentrations. Absolute numbers shown correspond to all cohesins on all DNAs counted across three independent experiments (*n* = 3). **f**, Stills showing loading of the Smc3$^{Psm3}$–kleisin cohesin without the crosslinker. Five DNA examples are shown at 50 mM and after 500 mM NaCl wash. DNA itself is not visualized. **g**, Quantification of Smc3$^{Psm3}$–kleisin cohesin crosslinking. Without addition of the crosslinker, 52% of all salt-resistant cohesins were spontaneously released from the tethered DNA. Under the same conditions, following crosslinking, only 3% of Smc3$^{Psm3}$–kleisin cohesin were released (*n* = 67, total molecules).

of spontaneous cohesin ring disengagement (interface opening) in the absence of force, and δ, a mechanical displacement parameter which determines how force affects the disengagement (Fig. 3a, Extended Data Fig. 4a–f and Methods). When we independently fitted the experimental distributions for cohesin variants pulled either via the head or the hinge domain to this model, we obtained the following values for $k_0$ and δ:

$$k_0^{(head)} = 0.0027 \text{ s}^{-1}, (0.0023, 0.0036), \delta^{(head)} = 1.23 \text{ nm}, (1.17, 1.29)$$

$$k_0^{(hinge)} = 0.0025 \text{ s}^{-1}, (0.0016, 0.0034), \delta^{(hinge)} = 1.61 \text{ nm}, (1.47, 1.71)$$

where the values in brackets show 90% confidence intervals (Fig. 2c,e, solid lines).

This shows that the datasets for cohesin variants where force was applied either via the head or the hinge domain were best described with the same $k_0$, but different δ. Bootstrap analysis confirmed that the sample size was sufficient, and that a further increase in the number of measurements would not affect this result (Extended Data Fig. 4g–i). This finding suggests that, when an external force is applied at two different locations, the cohesin ring opens at the same interface (characterized by the same spontaneous opening rate $k_0$), while the displacement parameter δ varies because the different force direction

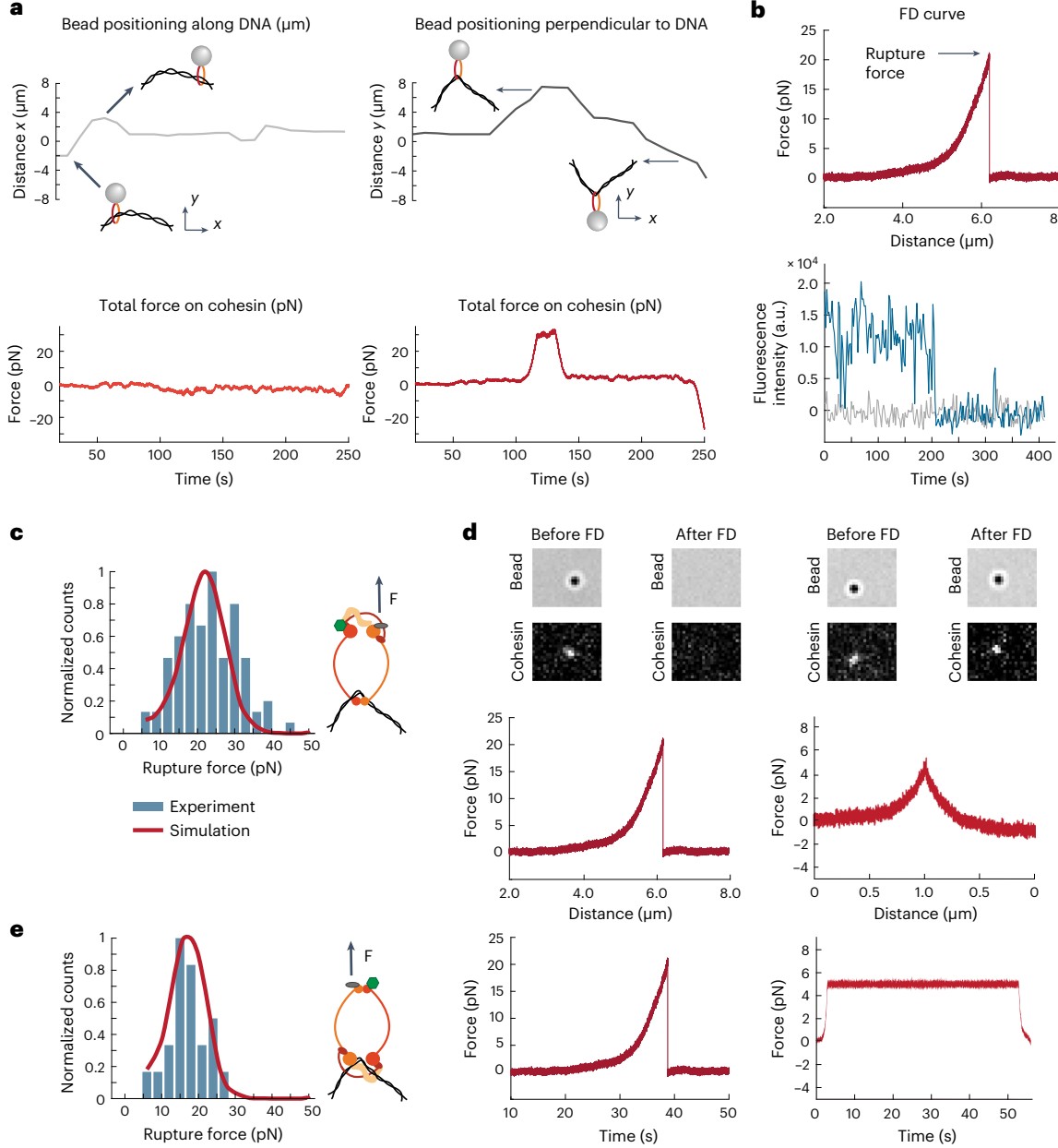

**Fig. 2 | Single cohesin on DNA resists ~20 pN forces. a**, Forces exerted on cohesin as it moves along the tethered λ-DNA (*x* direction, left) or perpendicularly to it (*y* direction, right). Distance shows the relative position of the bead along the corresponding axis. Relates to Supplementary Video 2. **b**, Typical example of an FD curve (top) and the corresponding single-step photobleaching trace (bottom) showing the rupture of a single cohesin from the tethered λ-DNA at a force of 20 pN. Relates to Supplementary Video 3. **c**, Normalized distribution of the total rupture forces for the cohesin variant where force was applied via the head domain (*n* = 89). In **c** and **e**, solid lines show theoretical distributions of rupture forces with optimal parameters determined by fitting experimental data. Parameter values are shown in the main text. **d**, Cohesin's fluorescent signal colocalizing with the bead before and after two pulling experiments, in which the rupture event was recorded (left), or when the applied force was insufficient (5 pN) for detachment. DNA is not visualized. Bottom graphs show corresponding FD curves for these experiments as a function of both distance and time. **e**, Normalized distribution of the total rupture forces for the cohesin variant where force was applied via the hinge domain (*n* = 21). The solid line is the theoretical distribution.

leads to the bond rupturing along a different trajectory[25]. Indeed, a ring would always be expected to break at the same, weakest interface, irrespective of where the force is applied.

## The cohesin ring opens at the hinge interface

To investigate where the cohesin ring breaks under external force, we began by examining the Smc3[Psm3]–kleisin interface, through which DNA is thought to pass during cohesin's enzymatic, ATP-dependent unloading from chromosomes. As described above, we used the

SpyCatcher-based crosslinker to covalently close this interface with over 90% efficiency after loading of Smc3[Psm3]–kleisin cohesin onto λ-DNA in the flow cell (Fig. 1g). We then applied force from the head domain of the Smc3[Psm3]–kleisin-crosslinked complexes. The histogram of the rupture forces revealed an average value of rupture force of ~20 pN, with a distribution similar to that of the wild-type complex (Fig. 3b). Moreover, when we fitted model parameters to this distribution, we obtained values that were statistically indistinguishable from those describing wild-type cohesin: $k_0^{(\text{head}/X)} = 0.0022\,\text{s}^{-1}$, (0.0011,

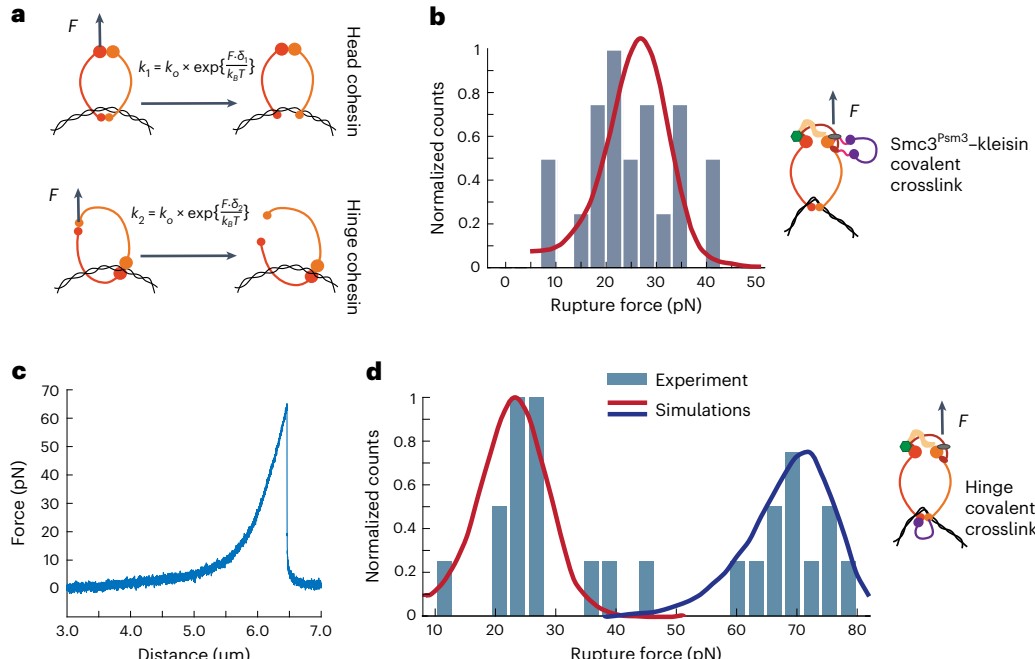

**Fig. 3 | Mechanical force disengages the cohesin ring at the hinge interface.**
**a**, Schematic representation of the cohesin ring disengagement under external
force for cohesin variants with the biotin tag for force application attached to
either the heads or the hinge domain. The formulas show how disengagement
rates for different geometries depend on force, $k_0$ and $\delta$. **b**, Normalized
distribution of the total rupture forces for Smc3[Psm3]–kleisin crosslinked cohesin

($n = 24$). **c**, Typical example of the FD curve showing a single rupture event at a
high force (~70 pN) for the hinge-crosslinked cohesin. **d**, Normalized distribution
of the rupture forces for the hinge-crosslinked cohesin ($n = 25$). In **b** and **d**, solid
lines show theoretical distributions of rupture forces with optimal parameters
determined by fitting experimental data. Parameter values are shown in the main
text.

0.0035), $\delta^{(head/X)} = 1.22$ nm, (1.1, 1.35) (Fig. 2c). Since covalent bonds
can resist forces over 1,000 pN, the rupture events could not have
occurred at the crosslinked Smc3[Psm3]–kleisin interface in this case.
Therefore, cohesin ring disengagement at ~20 pN must have occurred
at a different interface.

Next, we examined cohesin's hinge domain. The hinge presents a
relatively small contact area between the Smc1[Psm1] and Smc3[Psm3] sub-
units, and earlier experiments have observed that it may transiently
open[8,26,27]. To test whether the hinge could represent the weakest
interface in the cohesin ring, we again used the SpyTag–SpyCatcher
system. This time, we directly covalently crosslinked cohesin's hinge
domain by inserting a SpyTag into Smc3[Psm3] and SpyCatcher into
Smc1[Psm1] (Extended Data Fig. 5a). The resulting hinge crosslinking
efficiency was ~70% (Extended Data Fig. 5b,c). We found that the
hinge-crosslinked complex could topologically load onto DNA, albeit
with slightly reduced efficiency compared to wild-type cohesin
(Extended Data Fig. 5d), consistent with recent observations of the
budding yeast cohesin complex[28]. Considering both crosslinking
efficiency and the reduced ability of crosslinked molecules to load
onto DNA, approximately half of the DNA-loaded cohesin molecules
are expected to be successfully crosslinked under these conditions
(Methods).

Force application to the DNA-loaded cohesin complexes resulted
in a bimodal distribution of rupture forces. Approximately half of the
cohesin molecules detached at ~20 pN and half at ~70 pN (Fig. 3c,d),
revealing a marked impact of hinge crosslinking on cohesin's force
response. Fitting the observed rupture force distribution to our model
showed that the peak at 20 pN could be explained by the disengage-
ment of the same interface as for wild-type cohesin and described with
the statistically identical set of parameters: $k_0^{(20)} = 0.0021$ s$^{-1}$, (0.001,
0.0034), $\delta^{(20)} = 1.23$ nm, (1.1, 1.37). Thus, the peak at 20 pN probably
corresponds to the population of cohesin molecules that failed to

crosslink at the hinge and therefore behave as wild type. In turn, the
peak at 70 pN points to the rupture of a different interface, character-
ized by a distinct set of parameters indicating much stronger
link: $k_0^{(70)} = 2.7 \times 10^{-5}$ s$^{-1}$, ($1.6 \times 10^{-5}$, $3.9 \times 10^{-5}$), $\delta^{(70)} = 0.8$ nm,
(0.77, 0.83). Since this peak was absent from all previous distributions,
it must have arisen from cohesin molecules with successfully
crosslinked hinge domain. As covalent crosslinking requires forces
above ~1,000 pN to break, the 70 pN peak could not be the result of a
breakage of the crosslink, but rather represents the disengagement at
another, the second weakest, cohesin ring interface.

These experiments showed that in all cohesin constructs in which
the hinge can disengage, the ruptures occur at ~20 pN, and much higher
forces are required to break the cohesin ring when the hinge is cova-
lently closed. Thus, we established that the hinge is the weakest inter-
face in cohesin ring and that it breaks at ~20 pN.

**A single cohesin complex establishes DNA–DNA interactions**
Next, we addressed whether a single cohesin complex could entrap
two DNA molecules and, if so, what force such an interaction with-
stands. After adding fluorescently labeled circular DNA plasmids, we
found that both single-stranded DNA (ssDNA) (41 events) and dsDNA
plasmids (36 events) could be captured by single cohesins preloaded
onto λ-DNA. While second dsDNA capture is inefficient in bulk assays
compared to ssDNA[29], the sensitivity of our single-molecule experi-
ments allowed us to observe rarer dsDNA capture events. Capture was
evident from colocalization of cohesin and the second DNA (Fig. 4a,b),
which persisted even after 500 mM NaCl washes (Extended Data
Fig. 6a). Confirming the topological nature of second DNA capture,
we found that the plasmid was lost in 16 out of 21 cases after addition
of the restriction enzyme PacI, which recognizes the plasmid but not
the λ-DNA (Fig. 4c). The few plasmid molecules that persisted follow-
ing PacI addition were probably the result of either incomplete PacI

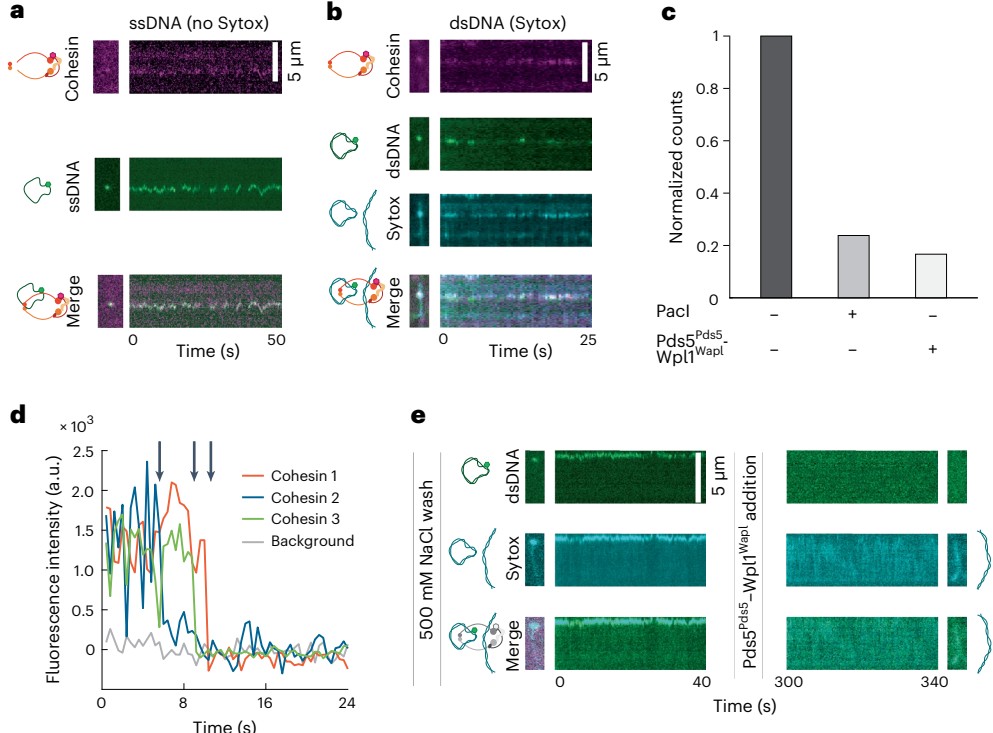

**Fig. 4 | A single cohesin complex captures two DNAs. a**, Example of kymographs showing LD655-labeled cohesin colocalizing with a MFP488-labeled single-stranded plasmid (*n* = 36 in at least five independent repeats). **b**, Example of kymographs illustrating LD655-labeled cohesin colocalizing with a MFP488-labeled double-stranded plasmid, also stained with Sytox Orange (Sytox) (*n* = 41 in at least five independent repeats). **c**, Fraction of second DNA molecules that remain attached to λ-DNA after addition of restriction enzyme PacI (*n* = 21) or unloading complex Pds5$^{Pds5}$–Wpl1$^{Wapl}$ (*n* = 6). Connection with the second DNA is stable when no additional proteins are added (*n* = 24). **d**, Three examples of single-step photobleaching traces of single LD655-labeled cohesin molecules that performed second DNA capture. Arrows point to bleaching events of three independent cohesins. **e**, Example of kymographs of second dsDNA capture by cohesin before and after Pds5$^{Pds5}$–Wpl1$^{Wapl}$ addition (*n* = 6 in three independent repeats).

cleavage in our reaction buffer, or of inefficient diffusion of the cut plasmid out of the sterically restricting cohesin ring.

Single-step photobleaching traces (Fig. 4d and Extended Data Fig. 6b) confirmed that single cohesin complexes were responsible for mediating the majority of both dsDNA–ssDNA (12 out of 13 cases analyzed) and dsDNA–dsDNA interactions (7 out of 10 cases analyzed). We observed that ssDNA capture was labile (Extended Data Fig. 6c and Supplementary Video 4) and that we could convert the second ssDNA to dsDNA using T7 DNA polymerase (Extended Data Fig. 6d), in agreement with previous observations[29].

Next, we interrogated the dependency of the reaction on ATP and the cohesin loader Scc2$^{Mis4}$–Scc4$^{Ssl3}$ (ref. 29). Virtually no second DNA capture was recorded in the absence of either ATP (1 event out of 151 cohesin-decorated λ-DNAs) or Scc2$^{Mis4}$–Scc4$^{Ssl3}$ (0 events out of 236 λ-DNAs), compared to 77 second DNA capture events on 203 λ-DNAs in the presence of both. Moreover, introduction of the cohesin unloading complex Pds5$^{Pds5}$–Wpl1$^{Wapl}$ following second DNA capture (Extended Data Fig. 6e) led to second dsDNA dissociation (Fig. 4c,e), confirming the potency of the cohesin unloader in dissolving DNA–DNA interactions[11]. Thus, the requirements for second DNA capture, its stable topological nature and sensitivity to a biological unloader suggest that single cohesin molecules have reconstituted physiologically relevant DNA–DNA interactions in our experimental setup.

**Cohesin ring disengagement dissolves DNA–DNA interaction**

Having established that individual cohesins can link two DNA molecules, we measured the force that a single cohesin complex can resist when holding together two DNAs (Fig. 5a). To do so, we used covalently biotinylated second DNA substrates to which we attached streptavidin-coated beads and recorded FD curves until we detected DNA–DNA ruptures (Fig. 5b and Supplementary Videos 5 and 6). Most ruptures appeared as a single peak, both for dsDNA substrates (24 out of 28 measurements) and for ssDNAs (17 out of 20 measurements) (Fig. 5c and Extended Data Fig. 7a). For those single-peak rupture events, we confirmed that the two DNAs were tethered by a single cohesin ring, as indicated by single-step photobleaching of cohesin's fluorescent label, either before (Fig. 4d) or after binding to the beads (Fig. 5d and Extended Data Fig. 7b).

Average rupture forces recorded with both second ssDNA and dsDNA were ~16 pN (Fig. 5e). There was no statistically significant difference between forces recorded with a second ssDNA or dsDNA (Extended Data Fig. 7c). However, the rupture forces recorded between two cohesin-linked DNAs were overall smaller than those measured when beads were attached directly to cohesin ($P$ = 3.8 × 10$^{-5}$ – Kolmogorov–Smirnov test). To understand this difference, we investigated the consequence of including the second DNA in our computational ring disengagement model, assuming the parameters of the cohesin ring rupture were exactly the same as those previously determined (Fig. 2c,e) and that disengagement happened randomly in either the hinge-like or head-like orientation. These simulations indeed resulted in the reduced predicted rupture forces consistent with our measurements (Fig. 5e). This is because stretching more DNA, which now includes both the first and the second DNA molecules, takes up additional time and reduces the effective force ramping rate, which increases the likelihood that cohesin ruptures at a slightly lower force (Extended Data Fig. 7d–f). Thus, the measured rupture force distribution of DNA–DNA interactions is explained by the rupture of the cohesin ring at the same interface and therefore is consistent with being determined by the same mechanism as in experiments where the force was applied directly to cohesin. Taken

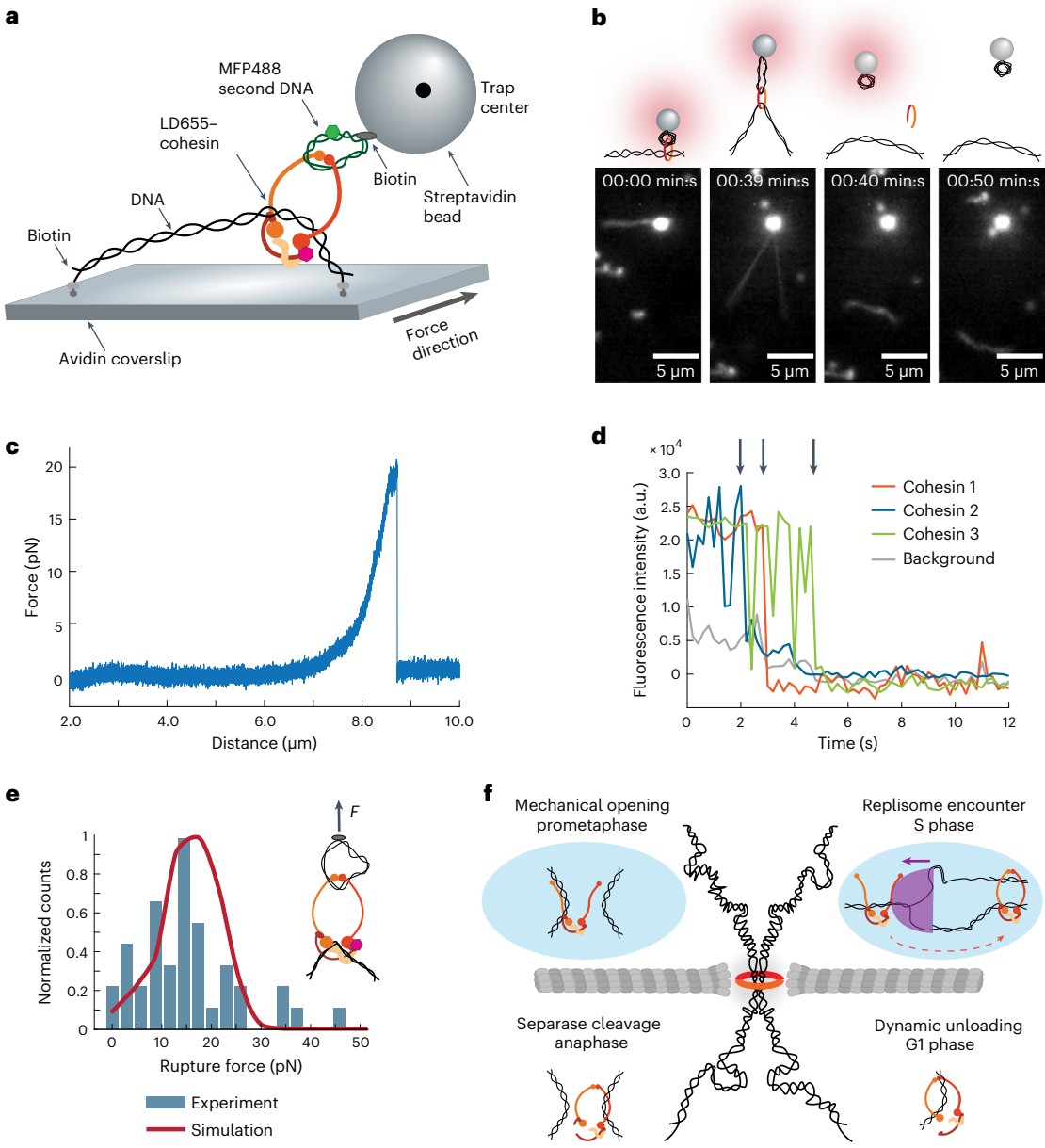

**Fig. 5 | The DNA–DNA interaction ruptures due to mechanical cohesin ring disengagement. a**, Graphical representation of the assay developed for visualizing and applying force to cohesin holding together two DNAs. **b**, Schematic illustration (top) and snapshots (bottom) showing detachment of the second dsDNA from the λ-DNA, which remains attached to the surface ($n = 24$ in at least three independent repeats). Relates to Supplementary Video 6. **c**, Characteristic FD curve showing detachment of the second dsDNA at ~20 pN. **d**, Three examples of single-step photobleaching traces of single bead-bound LD655–cohesins linking two DNAs. Arrows point to bleaching events for three independent molecules. **e**, Normalized distribution of rupture forces for cohesin between two DNAs (both ssDNA and dsDNA combined, $n = 41$). The solid line shows theoretical distribution of rupture forces. Parameters are the same as in Fig. 2c,e. For details, see text. **f**, Schematics illustrating different events during cell cycle when cohesin can be removed from DNA, either biochemically (bottom) or mechanically, via disengagement of the cohesin ring (top).

together, our experiments demonstrate that single cohesin ring holds together two DNA molecules and withstands forces up to 20 pN. All our results can be accounted for by a model in which the cohesin–DNA interaction breaks under external force due to cohesin disengagement at its weakest interface, the hinge domain.

## Discussion

In this work we have shown that the mechanical stability of the cohesin–DNA interaction for cohesin complexes topologically entrapping DNA

inside its main ring compartment is determined by the physical stability of the cohesin hinge domain. Our simulations confirmed that the simplest model that can account for all our data consists in the cohesin ring characterized by two interfaces that can disengage under mechanical force, leading to the release of DNA. We showed that the weakest interface that disengages first is the hinge with a stability limited by ~20 pN and the other is presumably the Smc3[Psm3]–kleisin, which is substantially stronger, but also eventually disengages at forces of ~70 pN (Extended Data Fig. 8). We have also demonstrated that single

cohesin ring can hold two DNA molecules and that the stability of the DNA–DNA interaction under external force maintained by the single cohesin complex is limited by the physical stability of the cohesin ring.

These results provide a framework for understanding the mechanical forces acting on sister chromatids during cell division. Mechanical tension is the main regulator of mitotic progression from metaphase to anaphase. Various estimates put the maximum forces applied to chromosomes from the spindle of up to ~700 pN (ref. [13]) and the absolute number of cohesins at ~200 molecules per centromere[30], with cohesins located close to the microtubule attachment sites probably bearing the majority of the physical load exerted by the spindle.

Cohesin depletion studies have reported that ~20% of chromosome-bound cohesin (that is, ~40 molecules) is sufficient to sustain sister chromatid cohesion[31,32], and that the apparent excess of cohesin molecules at the centromere might serve to avoid cohesion fatigue[32]. If individual cohesins entrap both sister chromatids in vivo, and assuming that a maximum load of ~700 pN is exerted by the spindle on the chromosome, each cohesin would have to resist ~18 pN, a value close to the force measured in our experiments. These estimates suggest that the minimal necessary number of cohesin molecules at centromeres is probably limited by the mechanical stability of the cohesin ring.

Our findings can explain the mechanism behind the transient splitting of chromosomes in pre-anaphase known as 'centromere breathing'[33]. The build-up of tension generated by spindle forces could force the cohesin molecules that experience the most strain to undergo mechanical disengagement at the hinge. However, the connection with the DNA could be re-established by either dynamic cohesin loading or simple hinge closure.

Since the hinge domain is connected to the head domains only via long flexible coiled coils, it is unlikely that the purely mechanical disengagement of cohesin at the hinge would be affected by the chemical state of its heads. One important modification of cohesin is the acetylation of the Smc3$^{Psm3}$ head domain[34] that stabilizes the cohesin–chromosome association in the absence of external force. While this modification reduces the enzymatic turnover of cohesin on DNA, given the large distance and flexibility between the hinge and head domains, we speculate that the physical stability of cohesin under external force determined by the hinge would presumably be unaffected by acetylation.

Mechanical opening of the cohesin ring may also have implications in other processes that involve interaction between chromosomes and cohesin, including transcription and DNA replication. RNA polymerases are known to act as barriers for cohesin movement[17]. Although the stall force for RNA Polymerase II is ~8 pN, associated proteins increase this by a factor of two[18], which might enable the transcription machinery to physically open cohesin molecules and assist in polymerase translocation past immobile cohesin, for example, when bound to CCCTC-binding factor[17]. Replicative helicases in turn generate forces in excess of 20 pN (ref. [19]). Their encounter with cohesin could result in transient hinge disengagement, allowing the replication machinery to mechanically open up cohesin rings during the establishment of sister chromatid cohesion. This may lead to cohesin removal altogether or may allow the replication or transcription machinery to pass beyond the disengaged cohesin ring, which would remain on DNA in its open form and possibly close back forming the ring again after allowing the bulky complexes to move past it (Fig. 5f).

In conclusion, we show that physical force is a physiologically possible mechanism for disengagement of the cohesin ring, in addition to cleavage by separase and the enzymatic removal action of Pds5$^{Pds5}$–Wpl1$^{Wapl1}$ (Fig. 5f). Mechanical disengagement of the cohesin ring is likely to play roles during both interphase and mitosis. In the future, it will be interesting to explore the effect of conditionally closing cohesin interfaces in vivo to probe the consequences of altering cohesin's mechanical stability in the context of transcription, replication and chromosome biorientation.

## Online content

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

## Methods

Detailed methods describing generation of constructs, protein purification, labeling and western blotting can be found in Supplementary Information.

### Topological cohesin loading assay

Topological loading of cohesin onto DNA was performed following a previously described protocol[20] with some modifications: 25 nM cohesin, 50 nM Scc2[Mis4]–Scc4[Ssl3] and 3.3 nM pBluescript II KS dsDNA were combined on ice in reaction buffer (35 mM Tris−HCl pH 7.5, 1 mM MgCl$_2$, 1 mM tris(2-carboxyethyl)phosphine (TCEP), 15% (w/v) glycerol and 0.003% (w/v) Tween20). Addition of 0.5 mM ATP initiated the reaction, which was incubated at 30 °C for 120 min, shaking at 1,000 rpm. The reaction was terminated by adding 500 µl of ice-cold Wash Buffer A (35 mM Tris−HCl pH 7.5, 500 mM NaCl, 0.5 mM TCEP, 10 mM ethylenediaminetetraacetic acid (EDTA), 5% (w/v) glycerol and 0.35% (w/v) Triton X-100). Anti-Pk antibody was added to protein A-conjugated Dynabeads (ThermoFisher) and allowed to adsorb by rotating the beads at 4 °C for at least 1 h. The anti-Pk-coated magnetic beads were added to the terminated reactions and allowed to bind while rocking at 4 °C for 2 h. The beads were then washed once with Wash Buffer B (35 mM Tris−HCl pH 7.5, 750 mM NaCl, 0.5 mM TCEP, 10 mM EDTA, 5% (w/v) glycerol and 0.35% (w/v) Triton X-100) and three times with Wash Buffer A. The cohesin-bound DNA was eluted from the beads using 15 µl of elution buffer (35 mM Tris−HCl pH 7.5, 50 mM NaCl, 1 mM EDTA, 0.75% sodium dodecyl sulfate (SDS) and 1 mg ml$^{-1}$ Protease K (TaKaRa)) and heating the sample at 50 °C for 20 min. The cohesin-bound DNA was separated from the magnetic beads and the DNA was analyzed using 0.8% agarose gel electrophoresis in Tris acetate EDTA (TAE) buffer, then stained with SYBR Gold (Thermo Fisher Scientific). Gel images were recorded using a Typhoon FLA 9500 biomolecular imager (GE Healthcare), and ImageJ was used to quantify the band intensities.

### Microfluidics device preparation

To perform single-molecule experiments, a microfluidics system was adapted from previously described work[16,17]. Flow cells were assembled using parafilm to generate microfluidics channels between glass slides and a hydrophobic coverslip. Gentle heat after assembly ensured slight melting of the parafilm to keep the channels in place. Each glass slide (VWR, 26 × 76 × 1.0 mm) was drilled with holes, into which metal tubings (New England Small Tube Corp) were inserted and glued with epoxy (Devcon), with the metal tubings forming the inputs and the outputs for the channels. The drilled glass slides were reused for experiments after a cleaning procedure: sonication in 100% acetone for 15 min, sonication in 100% ethanol for 15 min, blow-drying using compressed nitrogen air and plasma cleaning for 5 min.

The coverslips (Marienfeld, high-precision, 24 × 60 mm) were first cleaned by sonication in 100% acetone for 15 min, and sonication in 100% ethanol for 15 min followed by ten rinses with purified water. The coverslips were then blow-dried using compressed air (80% nitrogen) and plasma cleaned for 5 min on each side. To make them hydrophobic, the coverslips were silanized using a solution of 5% dichloromethylsilane (Sigma-Aldrich) dissolved in heptane (Sigma-Aldrich) for an incubation time of 60 min at room temperature. The silanized coverslips were then washed using two rounds of 5-min sonication in chloroform (Sigma-Aldrich) followed by a 5-min sonication in water. After a final sonication in chloroform, the coverslips were blow-dried with compressed nitrogen air.

### Single-molecule experiments

Flow cells were first extensively washed with a total of 600 µl of phosphate-buffered saline (PBS). Solutions were introduced at a rate of 100 µl min$^{-1}$ unless otherwise stated. For visualization-only experiments, flow cells were incubated for at least 30 min with 150 µl of anti-digoxigenin antibody (Anti-Digoxigenin-AP Fab fragments,

Roche, 150 U) diluted 50 times in PBS. For optical tweezers experiments, flow cells were incubated for at least 30 min with 150 µl of Avidin-DN (VectorLabs) diluted 50 times in PBS. Flow cells were again washed with 600 µl of PBS. To limit background fluorescence derived from nonspecific protein interactions with the surface, the flow cells were passivated using a 1% solution of Pluronic F-127 (Sigma-Aldrich) in PBS for 10 min, washed with PBS and further incubated with 10 mg ml$^{-1}$ β-casein (Sigma-Aldrich) in PBS for at least 2 h.

Before binding of the 48.5 kb λ-DNA (New England Biolabs) to the surface, the nucleic acid was modified as follows. For visualization-only experiments, λ-DNA was labeled with digoxigenin at its termini in a 50-µl reaction mixture containing 0.25 mg ml$^{-1}$ of λ-DNA, 1× Standard Taq buffer (New England Biolabs), 0.5 µl of Taq DNA polymerase (New England Biolabs), 1 mM dATP, dCTP, dGTP (Promega) and dUTP–digoxigenin (Jena Bioscience), then incubated at 72 °C for 30 min.

For optical tweezers experiments, λ-DNA was modified by annealing the following biotinylated oligonucleotides to the ends of the DNA:

$$\text{aggtcgccgccc[BiotinTg]} \qquad \text{(MR1)}$$

$$\text{gggcggcgacct[BiotinTg]} \qquad \text{(MR2)}$$

Modified λ-DNAs were purified to remove unincorporated nucleotides and primers using a spin column (Bio-Rad, Micro Bio-Spin P30). The concentration of the modified λ-DNA was determined by measuring absorbance at 260 nm using a Nanodrop ND1000 spectrophotometer.

After passivation of the surface, 150 µl of a 10 pM solution of either biotinylated (optical tweezers) or digoxigenin-labeled (visualization-only) λ-DNA diluted in PBS was introduced into the channel at a rate of 20 µl min$^{-1}$ using a syringe apparatus (Harvard Apparatus, Pico Plus Elite 11). Excess λ-DNA was washed out at a rate of 50 µl min$^{-1}$ with 150 µl of PBS and to block any free streptavidin molecules on the surface (optical tweezers), we incubated 5 mM biotin for 10 min. The flow cell was then equilibrated with a further 150 µl of R buffer (35 mM Tris−HCl pH 7.5, 50 mM NaCl, 1 mM MgCl$_2$, 15% (w/v) glycerol, 1 mM TCEP, 1% (w/v) glucose, 0.2 mg ml$^{-1}$ glucose oxidase and 35 µg ml$^{-1}$ catalase).

For single cohesin loading of the cohesin variants where the force was applied directly to the complex, either via the heads or via the hinge domain, 0.5 nM TMR-labeled cohesin, 1 nM Scc2[Mis4]–Scc4[Ssl3] and 0.5 mM ATP in R buffer supplemented with 0.1 mg ml$^{-1}$ β-casein were applied. For Smc3[Psm3]–kleisin cohesin and for hinge-crosslinked cohesin, 0.5 nM or 1.5 nM cohesin, respectively, 2 nM Scc2[Mis4]–Scc4[Ssl3], 0.5 mM ATP in R buffer supplemented with 0.1 mg ml$^{-1}$ β-casein were used. The reactions were introduced into the flow cell at a rate of 15 µl min$^{-1}$ and incubated for 15 min. The flow cell was then washed at a rate of 30 µl min$^{-1}$ with 150 µl of R buffer, followed by 150 µl of S buffer (35 mM Tris−HCl pH 7.5, 500 mM NaCl, 1 mM MgCl$_2$, 15% (w/v) glycerol, 1 mM TCEP, 1% (w/v) glucose, 0.2 mg ml$^{-1}$ glucose oxidase and 35 µg ml$^{-1}$ catalase) and 150 µl of T buffer (35 mM Tris−HCl pH 7.5, 130 mM NaCl, 1 mM MgCl$_2$, 15% (w/v) glycerol, 1 mM TCEP, 1% (w/v) glucose, 0.2 mg ml$^{-1}$ glucose oxidase and 35 µg ml$^{-1}$ catalase). Only for experiments in which the Smc3[Psm3]–kleisin variant was crosslinked after cohesin loading onto DNA, 30 nM of the SpyCatcher−SpyCatcher crosslinker was introduced into the flow cell at a rate of 15 µl min$^{-1}$ and incubated for 10 min. Imaging was performed in U buffer (35 mM Tris−HCl pH 7.5, 75 mM NaCl, 1 mM MgCl$_2$, 15% (w/v) glycerol, 1 mM TCEP, 1% (w/v) glucose, 0.2 mg ml$^{-1}$ glucose oxidase and 35 µg ml$^{-1}$ catalase) unless otherwise stated. In a subset of experiments, the tethered λ-DNA was stained by adding 2 nM Sytox Orange (Invitrogen) to U buffer.

For second DNA capture experiments, cohesin loading was performed using 1.5 nM LD655-labeled cohesin, 2 nM Scc2[Mis4]–Scc4[Ssl3], 0.5 mM ATP in R buffer supplemented with 0.1 mg ml$^{-1}$ β-casein, introduced into the flow cell at a rate of 15 µl min$^{-1}$. After a 15-min incubation, excess cohesin was washed at a rate of 30 µl min$^{-1}$ with 150 µl of S buffer followed by 150 µl of T buffer. Then 0.5 nM of the second DNA substrate

was mixed with 2 nM Scc2$^{Mis4}$–Scc4$^{Ssl3}$ and 0.5 mM ATP in U buffer and incubated at 30 °C for 5 min. The reaction mixture was introduced at a rate of 10 μl min$^{-1}$ and incubated into the flow cell for 10 min. The flow cell was then washed at 10 μl min$^{-1}$ with 150 μl of either S or T buffer and imaged in U buffer, unless otherwise stated.

The DNA substrates used for second DNA capture were 7.2 kb M13mp18 ssDNA (New England Biolabs) and M13mp18 I RF dsDNA (New England Biolabs), which were covalently modified with either MFP488 fluorophores (visualization-only) or biotin labels (optical tweezers), using Label IT nucleic acid labeling kits (Mirus). The MFP488 fluorophore (488 nm excitation) on second DNA substrates allowed for simultaneous visualization with LD655–cohesin (647 nm excitation) and, where necessary, with λ-DNA stained with Sytox Orange (532 nm excitation). The protocol supplied for labeling was modified by reducing the amount of Label IT reagent to 0.1 μl μg$^{-1}$, which resulted in roughly 1 fluorophore/biotin label per 120 bp.

For ssDNA-to-dsDNA conversion experiments, the ssDNA used was a 3 kb phage-derived pBluescript II KS (+) plasmid[29], modified by annealing the following biotin-containing oligonucleotides for conjugation of three fluorescent labels:

caaccaagtca[BiotindT]tctgagaatagtgtatgc (MR3)

tcagctcca[BiotindT]ggtcc (MR4)

cttgaag[BiotindT]ggtggcctaactacgg (MR5)

In these experiments, oligonucleotides were annealed to the DNA as opposed to chemically attaching several fluorescent labels to minimize steric hindrance for DNA polymerase.

The plasmid was then incubated with Qdot 525 Streptavidin Conjugate (Invitrogen) overnight at 25 °C. Following, 200 μM biotin was incubated for 60 min at 25 °C with the plasmid and excess Qdot-streptavidin/biotin eliminated using an Illustra S-400 HR Micro-Spin column (GE Healthcare). Second DNA capture was performed as described above. After removing excess ssDNA, 120 μl of a mixture containing 0.1 unit μl$^{-1}$ of T7 DNA polymerase (New England BioLabs) and 1 mM deoxynucleotide triphosphates (Sigma-Aldrich) in U buffer was added to the flow cell at a rate of 15 μl min$^{-1}$. The mixture was incubated for 15 min, and the flow cell washed with 150 μl of S or T buffer supplemented with 2 nM of Sytox Orange.

## Optical tweezers

Optical tweezer experiments were carried out on a JPK NanoTracker 2 system integrated with a custom-made TIRF microscope. For binding of cohesin or second DNA substrates to beads, biotin–streptavidin interactions were selected as they form the strongest known non-covalent interactions and can resist very high forces (160–400 pN) (refs. 35–37). The beads used for the experiments were either 0.5 μM streptavidin-coated beads (Bangs Lab) (1% solids w/v) or 1 μm polystyrene beads coated with streptavidin (Bangs lab) (1% solids w/v), which attached to the covalently conjugated biotin label on either cohesin or on the second DNA substrate, respectively. Before introduction into the flow cell, the beads were diluted 20 times into T buffer supplemented with 0.1 mg ml$^{-1}$ of β-casein, washed three times (spun at 10,000$g$, at 4 °C for 15 min) and resuspended into the same buffer.

Beads were introduced into the flow cells at a rate of 5 μl/min and incubated for at least 15 min before being washed out with T buffer at the same rate. All optical trapping measurements were performed and recorded in T buffer. Force ramp measurements were all conducted at a constant speed of 0.16 μm s$^{-1}$ in the $y$ direction. Most experiments were performed in the absence of any DNA dye, but in experiments were the DNA and beads were visualized (such as force application on the second DNA), 2 nM Sytox Orange was added to the T buffer used to wash out excess beads. Addition of low concentrations of Sytox

Orange were also used to confirm the integrity of the λ-DNA, without unduly altering its physical properties. In experiments performed in the presence of ATP, 1 mM of the nucleotide was added to T buffer as excess beads were washed out of the flow cell.

## Data collection

FD data were processed using JPK Processing software (JPK NanoTracker version 6.1). For imaging, fluorescence emission was collected by an Andon iXon Life 888 EMCCD camera running at maximum electron multiplying gain. Typical exposure time was 50 ms. The Andor Solis software (version 4.31) was used to acquire and record microscopy data.

Experiments not requiring optical trapping were performed on a Nikon Eclipse Ti2 commercial TIRF microscope with a sCMOS camera (Photometrics, Prime 95B). The Nikon NIS-Elements software (version 5.41) was used to acquire microscopy data. The frame rate used varied depending on the experiment, ranging from 0.5 frames s$^{-1}$ to 1 frame every 2 s. Images were saved as TIFF files without compression and further analyzed using ImageJ.

## Data processing

Images and videos were analyzed using FIJI ImageJ software. The length of the DNA molecules was manually measured converting distance in pixels into kb, and kymographs were generated to observe colocalization of differentially labeled fluorescent molecules. Photobleaching data were first visually analyzed using ImageJ and later quantified using a custom-made MATLAB script generated to extract EMCCD counts following subtraction of background fluorescence. Analysis of the FD curves was performed using the JPK Processing Software and illustrated using MATLAB. Forces of rupture of each experiment were manually tracked and plotted using MATLAB illustrating contributions for both $x$ and $y$ directions. For statistical comparison between different groups of force rupture distributions, MATLAB was used to run a Welch $t$-test (unequal variance $t$-test) as the populations followed a normal distribution but did not have the same variance. Images were processed using Adobe Illustrator CC.

## Monte-Carlo simulations

To simulate the distribution of rupture forces, we considered the λ-DNA attachment geometry as occurring in our single-molecule assay (Extended Data Fig. 4a). A DNA molecule with a persistence length of 50 nm and a contour length of 16.32 μm was assumed to be tethered to the surface by its two ends. The distance between the ends ($D$) was assumed to be randomly distributed with mean value 8.8 μm and standard deviation 0.5 μm, both of which were inferred from experimental data. Cohesin was assumed to be positioned at the center of the DNA during force-application experiments. The time step in simulations was varied between 0.1 and 0.01 s. At each time step, DNA was moved with respect to cohesin in the direction perpendicular to the DNA attachment points, with the experimental force loading rate of 0.16 μm s$^{-1}$ (Extended Data Fig. 4a–c).

The resulting DNA length at the time point was calculated from the updated position of cohesin:

$$L_{DNA} = 2\sqrt{\left(\frac{D}{2}\right)^2 + x^2}, \tag{1}$$

where $x$ is the distance between the cohesin anchor point and the segment connecting the DNA tethering points (Extended Data Fig. 4a).

Next, we calculated DNA tension and the force acting on cohesin at each time step:

$$F_{DNA} = \frac{k_B T}{P}\left(\frac{1}{4\left(1 - \frac{L_{DNA}}{L_0}\right)^2} - \frac{1}{4} + \frac{L_{DNA}}{L_0}\right), \tag{2}$$

$$F_{\text{cohesin}} = 2 \times F_{\text{DNA}} \sin \alpha, \tag{3}$$

where $P$ is DNA persistence length, $L_{\text{DNA}}$ is current DNA length, $L_0$ is DNA contour length, and $\tan \alpha = 2\frac{x}{D}$.

The rate of cohesin disengagement at the current tension was determined as

$$k_{\text{diss}}(F) = k_0 \exp(\delta \times F_{\text{cohesin}}/k_{\text{B}}T). \tag{4}$$

Finally, the detachment time was calculated as

$$t_{\text{detach}} = -\frac{1}{k_{\text{diss}}(F)} \ln(1 - r), \tag{5}$$

where $r$ is a random number evenly distributed in the interval [0;1]. Cohesin was considered disengaged at the current timestep if $t_{\text{detach}} \leq \mathrm{d}t$.

Once the rupture occurred, the current value of the rupture force was recorded, and the simulation repeated over ~$10^4$ times to collect distribution of rupture forces. Average rupture forces ($\mu$) and standard deviations ($\sigma$) corresponding to different sets of parameters were calculated for the force distributions for different $k_0$ and $\delta$ values (Extended Data Fig. 4d–f).

To simulate rupture events when the force was applied to the second DNA we used a similar approach, but with some modifications. In this case, the relationship between DNA lengths and forces is given by the following system of equations:

$$\begin{cases} F_1 = \frac{k_{\text{B}}T}{P}\left( \frac{1}{4\left(1 - \frac{L_1}{L_0}\right)^2} - \frac{1}{4} + \frac{L_1}{L_0} \right) \\[2ex] F_2 = \frac{k_{\text{B}}T}{P}\left( \frac{1}{4\left(1 - \frac{L_2}{L_0}\right)^2} - \frac{1}{4} + \frac{L_2}{L_0} \right), \\[2ex] \qquad F_1 = 2F_2 \sin \alpha \\[1ex] \qquad \frac{L_2}{2} \sin \alpha + L_1 = x \\[1ex] \qquad tg\alpha = 2\frac{x - L_1}{D} \end{cases} \tag{6}$$

where $F_1$, $L_1$, $F_2$ and $L_2$ are the forces and lengths of the first and second DNA, respectively; $\frac{\pi}{2} - \alpha$ is the angle between the two DNAs at the point of contact; and $x$ is the distance between the cohesin point and the segment between the two tethering points of the first DNA (Extended Data Fig. 7d).

This nonlinear system of equations was solved numerically at each time step to calculate the tension applied to DNA, the DNA length for both molecules as well as the force applied to cohesin. Next, the disengagement rate and time of rupture were calculated as in equations (4) and (5).

For generating the parameters in Fig. 4e, $k_0$ was $2 \times 10^{-3}$ as in Fig. 2a. To simulate possible different orientations of cohesin, $\delta$ was chosen randomly to be either 1.23 nm or 1.61 nm with equal probability for each simulated FD curve.

## Model fitting and parameter determination
Since our model is nondifferentiable, we used two-dimensional golden search algorithm to find sets of $k_0$ and $\delta$ parameters that minimized the least square distance between the means and variances of the experimental and simulated distributions of rupture forces. This algorithm converges to a single optimal solution because the dependence of both mean and variance of the rupture forces on $k_0$ and $\delta$ is monotonic (Extended Data Fig. 4e).

To determine the 90% confidence interval for the parameters, we calculated the maximum likelihood distribution for $k_0$ and $\delta$ directly by

sampling it using a Monte-Carlo approach. The individual probabilities were defined as

$$P_i = \exp\left\{ -\frac{(\mu_i - \text{mean}(F))^2}{\text{var}_{\text{mean}}} - \frac{(\sigma_i - \text{std}(F))^2}{\text{var}_{\text{std}}} \right\}. \tag{7}$$

Here $\mu$ and $\sigma$ are the mean and standard deviations of the simulated distributions, $F$ is the set of experimentally measured forces, and $\text{var}_{\text{mean}}$ and $\text{var}_{\text{std}}$ are the experimentally determined variances in the mean and standard deviations, respectively.

To determine the confidence interval for the parameter $k_0$, we sampled the partial distribution by calculating the probabilities directly using equation (7) for randomly samples values of $k_0$ while keeping the value $\delta$ fixed at its optimum determined by the golden-search algorithm. 90% confidence interval was then determined directly from the distribution as a region under the curve covering 90% of the total area. For determining the confidence interval for $\delta$ we used the same algorithm except $\delta$ was now sampled and $k_0$ was kept constant at its optimal value. Unless stated otherwise, values in the main text in brackets following the value of a parameter indicate 90% confidence interval determined by this method.

For Extended Data Fig. 4h the standard deviation in determining $k_0$ and $\delta$ was calculated directly as the standard deviation between parameters determined for the different subsamples.

Confidence intervals determined by the Monte-Carlo sampling were consistent with the standard deviations determined for the sub-sampled data on Extended Data Fig. 4h;

$P$ values in Extended Data Fig. 4i were calculated using the Kolmogorov–Smirnov test.

## Determination of the average number of cohesin molecules per bead
To infer the distribution of the number of cohesins per bead from the experimentally obtained distributions for the number of cohesins and beads per DNA, we used the following algorithm. First, we introduced parameter $\lambda$—efficiency of the bead–cohesin interaction. Our sampling procedure typically consisted of 10,000 steps; and at each step, we picked a DNA with cohesin distributed according to a sample randomly chosen from the experimentally measured distribution (Extended Data Fig. 2a) and assumed that each cohesin can bind a bead with probability $\lambda$. In case of successful binding, the bead had one cohesin attached to it. If there were other cohesins on this DNA, we assumed that the same bead binds them with 100% efficiency if they were spatially closer than 1 μm to the original cohesin, which resulted in the increased the number of cohesins attached to this bead. If other cohesins on DNA were further away than 1 μm from the initial cohesin, we assumed that they could also bind to the same bead, but with probability $\lambda$, which again would increase the number of cohesins bound to the bead.

This sampling yielded distributions of the number of cohesins per bead and the number of beads per DNA as a function of $\lambda$. We then varied $\lambda$ to match the number of beads per DNA to the experimentally observed value (Extended Data Fig. 2a), which yielded the value of $\lambda$ corresponding to our experimental conditions. The number of cohesins per bead for this $\lambda$ gave us the corresponding distribution of the number of cohesins per bead in our experiments (Extended Data Fig. 2, right).

## Reporting summary
Further information on research design is available in the Nature Portfolio Reporting Summary linked to this article.

## Data availability
Source data are provided with this paper. The rest of the raw data will be made available by the authors upon request.

## Code availability

Software used for the data processing and MMC simulations is freely available in the GitHub repository (https://github.com/FrancisCrickInstitute/DNA_Cohesin_MMC).

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

## Acknowledgements

We thank B. Hoogenboom for discussions, members of both laboratories for critical reading of the manuscript and M. Tang for single-molecule experiment discussions. This study was supported by the European Research Council grant AdG 670412 to F.U. and the Francis Crick Institute, which received funding from the UK Medical Research Council, Cancer Research UK and the Wellcome Trust through awards FC001198 to F.U. and FC001750 to M.I.M. This study was also supported through a fellowship from Boehringer Ingelheim Fonds to M.R. The funders had no role in study design, data collection and interpretation, or the decision to submit the work for publication.

## Author contributions

M.R., F.U. and M.I.M. conceived and designed the study. M.R. performed experiments and analyzed the data. G.P. helped with setting up optical trapping assays. M.R., T.L.H. and K.G. cloned, expressed and purified the proteins. M.I.M. performed simulations. M.R., F.U. and M.I.M. wrote the manuscript.

## Funding

## Competing interests

The authors declare no competing interests.

## Additional information

**Extended data** is available for this paper at https://doi.org/10.1038/s41594-023-01122-4.

**Correspondence and requests for materials** should be addressed to Frank Uhlmann or Maxim I. Molodtsov.

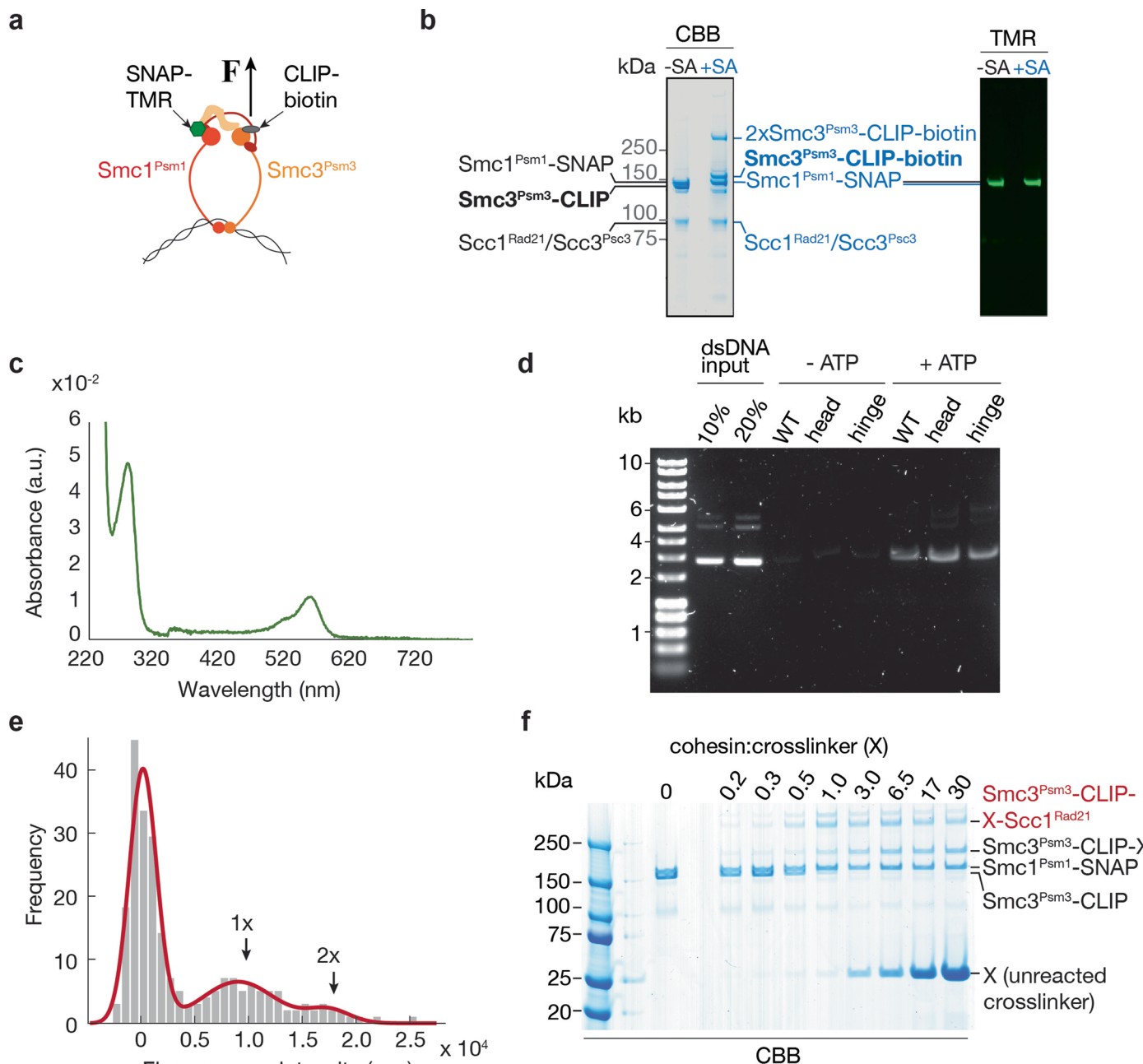

**Extended Data Fig. 1 | Characterization of the recombinant cohesin complexes and topological DNA loading. a**, Schematic representation of the cohesin variant modified at the head domain with the CLIP-tag for biotin conjugation for force application and the SNAP-tag for fluorophore attachment at the C-termini of Smc3^Psm3 and Smc1^Psm1, respectively. **b**, Left, sodium dodecyl-sulphate polyacrylamide gel electrophoresis (SDS-PAGE) gel stained with Coomassie Brilliant Blue (CBB) showing the purified subunits of the cohesin complex (-streptavidin; -SA) and confirmation of biotin attachment to the CLIP-tag, as all the detectable Smc3^Psm3-CLIP-biotin subunit shifts upon addition of streptavidin (+streptavidin; +SA). Note that the additional 2x Smc3^Psm3-CLIP-biotin is the result of two separate Smc3^Psm3-CLIP-biotin subunits binding to one single streptavidin molecule, as the latter bears four binding sites for biotin. Also note that Scc1^Rad21 and Scc3^Psc3 migrate at the same molecular weight. Right, in-gel TMR fluorescence of the Smc1^Psm1-SNAP subunit, confirming successful TMR attachment. This result was obtained in more than three independent protein purifications. **c**, Example absorbance spectrum of 150 nM TMR-cohesin. The labelling efficiency was ~95%, assuming the molar extinction coefficient for the TMR dye of 65,000 M⁻¹cm⁻¹. **d**, Topological cohesin loading assay on DNA.

Comparison to the wild-type protein (WT) shows that the SNAP- and CLIP-labelled cohesin variants with the biotin tag attached either *via* the head domain (head) or the hinge domain (hinge) were equally proficient in ATP-dependent, bulk biochemical loading onto double-stranded DNA. This experiment was repeated twice. **e**, Brightness distribution of background-subtracted fluorescence intensity of TMR-cohesin molecules diffusing along the tethered λ-DNA in a typical field of view. The first peak centred at 0 a.u. represents the normalised intensity of the background fluorescence, without any fluorescently-labelled cohesin. As seen in Fig. 1c, the peak at ~1.0 × 10⁴ a. u. corresponds to one single fluorophore, whereas the one at ~2.0 × 10⁴ a. u. represents a small population of double fluorophores. **f**, SDS-PAGE gel illustrating bulk Smc3^Psm3-kleisin crosslinking mixed with increasing crosslinker concentrations (range 0.05 – 10 μM, the molar ratio relative to cohesin is displayed). Band identities are indicated. Note, this is a bulk experiment without DNA, where crosslinker can couple different cohesin molecules. This does not happen on isolated cohesin on single DNA molecules in the microscopy setup. Two independent repeats were performed.

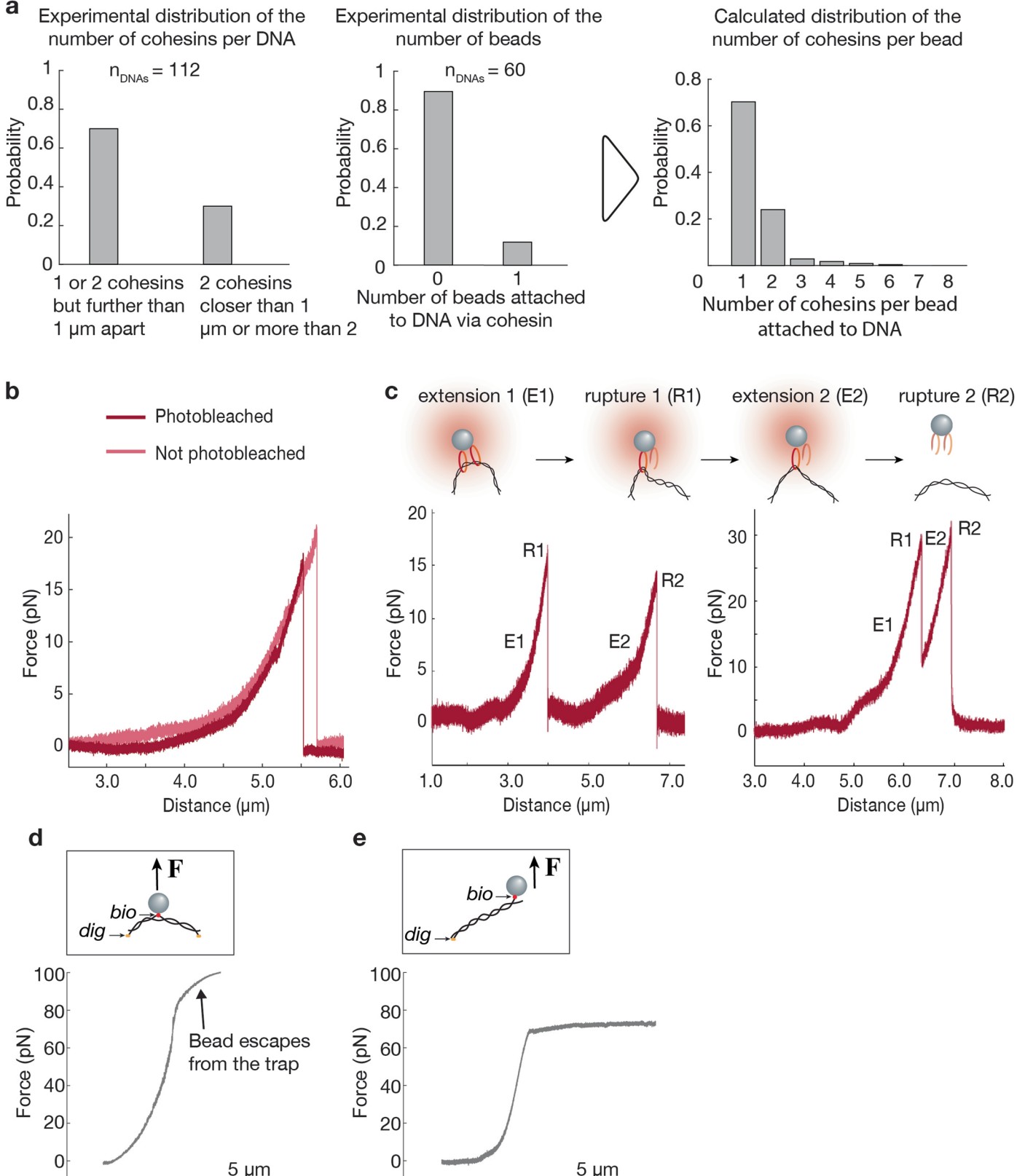

**Extended Data Fig. 2 | Additional characterization of the cohesin-DNA interaction. a**, Distribution of the number of cohesins per λ-DNA and number of attached beads per λ-DNA. The corresponding distribution of the number of cohesins per bead (right) is derived from the two experimental distributions on the left (see Methods). **b**, Examples of force-distance curves for cohesin molecules bound to λ-DNA which were either photobleached or not prior to force application, showing photobleaching has no effect on the shape of the curve and the rupture force. **c**, Examples of two force-distance curves exhibiting two rupture peaks, indicative of two cohesins detaching from the same DNA that were either closer or further apart. The corresponding graphical representation is shown on top. **d**, Graphical representation of the experiment (top) and the measured force-distance curve (bottom) of a bead attached to λ-DNA via covalently attached biotin instead of cohesin, showing escape of the bead from the trap when the force exceeds 80 pN. **e**, Graphical representation of the experiment (top) and the measured DNA force-extension (bottom), showing the characteristic melting curve for λ-DNA at 65 pN force.

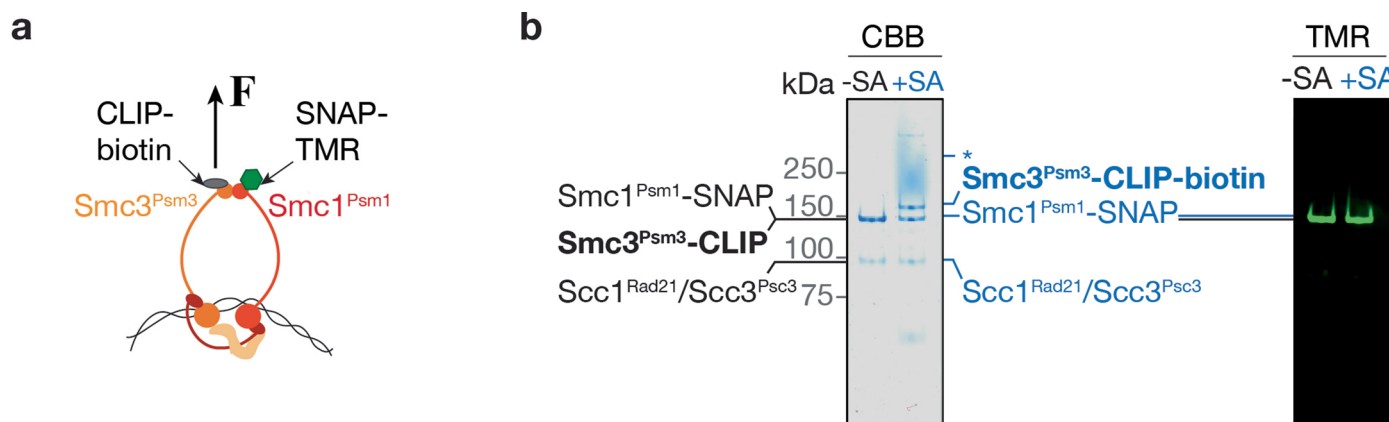

**Extended Data Fig. 3 | Characterization of cohesin with a biotin handle at the hinge. a**, Schematic representation of cohesin, with the CLIP-tag for biotin conjugation and the SNAP-tag for dye attachment located within the hinge region of Smc3$^{Psm3}$ and Smc1$^{Psm1}$, respectively. **b**, Left, SDS-PAGE gel stained with CBB showing the purified subunits of the cohesin complex (-SA) and confirmation of CLIP-tag biotin attachment through the shift of the Smc3$^{Psm3}$-CLIP-biotin subunit in the presence of streptavidin ( + SA). The smear (\*) is likely indicative of more than one Smc3$^{Psm3}$-CLIP-biotin subunit coupled to a streptavidin tetramer. Right, in-gel TMR fluorescence of the Smc1$^{Psm1}$-SNAP subunit, confirming successful TMR attachment. This result was obtained in at least three independent protein purifications.

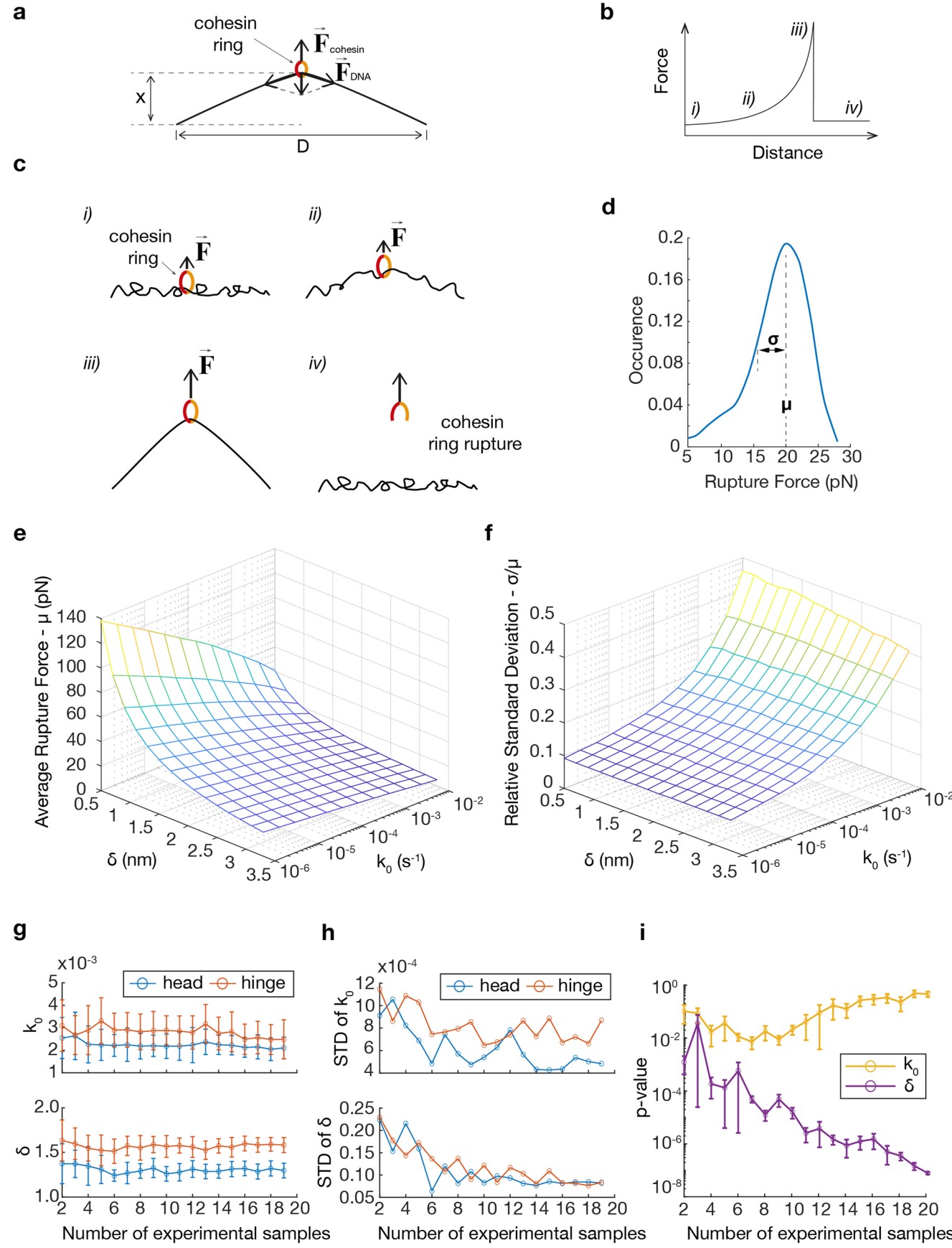

**Extended Data Fig. 4 | See next page for caption.**

**Extended Data Fig. 4 | Monte-Carlo simulations of rupture forces.**
**a**, Schematics of the geometry used for the simulation. D is distance between the DNA anchor points. As $x$ increases during the force-distance measurement, the force acting on cohesin ($F_{cohesin}$) increases. This force is balanced out by the tension ($F_{DNA}$) which stretches the DNA. **b**, Theoretical force-distance curve that shows different stages of DNA extension during force application on cohesin: $i$) and $ii$) extension at low forces; $iii$) extension at high forces that stretch DNA close to its contour length and $iv$) disengagement of the cohesin ring from DNA. **c**, Cartoons illustrating cohesin and DNA at the different stages of the force-distance curve shown in panel b. **d**, A typical simulated distribution of rupture forces based on $10^4$ simulated force-distance curves. $\mu$ – average force and $\sigma$ – standard deviation of the distribution. **e**, Average simulated force ($\mu$) is shown as a function of both parameters $k_0$ and $\delta$. **f**, Relative standard deviation ($\sigma/\mu$) of the rupture force is shown as a function of the same set of $k_0$ and $\delta$ parameters as

in panel e. **g**, Parameters $k_0$ (top) and $\delta$ (bottom) for cohesin pulled from the head or hinge (shown in blue and red) were determined by fitting subsamples of the experimental data to the model. The number of subset samples used was equal for each dataset (head and hinge) and is shown on the x-axis. The center point represents mean values and error bars represent standard deviations across 20 independent subsets. **h**, Standard deviations (STD) of the fitted parameters derived from panel g as a function of the number of experimental samples. **i**, p-values calculated for the testing hypothesis that $k_0$ and $\delta$ for the cohesin variants pulled either via the head or the hinge domain are the same for subsets of different sample size. The p-value for $\delta$ decreases, which shows that this parameter is statistically different for the two datasets, however, the p-value remains high for the $k_0$ showing that it cannot be distinguished between the two cohesin variants regardless of the increasing sample size. Error bars are SEM across eight different independent calculations.

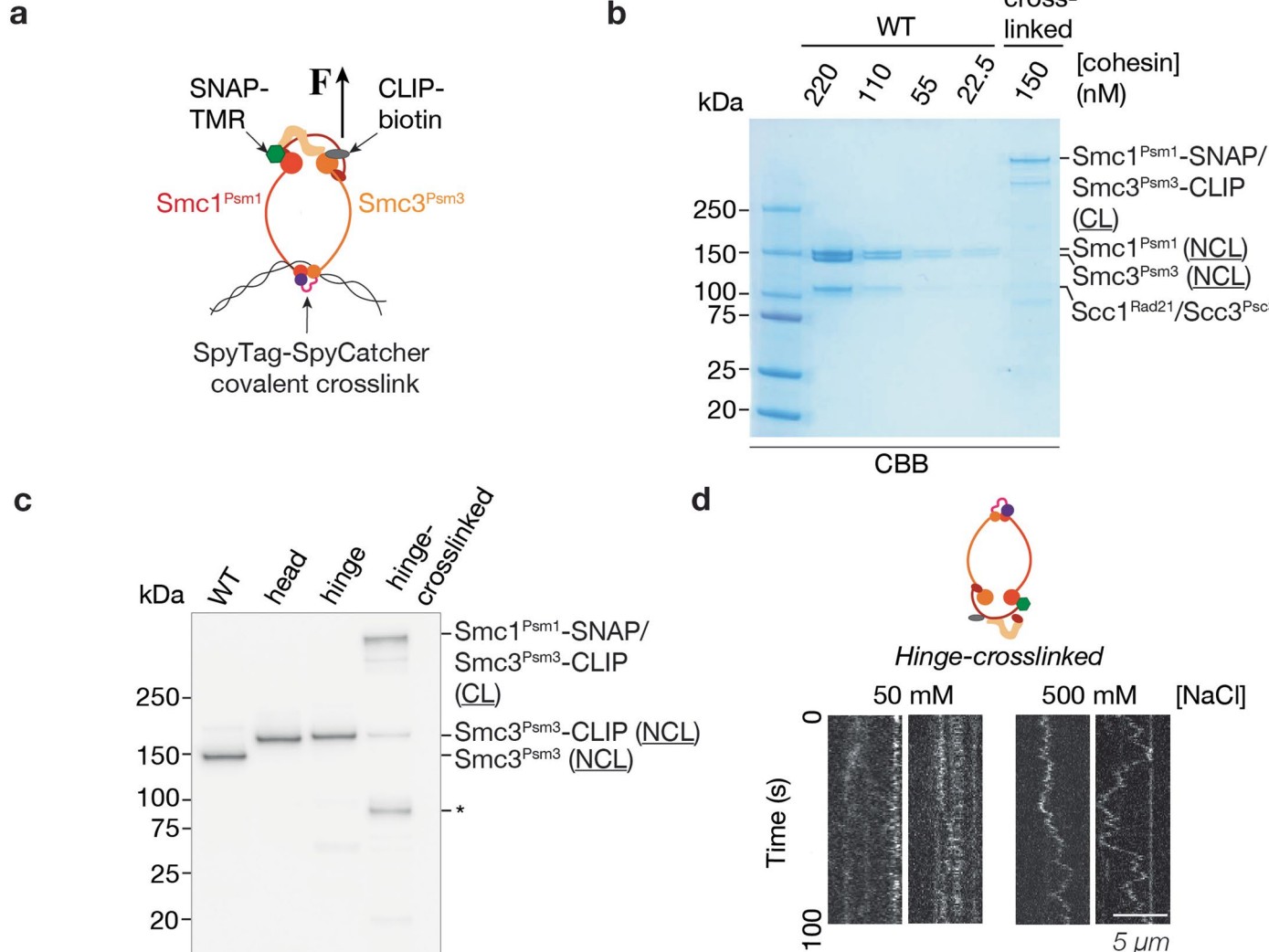

**Extended Data Fig. 5 | Characterization of the hinge-crosslinked cohesin complex. a**, Schematic representation of the hinge-crosslinked cohesin variant bearing CLIP-biotin and SNAP-TMR modifications at the C-termini of Smc3$^{Psm3}$ and Smc1$^{Psm1}$, respectively. The hinge domain is crosslinked by the SpyTag-SpyCatcher system through the creation of a covalent isopeptide bond. **b**, SDS-PAGE gel stained with CBB showing the hinge-crosslinked Smc1$^{Psm1}$-SNAP/ Smc3$^{Psm3}$-CLIP subunits (crosslinked; CL) migrating at ~300 kDa, compared to wild-type (WT) cohesin subunits (non-crosslinked; NCL) migrating at ~150 kDa at different cohesin concentrations. Additional bands in this lane are due to a small quantity of protein degradation products, often seen during the production of fusion proteins. **c**, Western blot using an anti-Pk antibody recognising the 3xPk

tag located on the Smc3$^{Psm3}$ subunit of the cohesin complex, comparing the wild type complex without SNAP- or CLIP-tags (WT), the cohesin variant with the biotin tag attached *via* the head domain (head), *via* the hinge domain (hinge), or with the hinge crosslinked (hinge-crosslinked). The crosslinking efficiency (~70%) was calculated by comparing the signal of the crosslinked Smc1$^{Psm1}$-SNAP/Smc3$^{Psm3}$-CLIP fusion subunit (CL) with the non-crosslinked Smc3$^{Psm3}$-CLIP subunit (NCL). The background band (*) is likely the result of protein degradation. **d**, Example kymographs showing hinge-crosslinked cohesin loaded onto λ-DNA after a 50 mM or 500 mM NaCl wash. More than 50 instances were observed in three independent experiments.

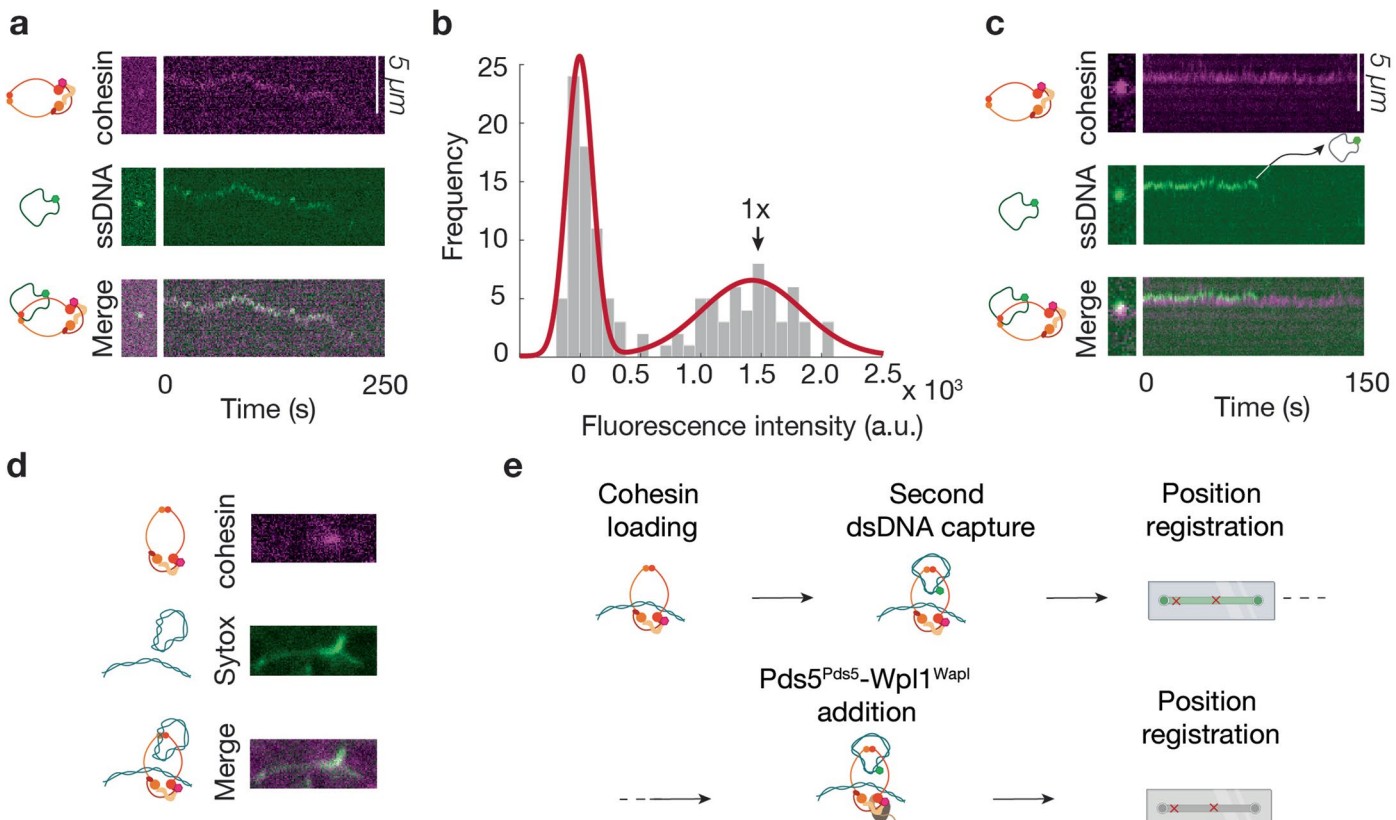

**Extended Data Fig. 6 | Characterization of the second DNA capture by cohesion. a**, Example kymographs showing LD655-labelled cohesin colocalising with the MFP488-labelled single-stranded plasmid after a 500 mM NaCl wash. Number of events observed = 14 in at least three independent experiments. **b**, Brightness distribution of background-subtracted fluorescence intensity of LD655-cohesin molecules which performed second DNA capture. The first peak centred at 0 a.u. represents the normalised intensity of the background fluorescence, without any fluorescently-labelled cohesin. As seen in Fig. 4c,

the peak at ~1.5 × 10³ a. u. corresponds to one single fluorophore. **c**, Example of kymographs showing spontaneous dissociation of MFP488-labelled ssDNA from LD655-cohesin after second DNA capture (n = 4). Relates to Supplementary Video 4. **d**, Example showing LD655-cohesin holding together the tethered λ-DNA and the second DNA substrate, first captured as ssDNA and then converted to dsDNA (confirmed by Sytox Orange staining – Sytox; n = 3). **e**, Experimental workflow for addition of the Pds5^Pds5-Wpl1^Wapl complex after second DNA capture by cohesin. Relates to Fig. 4e.

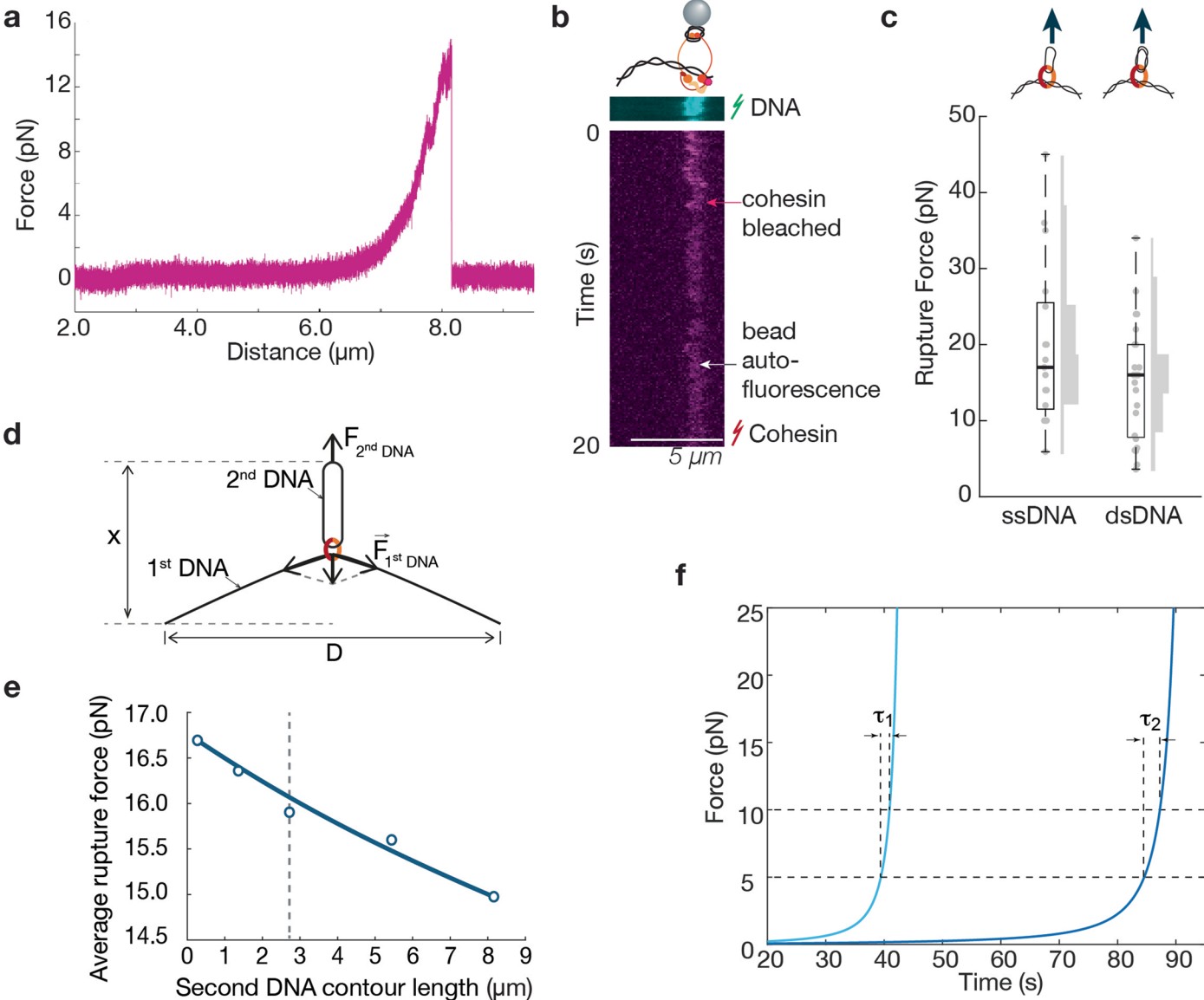

**Extended Data Fig. 7 | Disengagement at the weakest cohesin interface limits the strength of the cohesin-mediated DNA–DNA interaction. a**, Characteristic force-distance curve showing detachment of the second ssDNA at 16 pN, while λ-DNA remains attached to the surface. Relates to Supplementary Video 5. **b**, Example of kymograph showing simultaneous visualisation of a Sytox Orange-stained bead bound to the second DNA and a single LD655-cohesin. Arrow points to the moment at which cohesin photobleaches. Relates to Fig. 5d. **c**, Distribution of forces obtained when force was applied to a second DNA, as a single-stranded (n = 17) or double-stranded plasmid (n = 24). In the box plots, the central mark indicates the median, the bottom and top edges of the box indicate the 25th and 75th percentiles, respectively. The whiskers extend to the most extreme data points, none of which are considered to be outliers. **d**, Schematics of the geometry used to simulate experiments in which the force was applied to the second DNA. Notations follow Extended Data Fig. 4a. **e**, Average expected rupture force as a function of the length of the second DNA. Simulation results (blue circles) and trendline (solid line) are shown. The dashed line shows the length of the circular DNA plasmid used in experiments (7.2 kb, contour length of 2.47 μm). Other simulation parameters as in Fig. 5e. **f**, Two simulated force-distance curves are shown as a function of time for the two DNA molecules as they are being stretched, one of which is 2x longer than the other. This simulation shows that a longer DNA molecule spends more time at each force interval (for example between 5 and 10 pN indicated as $\tau_1$ and $\tau_2$, $\tau_2 > \tau_1$), therefore leading to a higher chance of cohesin disengagement before the DNA can reach higher forces.

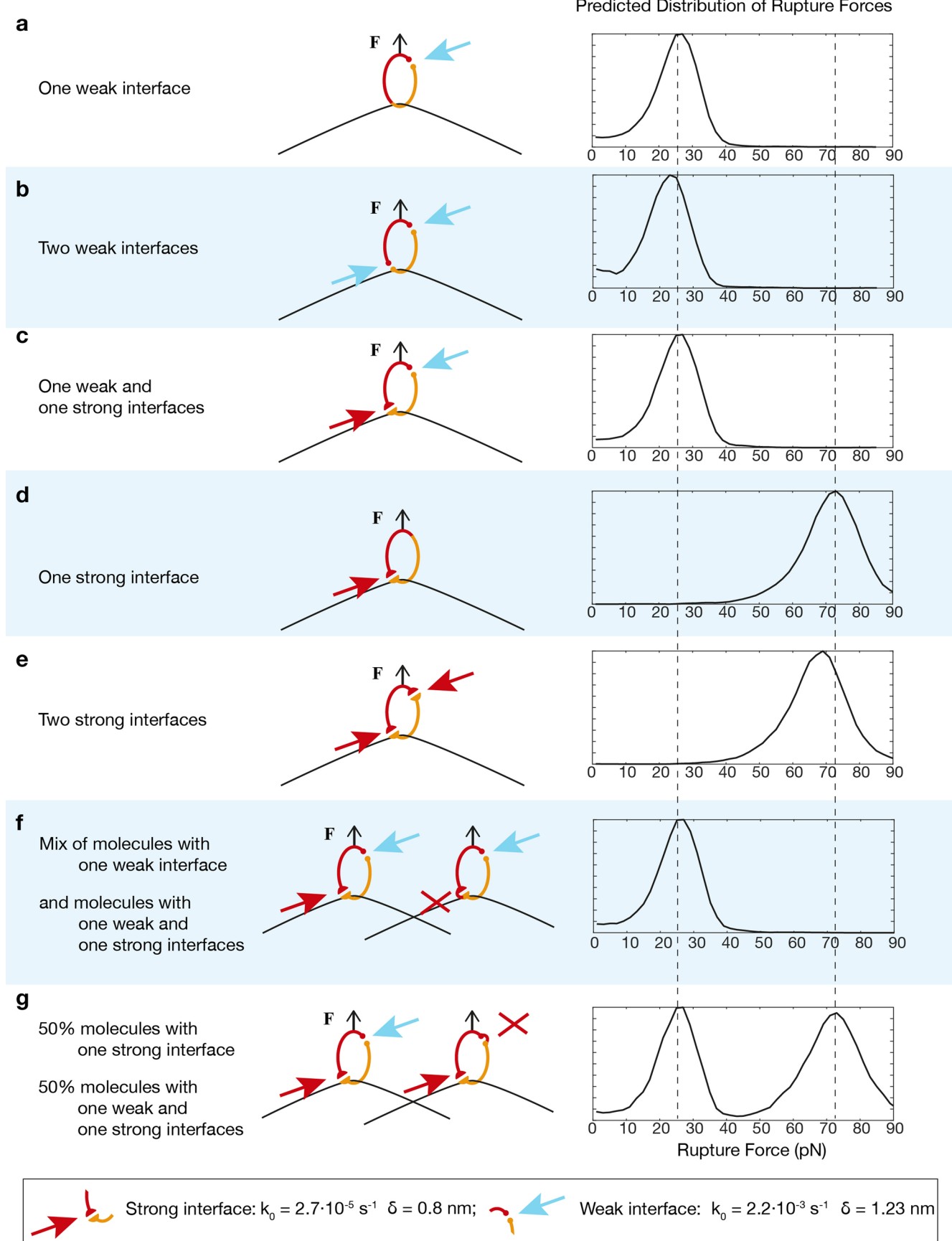

**Extended Data Fig. 8 | See next page for caption.**

**Extended Data Fig. 8 | Predicted distributions of rupture forces for cohesin complexes with two interfaces. a**, Cohesin with only one weak interface ruptures at ~20 pN force. **b**, Mechanical stability of cohesin with two weak interfaces is determined by each individual interface and it ruptures at ~20 pN when one of the interfaces opens. **c**, The mechanical stability of cohesin with one weak and one strong interface is determined by its weak interface only. **d**, Rupture at high forces is observed only when there are no weak interfaces, *for example* when there is only one strong interface. **e**, Rupture at high forces is also observed when there are only two strong interfaces. **f**, For a complex with one weak and one strong interface, covalently closing the strong interface does not affect the rupture force distribution (compare with Figs. 2c and 3b). **g**, A bimodal force rupture distribution can only be obtained when two cohesin species are present. One with both weak and strong interfaces – rupturing at lower forces – and one with only the strong interface (or the weak interface covalently closed) – rupturing at higher forces.

# Reporting Summary

## Statistics

For all statistical analyses, confirm that the following items are present in the figure legend, table legend, main text, or Methods section.

| n/a | Confirmed | |
|---|---|---|
| ☐ | ☒ | The exact sample size (*n*) for each experimental group/condition, given as a discrete number and unit of measurement |
| ☐ | ☒ | A statement on whether measurements were taken from distinct samples or whether the same sample was measured repeatedly |
| ☐ | ☒ | The statistical test(s) used AND whether they are one- or two-sided *Only common tests should be described solely by name; describe more complex techniques in the Methods section.* |
| ☐ | ☒ | A description of all covariates tested |
| ☐ | ☒ | A description of any assumptions or corrections, such as tests of normality and adjustment for multiple comparisons |
| ☐ | ☒ | A full description of the statistical parameters including central tendency (e.g. means) or other basic estimates (e.g. regression coefficient) AND variation (e.g. standard deviation) or associated estimates of uncertainty (e.g. confidence intervals) |
| ☐ | ☒ | For null hypothesis testing, the test statistic (e.g. *F*, *t*, *r*) with confidence intervals, effect sizes, degrees of freedom and *P* value noted *Give P values as exact values whenever suitable.* |
| ☐ | ☒ | For Bayesian analysis, information on the choice of priors and Markov chain Monte Carlo settings |
| ☒ | ☐ | For hierarchical and complex designs, identification of the appropriate level for tests and full reporting of outcomes |
| ☒ | ☐ | Estimates of effect sizes (e.g. Cohen's *d*, Pearson's *r*), indicating how they were calculated |

*Our web collection on statistics for biologists contains articles on many of the points above.*

## Software and code

Policy information about availability of computer code

Data collection | The JPK optical trap control software (v 6.1) was employed to operate the optical trap and collect force-distance data. The Andor Solis (v 4.31) software was used to acquire and record microscopy data simultaneous with force application/measurements by the JPK software. The Nikon NIS-Elements (v 5.41) software was used to image microscopy data obtained without force application experiments.

Data analysis | The JPK Processing (v 6.1) Software was used to visualise force-distance curves and extract the magnitudes of rupture forces. FIJI (ImageJ, v 1.54) was used for visualisation and image analysis. Custom-made MATLAB codes were used to for single-molecule fluorescence intensity analysis, force rupture determination and visualisation, histogram fitting, statistical testing and simulations. All codes are publicly available on GitHub (https://github.com/FrancisCrickInstitute/DNA_Cohesin_MMC).

For manuscripts utilizing custom algorithms or software that are central to the research but not yet described in published literature, software must be made available to editors and reviewers. We strongly encourage code deposition in a community repository (e.g. GitHub). See the Nature Portfolio guidelines for submitting code & software for further information.

## Data

Policy information about availability of data

All manuscripts must include a data availability statement. This statement should provide the following information, where applicable:
- Accession codes, unique identifiers, or web links for publicly available datasets
- A description of any restrictions on data availability
- For clinical datasets or third party data, please ensure that the statement adheres to our policy

Source data used for the generation of main figures is provided with this paper. Example data is included with the software codes provided freely on GitHub. The rest of the raw data will be made available by the authors upon request.

## Research involving human participants, their data, or biological material

Policy information about studies with human participants or human data. See also policy information about sex, gender (identity/presentation), and sexual orientation and race, ethnicity and racism.

| | |
|---|---|
| Reporting on sex and gender | N/A |
| Reporting on race, ethnicity, or other socially relevant groupings | N/A |
| Population characteristics | N/A |
| Recruitment | N/A |
| Ethics oversight | N/A |

Note that full information on the approval of the study protocol must also be provided in the manuscript.

# Field-specific reporting

Please select the one below that is the best fit for your research. If you are not sure, read the appropriate sections before making your selection.

☒ Life sciences ☐ Behavioural & social sciences ☐ Ecological, evolutionary & environmental sciences

For a reference copy of the document with all sections, see nature.com/documents/nr-reporting-summary-flat.pdf

# Life sciences study design

All studies must disclose on these points even when the disclosure is negative.

| | |
|---|---|
| Sample size | Sample size was not predetermined. We collected data until further increase in the sample size either did not directly improve variances of the k0 and delta parameters inferred from the data or until the bootstrap analysis indicated that the further increase was not expected to change the significance of the difference between parameters inferred from experiments in different conditions. |
| Data exclusions | Force-distance data exhibiting more than one peak were excluded from total rupture forces distributions. The rationale behind the exclusion of these data is that multiple force-distance peaks likely indicate more than one cohesin molecule connecting the beads and DNA. As their single-nature could not be confirmed, accurate analysis was not possible. |
| Replication | The number of repeat measurements made is stated in the figure legends. All experiments were repeated at least three times (up to one hundred times), performed on different days and in independent flow cells. |
| Randomization | Single-molecule data with the same conditions for DNA substrates were inherently randomized and no other special randomization was implemented. |
| Blinding | Investigators were not blinded. All samples were prepared on the same day the measurement was performed and prepared by the same author collecting the data. |

# Reporting for specific materials, systems and methods

We require information from authors about some types of materials, experimental systems and methods used in many studies. Here, indicate whether each material, system or method listed is relevant to your study. If you are not sure if a list item applies to your research, read the appropriate section before selecting a response.

## Materials & experimental systems

| n/a | Involved in the study |
|---|---|
| ☐ | ☒ Antibodies |
| ☒ | ☐ Eukaryotic cell lines |
| ☒ | ☐ Palaeontology and archaeology |
| ☒ | ☐ Animals and other organisms |
| ☒ | ☐ Clinical data |
| ☒ | ☐ Dual use research of concern |
| ☒ | ☐ Plants |

## Methods

| n/a | Involved in the study |
|---|---|
| ☒ | ☐ ChIP-seq |
| ☒ | ☐ Flow cytometry |
| ☒ | ☐ MRI-based neuroimaging |

# Antibodies

| | |
|---|---|
| Antibodies used | For single-molecule experiments:<br>Antibody: Anti-Digoxigenin-AP, Fab fragments<br>Supplier: Roche<br>Catalogue number:  11093274910<br><br>For Western blotting:<br>Antibody: Mouse monoclonal anti-V5 tag (anti-Pk tag)<br>Supplier: Bio-Rad<br>Catalogue Number: MCA1360<br><br>Antibody: Mouse monoclonal anti-HA tag<br>Supplier: Sigma-Aldrich<br>Catalogue Number: 11583816001<br><br>Antibody: Anti-mouse IgG (HRP conjugated)<br>Supplier: GE Healthcare<br>Catalogue Number: NA931<br><br>Antibody dilutions used for Western blotting were 1:10000. Other dilutions are indicated in the Methods and Supplementary Methods of the Paper. |
| Validation | For the anti-Digoxigenin-AP, see supplier's website, containing relevant references of validation: https://www.sigmaaldrich.com/GB/en/product/roche/11093274910<br><br>For the mouse monoclonal anti-V5 antibody, see supplier's website, under "product specific references: https://www.bio-rad-antibodies.com/monoclonal/viral-v5-tag-antibody-sv5-pk1-mca1360.html?f=purified&JSESSIONID_STERLING=5864A39766958B0AB4D28A9E7712D453.ecommerce1&evCntryLang=UK-en&EU_COOKIE_PREFS=111&cntry=UK&thirdPartyCookieEnabled=true<br><br>For the mouse monoclonal anti-HA antibody, see supplier's website, containing relevant references of validation: https://www.sigmaaldrich.com/GB/en/product/roche/roaha |

