## [Peer Review File · Nature Structural & Molecular Biology]

Peer Review Information

Manuscript Title: Mechanical disengagement of the cohesin ring

Corresponding author name(s): Maxim Molodtsov, Frank Uhlmann

Reviewer Comments & Decisions:

Decision Letter, initial version:

Message: 19th Jan 2023

Dear Dr. Molodtsov,

Thank you again for submitting your manuscript "Mechanical disengagement of the cohesin rings". My sincere apologies for not writing back earlier with a decision. The delay was caused by not being able to find reviewers due to the end-of-the-year holidays and not receiving their reports until a few days ago. Nevertheless, we have now received comments from two expert referees who evaluated your paper (appended below). In light of those reports, I am afraid that we cannot offer to continue considering this manuscript for publication at Nature Structural & Molecular Biology.

You will see that though the reviewers appreciate the interesting biological question raised and biophysical approaches undertaken to address it, they voice important concerns on a mechanistic and technical level. More specifically, both reviewer #1 (R#1) and reviewer #2 (R#2) note that, under the presented conditions, it is mechanistically uncertain where and from which specific molecules rupture events originate. They also remain unpersuaded that the presented data adequately support the finding that ring opening occurs at the hinge interface. As pertinently, both reviewers raise serious technical concerns with respect to the setup used, which would require either extensive additional experiments, expanded sample sizes/tighter controls, and remedying underlying assumptions in R#1's case, or setting up and utilising a whole different experimental system in R#2's case.

However, if further experimentation, analyses, and revisions allow you to fully address the referees concerns, we would be prepared to consider an appeal of our decision, on the condition that no related work is published in the interim or has been accepted in our journal. Please note that, until we have the opportunity to read the revised manuscript in its entirety, we cannot promise that it will be sent back for peer review. If you decide to appeal and deem that you have acquired the necessary new data to address the reviewers' concerns, feel free to contact me to discuss a potential revision.

I am sorry we could not be more positive on this occasion. I hope that you find the referees' comments useful in deciding how best to proceed and strengthening this manuscript.

Sincerely,

Dimitris Typas
Associate Editor
Nature Structural & Molecular Biology
ORCID: 0000-0002-8737-1319

Reviewers' Comments:

Reviewer #1:

Remarks to the Author:

The paper by Richeldi et al. uses single-molecule optical tweezers combined with TIRF microscopy to determine the force required for open a single cohesin complex before it releases DNA. Richeldi et al. further suggests that the cohesin ring opening occurs at the hinge interface. Estimating the cohesion force is of high interest since it can correlate to the force that cohesin can withstand the pulling of the mitotic spindle. However, I find the experimental data do not sufficiently support the key claims of this paper, i.e. 1) the rupture force is obtained from the detachment of a single cohesin 2) the rupture force is from the opening of the trimeric cohesin ring that is topologically loaded on DNA 3) the rupture events occur at the hinge domain of the cohesin ring. In general, the sample sizes of the experiments are low and lack proper controls in some cases. Unless the authors provide a significant number of additional experiments and controls to support their claim, I am afraid that the manuscript is not suitable for publication in NSMB.

Major:

1. The authors claim the observed rupture forces are obtained from the detachment of single proteins. This is supported by the photobleaching experiments and the FD curves exhibiting a single peak. However, the authors state that only 30% of photobleaching events are single events, meaning most cohesins exist as oligomers on DNA. Although the authors show one example (Extended Data Fig. 2b), where the bead-bound cohesin bleaches out in a single step before FD measurement, it appears they have not done the FD measurement selectively for the cohesins with single-step bleaching. The distribution of disruption forces seems to originate from both monomers and oligomers.

Moreover, the authors do not show the 'detachment' of the cohesin complexes from DNA, i.e., the disappearance of the protein signals after the bead detachment. Therefore, it is not clear to me whether the rupture event comes from the separation of a single protein from DNA or the detachment between the proteins in the oligomers. The authors should compare the protein signals before and after the rupture event and check whether the protein signals disappear from DNA in one step for single cohesins and remain on the bead.

2. The authors claim that the rupture force is from an opening of a trimeric cohesin ring

that was topologically loaded on DNA. This is based on the assumption that high-salt washing (500mM NaCl) would leave cohesins topologically bound on the DNA in a high-salt resistant manner. However, Shaltiel et al. Science 2022 showed that ATP-dependent topological DNA loading could occur without the opening of the trimeric ring. How can the author ensure the DNA is not entrapped at another compartment, e.g., kleisin chambers, as seen in the case of condensin? Without the control with single-chain crosslinked cohesin not showing any salt-resistant binding, the basic assumption of topological loading through SMC–kleisin ring opening is not valid.

Along these lines, I find it surprising that the authors observed similar distributions of rupture forces when the cohesin was washed with low salt buffer (130mM NaCl) as compared to the high salt buffer (500mM NaCl; Extended Data Figures 2a), even though 130mM NaCl is certainly not enough to remove salt-dependent, non-topological binding (e.g. 10.1016/j.molcel.2020.04.026). Are the authors sure that 500 mM NaCl has enough ionic strength to remove the non-topological bindings? Can the authors make sure to remove all non-topological bindings by washing with higher salt e.g., 1-2M NaCl?

3.The authors claim that the rupture occurs at the hinge interface of cohesin. This is supported by comparing force distributions obtained from hinge/head pulled cohesins with MC simulations and (2) by the bimodal force distributions obtained when the hinge was cross-linked. However, I have significant concerns in both of the data and the conclusions drawn here.

Regarding the data in Fig.2a, are the authors sure that the values shown here (k_0 and δ) provide the best possible fit? The experimental data for head cohesin, for example, exhibits broader distribution compared to the simulation. The data for hinge cohesin has too few data to properly compare with the simulation. The authors should increase the sample size and provide goodness-of-fit values for each case. Also, can the authors provide the standard deviation in the force distribution? Is the difference between 18 pN and 24 pN values in Fig.2a reliable? One would imagine - with different bin widths and sample size; the values might be different.

I understand the author's logic that the opening of the cohesin ring should come from the weakest interface. However, this assumes that there is a significant difference in the force required for opening hinge vs kleisin/head, which is unknown. If the differences are not significant, how the force is applied at the molecular scale might determine which interface opens. Furthermore, without knowing k_0 for hinge opening and kleisin/head opening, the author cannot claim that the hinge should be the weakest point. Although hinge has a smaller contact area, it is also well-known that DNA passes through the opening of kleisin/head gate. Why do the authors discard the possibility that ring opening can occur at the head/kleisin interface?

Regarding the experiments in Fig.2b, the authors use hinge-crosslinked cohesin and measure the rupture force to validate their hypothesis. They observe half of events occur at higher forces in this case. The authors specifically pinpoint that the 70 pN is from the cross-linked hinge opening, while this can also be the opening of other interfaces with higher interaction strength. Furthermore, the force required to break a covalent bond is typically an order of magnitude (>1000 pN), higher than 70 pN. In addition, the number of events is too low in Fig.2b. The extracted parameters from the fit of the data with a bimodal distribution likely have large uncertainties.

Most importantly, to support the claim of exclusive hinge opening, the authors should use

kleisin/head cross-linked cohesin and measure the corresponding rupture force. If this leads to a value close to 20pN, this will pinpoint the opening to the hinge interface. Their modelling approach alone is not at all sufficient to support this claim.

4.Regarding the topological loading of the second DNA, where does the entry of the second DNA occurs? Do you observe the entrapment of the second DNA from the hinge crosslinked cohesin?

5.Can the authors measure k_0 (spontaneous release) values without applying force, at least for the non-crosslinked ones?

6.Line180-182. I don't follow the logic here. Even with the additional DNA, the stretching should reach >90% even at 1 pN, much lower than the 20 pN disruption force. I would imagine the DNA stretching to make a negligible difference. If the authors think the DNA stretching takes longer, wouldn't that increase the releasing force rather than reduce it?

7. Extended Figure2d – Here, the authors observe the FD curve go to zero after the release of one of the cohesins. Is this the majority of cases when the authors observe multiple peaks? The suggested interpretation would be only true when the distance between two cohesins is large enough so that a large fraction of double-tethered DNA was stuck to the bead and not experiencing pulling forces till one of the proteins is released. If the two cohesins were located in close proximity, the force should not reduce to zero. Secondly, for two cohesins in proximity, the disruption force should be twice larger than the disruption force for one cohesin.

8. Supplementary video 3 and 4. I see that the contour length of DNA just before the disruption is $\sim 8 \mu\text{m}$ in both cases. At the force of $\sim 20 \text{ pN}$, I expect the DNA to be fully attached and close to its contour length ($16 \mu\text{m}$). This indicates there might be multiple attachment points on the DNA around the bead.

Minor:

- Line408 - "dichlorometylsilane" instead of "dychlorometylsilane"
- Line 571- Did the author mean " $\tan \alpha$ "

Reviewer #2:

Remarks to the Author:

Review: Richeldi et al., "Mechanical disengagement of the cohesin ring", NSMB-A47027-T (2023)

Overview

One key role of cohesin complexes is to hold sister chromatids together until all chromosomes are aligned at the metaphase to anaphase transition, which requires outstanding mechanical properties of cohesion. Holding sister chromatids together is thought to involve the topological entrapment of two DNAs by one cohesin ring, which has been demonstrated using "CD" formation of circular minichromosomes before in yeast, but remains to be reconstituted in vitro and to be demonstrated properly in vivo using chromosomal DNAs. Richeldi et al now provide experiments that deal with the former.

The authors utilise previously-published protocols to bind cohesin tightly to DNA in an ATP-dependent manner and investigate using single molecule and optical tweezer experiments several important questions regarding the mechanical aspect of the cohesin-DNA interaction. How much force can a single cohesin complex withstand before opening and releasing the DNA? When releasing DNA upon mechanical pulling, which cohesin interface opens to release the DNA? Can a single cohesin complex hold two DNA molecules?

The main findings presented in this study show that: 1) A single cohesin complex can withstand about 20 pN before opening releasing DNA; b) A single cohesin complex can bind two DNA molecules at the same time (either ssDNA and dsDNA). Finally, the authors propose that mechanical disengagement of the DNA could play a role during cell division and interphase.

Most experiments are done well and the manuscript is concise and clear. My main concern is the lack of direct evidence of topological entrapment after the loading reaction. It would have been wonderful to see this sort of data on cohesin tested for topological entrapment through something akin the 6C assay developed in the Nasmyth group (<https://pubmed.ncbi.nlm.nih.gov/25414305/>). It would have been fairly easy to set this up for pombe cohesin and to repeat the 6C entrapment experiments before embarking on such a number of important single molecule experiments.

The 6C-crosslinks (and other C-C crosslinks) would also have provided a powerful way to create different FD curves since some of the interfaces/gates could have been covalently linked/closed. It is very difficult to be sure with the current data as presented (and previously published) what we are looking at: is the DNA inside the ring or not? Salt resistance I do not find very convincing. Unrelated to the above issue, there may be quality problems with the hinge-crosslinked version of the protein (see below) and it would have also been easy to include constructs that covalently close gates through fusion constructs, as has been shown by James Collier (<https://elifesciences.org/articles/80310>).

As a result, some of the data leaves me with the feeling that we are flying blind, measuring forces on something that could have been (much) better characterized.

The finding of two DNAs potentially being entrapped in the same ring is exciting and goes somewhat along previous data from Luis Aragon (<https://www.science.org/doi/10.1126/sciadv.aay6804>). I found some of the data on this very important topic could have been presented better, for example including traces that show all components of the system coming in one by one. This is something the authors might be able to fix without too much trouble.

My other major comment (6C assay) is difficult to rectify without significant additional experimentation. It would be good to see those performed, certainly if the certainty that publishing in NSMB is meant to convey is to be achieved. I do try to avoid suggesting additional experiments normally, but it seems important in this case and would have provided a number of additional controls for the FD curves.

Just to mention: we are not experts in single molecule methods or optical tweezer experiments. We are also not very familiar with the modelling procedures, other reviewers will need to provide critical input on these.

Comments

- 1) Line 57: "...which in turn provides binding interfaces" might have to be rephrased. The way I read it, Scc3 mediates the binding of Scc2-Scc4, Pds5 and Wapl to the cohesin ring. However, to my knowledge that's not the role of Scc3 and these factors are directly binding the trimer.
- 2) Line 78: See above comment about the loading reaction. How can we be sure this is loaded topologically? The 6C assay would have been extremely helpful.
- 3) Line 88: How do we know what ruptures? Do we need control experiments that show when the DNA detaches etc? Why not?
- 4) Line 93: If 30% of FD curves showed one complex loaded onto one DNA molecule and 40% of the FD curves showed multiple complexes loaded onto a DNA molecule, maybe for completeness mention how the remaining 30% of the FD curves look like?
- 5) Line 115: Indeed, a smooth ring would be expected to always rupture at the weakest interface but the experimentation I think does not fully demonstrate that. Closing interfaces one by one would have been stronger evidence.
- 6) Lines 120-131: The authors report a hinge crosslinking efficiency of 70% and a bimodal distribution of the rapture forces, with half cohesin complexes detaching at 20 pN and half at 70 pN (lines 121-124). They explain this bimodal distribution by saying that 20 pN force is required to break complexes that failed to crosslink (therefore breaking at the hinge) and 70 pN force is required to break hinge crosslinked complexes. But if the crosslinking efficiency is 70%, shouldn't 70% of the complexes dissociate at 70 pN and 30% at 20 pN, in order to align the data with this hypothetical explanation, rather than 50-50? Sorry if we misunderstood ...
- 7) Line 132: Covalent fusions or Spytag/catcher could have been used to interrogate other interfaces. Or cysteine/BMOE crosslinking.
- 8) Line 143: Why did only 16 out of 21 cases lose the DNA after DNA cleavage?
- 9) Line 160: Scc2/4 and Pds5/Wapl experiments produce very clear results, good. Again, interface closures/fusions would have been great additions here.
- 10) Line 204: What is the evidence that cohesin ring breakage is the basis for centromere breathing?
- 11) Line 208: "simple hinge closure". Do the authors observe events of hinge opening (unloading) and closing again (reloading) in their experiments?
- 12) Lines 215-218: Therefore, would RNA Pol II unload cohesin at CTCF sites? If so, why wouldn't the helicase (line 221)?
- 13) Lines: 223-224: "physical force is an alternative mechanism for disengagement of the cohesin ring". I find this sentence has little meaning phrased like this. "Physical force" is always an alternative because anything will come off if you pull hard enough. Maybe "physical force is a physiologically possible mechanism..." would be more appropriate.

However, if on average there are 200 cohesin complexes loaded at the centromere, each withstanding 20 pN, the spindle would have to apply around 4,000 pN force, which is way higher than the 700 mentioned by the authors (lines 195-6). Therefore, in this case "physical force is a physiologically possible mechanism" would not be true.

14) Discussion: The authors could expand. What questions does this study raise? For example, can the authors speculate further about and suggest approaches to investigate the relevance of cohesin mechanical disengagement *in vivo*? Additionally, at the end of the Introduction and the Discussion, the authors highlight the importance of their funding in relation to the physical barriers that cohesin complexes find during loop extrusion, such as polymerases and helicases. I think in the Discussion the authors could expand on this, adding possible experimental approaches to validate the importance of cohesin mechanical disengagement in these situations.

15) Figure 1a: Both the coverslip and the bead bind to biotin, Is that a problem?

16) Figure 1b: If you add Scc2-4 in the loading assay, I'd include Scc2-4 in your graphical representation, or (maybe better) I'd mention in the legend that Scc2-4 isn't depicted for simplicity. The text reports that Scc2-4 is the "cohesin loader", but in the graphical representation the loaded complex is shown with Scc3 only, which might be misleading.

17) Figure 1c: Why does trace 3 (green) go to background level before the photobleaching step? Other traces go to background level before the photobleaching also in the other graphs ...

18) Figure 2 left & right: The distributions look bimodal?

19) Figure 3a/b: As mentioned already, it would be really nice if more complete traces could be shown that show all of the components present and they appearance. This way we simply see coloured traces that are always the same. It is not terribly convincing this way, unfortunately.

20) Figure 4a: Move the text "streptavidin-coated bead" on top of the red area.

21) Lines 349, 350 and 358: Which buffers were used?

22) Lines 351 and 359: What equipment and parameters were used?

23) Line 371: As mentioned already, it is unclear to me what topological state the complex/DNA are in. 6C assay?

24) Line 411: Followed -> followed by

25) Suppl. Fig 1b: Why do Scc1 and Scc3 migrate at the same height in the gel even though they have very different MWs? What is the 2xSmc3-CLIP-biotin band?

26) Suppl. FigS4b: With the Spytag/catcher crosslink, there should only be one 300 kDa band. There are many bands. Also, what is the band below the Smc1/Smc3 bands for WT proteins on the left? The WB in S4d is also not clean for this sample. I think this might indicate some serious quality issues with the crosslinked sample. 70% efficiency is low for

Spytag/catcher, again highlighting that the choice of location and linker may not be as good as the one here for yeast: <https://elifesciences.org/articles/80310>.

Author Rebuttal to Initial comments

Remark to all reviewers

We would like to thank all reviewers for their critical reading of our manuscript. We were delighted by their positive assessment, recognising the innovative aspects of our work and considering our manuscript to be of “high interest”. The reviewers found that our study investigates an “important questions regarding the mechanical aspect of the cohesin-DNA interaction”.

The reviewers’ constructive suggestions helped us to significantly improve the quality of our manuscript. Before providing a point-by-point response to each of the points raised, we would like to briefly summarise our main improvements and additions:

1. We employed single-molecule visualisation in conjunction with force-distance curve measurement to confirm that the observed detachment events indeed stem from the rupture of single cohesin molecules.
2. We have designed a new biochemical system, which allowed us to covalently close the Smc3/head-kleisin interface. We employed this system because the 6C cohesin assay, which was suggested by both reviewers, has a typical crosslinking efficiency not high enough for our *in vitro* experiments and analyses. With our new assay, based on the SpyTag-SpyCatcher system, we show directly that a single cohesin molecule entraps DNA inside its ring.
3. This new system allowed us to covalently crosslink the Smc3^{Psm3}-kleisin interface after cohesin was topologically loaded onto DNA and to apply force onto individual Smc3^{Psm3}-kleisin-crosslinked cohesin molecules. Our new results confirmed our original conclusion that it is indeed the hinge that represents the mechanically weakest interface that disengages when cohesin is subjected to external force.
4. We have significantly improved and extended our statistical analyses, and we now show that all our conclusions are based on statistically significant results.
5. We have extended our modelling to include two interfaces in cohesin that can disengage. Comparison with experiments confirmed that all our results can be accounted for by a simplest model of the cohesin ring disengagement and allowed us to determine the stabilities of the hinge as well as Smc3^{Psm3}-kleisin interfaces from our experimental data.

We have revised and added additional panels to Figures 1-3 and to Extended Data Figures 1,2,4,6,8 as well as made essential corrections and extensions to the original main text of the manuscript (see marked changes in the revised manuscript). In addition, we addressed all the comments from all the reviewers, as detailed in our point-by-point response below.

REVIEWER #1

Remarks to the Author:

The paper by Richeldi et al. uses single-molecule optical tweezers combined with TIRF microscopy to determine the force required for open a single cohesin complex before it releases DNA. Richeldi et al. further suggests that the cohesin ring opening occurs at the hinge interface. Estimating the cohesin force is of high interest since it can correlate to the force that cohesin can withstand the pulling of the mitotic spindle. However, I find the experimental data do not sufficiently support the key claims of this paper, i.e. 1) the rupture force is obtained from the detachment of a single cohesin 2) the rupture force is from the opening of the trimeric cohesin ring that is topologically loaded on DNA 3) the rupture events occur at the hinge domain of the cohesin ring. In general, the sample sizes of the experiments are low and lack proper controls in some cases. Unless the authors provide a significant

number of additional experiments and controls to support their claim, I am afraid that the manuscript is not suitable for publication in NSMB.

We thank the reviewer for their comment that establishing the force that cohesin can resist is of “high interest”. As suggested by the reviewer, we performed substantial number of additional experiments to address the three main concerns: we confirmed that i) forces in our experiments are applied to single cohesin molecules, ii) single cohesin molecules in our assays topologically entrap DNA and the entrapment requires DNA to pass through the Smc3-kleisin gate, and iii) the mechanical rupture is due to disengagement at the hinge interface. We furthermore provided control experiments and statistical analyses that address the reviewer’s additional comments, as detailed in our point-by-point response below.

Major:

1. The authors claim the observed rupture forces are obtained from the detachment of single proteins. This is supported by the photobleaching experiments and the FD curves exhibiting a single peak. However, the authors state that only 30% of photobleaching events are single events, meaning most cohesins exist as oligomers on DNA. Although the authors show one example (Extended Data Fig. 2b), where the bead-bound cohesin bleaches out in a single step before FD measurement, it appears they have not done the FD measurement selectively for the cohesins with single-step bleaching. The distribution of disruption forces seems to originate from both monomers and oligomers.

We thank the reviewer for this comment and apologise for the confusion caused by the insufficient explanation of the 30% number stated in the original manuscript. In the revised manuscript, we now clearly show that 80% of the cohesin molecules loaded onto DNA photobleached in one single step, whereas the remaining 20% bleach instead in two steps, indicating a dimer of complexes (New Fig. 1d). Therefore, the majority of cohesins were loaded as single molecules. Some DNAs had more than one cohesin, and out of cases out of all the DNAs analysed, 55% of the DNAs had either one single cohesin loaded or two cohesins, which were separated spatially by a distance larger than the size of a bead. The original 30% number was referred to the fraction of all DNAs that had only a single cohesin molecule on it.

However, our data did not imply that only 30% of the beads were attached to single cohesins. This is because there is an additional parameter in the system, which is the probability of cohesin binding a bead. This parameter was low and on average only 10% of all DNA molecules had a bead attached to it via cohesin, suggesting that only a fraction of all single cohesin complexes were attached to a bead.

In the revised manuscript we have quantified directly the distribution of the number of cohesins per bead and show that the fraction of beads carrying only one cohesin matches the fraction of FD single-step rupture curves. We have calculated the distribution of the number cohesins per attached bead from the distribution of the number of cohesins per DNA and number of beads per DNA (New Extended Data Fig. 2a). This calculation showed that given the experimentally measured distributions of the number of cohesins per DNA and the number of beads per DNA, there is a ~70% probability that each bead on DNA is attached only via a single cohesin, whereas the rest of the beads (~30%) are attached via more than one cohesin (New Extended Data Fig. 2a).

In the revised manuscript we also collected more force-distance data and when all our data is taken together, we now show that 70% of the all rupture events exhibited single-peak ruptures and 30% of

all events showed multiple peaks (n=92). This data is in excellent agreement with the fraction of beads expected to be attached to a single cohesin and suggests that all single-rupture events correspond to the ruptures of single cohesin rings. We have made this clearer in the revised manuscript.

Moreover, the authors do not show the 'detachment' of the cohesin complexes from DNA, i.e., the disappearance of the protein signals after the bead detachment. Therefore, it is not clear to me whether the rupture event comes from the separation of a single protein from DNA or the detachment between the proteins in the oligomers. The authors should compare the protein signals before and after the rupture event and check whether the protein signals disappear from DNA in one step for single cohesins and remain on the bead.

We thank the reviewer for the suggestion, which we followed. In the revised manuscript, we performed additional experiments in which we visualised cohesin before and after the force was applied (New Fig. 2d). We observed that cohesin's fluorescent signal, which at first colocalised with the bead, was lost after the force application which resulted in a single-peak rupture FD curve. Instead, in the control experiment where a smaller amount of force was applied (5pN, not sufficient for rupture), the fluorescent signal corresponding to the cohesin molecule remained on DNA (New Fig. 2d). This confirms that the disappearance of the fluorescent signal is due to the detachment of the cohesin complex and not photobleaching. These experiments also additionally confirmed that the single-peak rupture is indeed a signature of a single cohesin being detached from DNA and provided additional confirmation that we applied forces and measured the detachment of single cohesin molecules.

Visualising the cohesin's fluorescent signal on the bead after its detachment from DNA, as the reviewer suggested, was not technically possible. When the bead is held in the optical trap, laser scattering masks completely weak fluorescent signal coming from the single cohesin molecule on the bead even at low intensities of the trapping laser because unlike the illumination lasers, the trapping laser is highly focused to a diffraction limited spot on the bead. Once the laser is off and the bead is released, it is free to rotate and diffuses away very quickly, as it is not held in place by the cohesin-DNA interaction anymore. The quick diffusion of the bead away from the coverslip prevents visualization of cohesin bound to its surface.

2. The authors claim that the rupture force is from an opening of a trimeric cohesin ring that was topologically loaded on DNA. This is based on the assumption that high-salt washing (500mM NaCl) would leave cohesins topologically bound on the DNA in a high-salt resistant manner. However, Shaltiel et al. Science 2022 showed that ATP-dependent topological DNA loading could occur without the opening of the trimeric ring. How can the author ensure the DNA is not entrapped at another compartment, e.g., kleisin chambers, as seen in the case of condensin? Without the control with single-chain crosslinked cohesin not showing any salt-resistant binding, the basic assumption of topological loading through SMC–kleisin ring opening is not valid.

We agree that the topological nature of cohesin loading onto DNA was not sufficiently documented in our original manuscript. Therefore, we performed multiple additional experiments, which altogether show that in our conditions the DNA is entrapped within the cohesin ring. Firstly, we cleaved at one site the tethered DNA using the restriction enzyme XhoI to determine whether cohesin would slip off the broken DNA end – a mark of topological loading – or would remain bound. We observe that after DNA cleavage, cohesin slides off the DNA, even under low applied flow (10 μ l/min) and at physiological salt conditions of 130 mM NaCl (New Supplementary Video 1). Of the 25 cohesin molecules tested, all 25 left the DNA after cleavage. Secondly, we show that topologically-loaded cohesin, once bound to the bead, can slide freely along the tethered λ -DNA when moved in

the x direction, but strongly resists detachment when pulled in the y direction (perpendicularly to DNA) consistent with topological entrapment (New Fig. 2a). Lastly, we show that 130 mM NaCl is sufficient to remove all non-salt-resistant cohesin in our assay (see our response to the next point below and New Fig. 1e). Thus, through these additional experiments, we verified that the interaction between the reconstituted cohesin and DNA in our conditions is topological in nature.

As the reviewer suggested in one of their points thereafter, a powerful way in which the topological outcome of cohesin-loader and ATP-hydrolysis-dependent cohesin loading was previously characterised involves the covalent circularisation of the cohesin complex, known as 6C cohesin (work by James Collier). While this approach was designed to investigate the products of a bulk cohesin loading reaction, unfortunately, 6C cohesin crosslinking cannot easily be transferred to our single molecule setup. 6C crosslinking efficiency, following completion of a cohesin loading reaction, is typically in the 10% range. DNA retention following protein denaturation is thus indicative of topological entrapment. Conversely, 90% cohesin loss following crosslinking in a flow cell, as compared to 100% loss, would not constitute a robust readout for the topological entrapment. More importantly, the required conditions for protein denaturation in SDS at 50°C are not achievable in our microfluidic setting.

The reviewer's alternative suggestion is to perform crosslinking before the loading reaction. While this is an interesting approach, it comes with the limitation that crosslinking biochemically inactivates cohesin (even wild type cohesin is inactivated by cysteine crosslinker treatment, likely due to modification of the numerous naturally occurring cysteines). 6C experiments therefore can only be performed following completion of a loading reaction.

Cohesin loading onto DNA under our conditions strictly depended on the cohesin loader and on ATP hydrolysis, hallmarks of the topological loading reaction. Also, as we describe in detail below, we have developed an approach alternative to 6C in which we have closed the Smc3-kleisin interface using a SpyCatcher-based protein crosslinker. Covalent closure of this interface after the topological loading almost completely prevented ATP-dependent unloading of cohesin consistent with unloading through opening of the kleisin N-gate (Higashi *et al.* 2020, Collier *et al.* 2022). Therefore, topological entrapment in our experiments required the DNA to pass the Smc3-kleisin interface, thereby confirming DNA entry into the cohesin ring in our setting (new Fig. 1g).

Along these lines, I find it surprising that the authors observed similar distributions of rupture forces when the cohesin was washed with low salt buffer (130mM NaCl) as compared to the high salt buffer (500mM NaCl; Extended Data Figures 2a), even though 130mM NaCl is certainly not enough to remove salt-dependent, non-topological binding (e.g. 10.1016/j.molcel.2020.04.026). Are the authors sure that 500 mM NaCl has enough ionic strength to remove the non-topological bindings? Can the authors make sure to remove all non-topological bindings by washing with higher salt e.g., 1-2M NaCl?

We have followed the reviewer's suggestion and tested how different salt concentrations affect the removal of non-topologically loaded cohesin. In the revised manuscript, we show that in our assay 130 mM NaCl is sufficient to remove all non-salt-resistant cohesin from DNA and increasing the salt concentration up to 2 M NaCl results in no further difference in the amount of cohesin that remains on DNA (New Fig. 1e). On the contrary, non-topological loading conditions (in the absence of ATP and the loader) result in almost complete removal of the protein complexes following 130 mM and higher NaCl washes (New Fig. 1e). These new results also explain why we did not observe any significant difference in the rupture forces between experiments in which cohesin was washed with 130mM versus 500 mM NaCl (original Extended Data Fig. 2a), as all non-topologically loaded

cohesins were removed in both conditions. For these reasons, in the revised manuscript we combined data from these two conditions and used them as a single dataset.

3. The authors claim that the rupture occurs at the hinge interface of cohesin. This is supported by comparing force distributions obtained from hinge/head pulled cohesins with MC simulations and (2) by the bimodal force distributions obtained when the hinge was cross-linked. However, I have significant concerns in both of the data and the conclusions drawn here.

Regarding the data in Fig.2a, are the authors sure that the values shown here (k_0 and δ) provide the best possible fit? The experimental data for head cohesin, for example, exhibits broader distribution compared to the simulation. The data for hinge cohesin has too few data to properly compare with the simulation. The authors should increase the sample size and provide goodness-of-fit values for each case. Also, can the authors provide the standard deviation in the force distribution? Is the difference between 18 pN and 24 pN values in Fig.2a reliable? One would imagine - with different bin widths and sample size; the values might be different.

We thank the reviewer for the comment and apologise for the confusion caused by the insufficiently well explained comparison between the theoretical and experimental data, which we corrected in the revised version. The reviewer is correct that our theoretical model predicts the distribution of rupture forces based on the two parameters of the model, k_0 – the rate of spontaneous interface dissociation without force and δ – mechanical displacement parameter which determines how force affects the disengagement.

To find the parameters that would best fit the experimental distributions, we used least squares approach. Briefly, we found k_0 and δ values that minimised the squared sum of distances between the experimental and theoretically predicted means and standard deviations (std) of the rupture force distributions. Since our model does not allow for analytical solution (i.e. there is no analytical relationship describing how the mean and std of the rupture force distributions depend on k_0 and δ and therefore are non-differentiable), we used a two-dimensional golden search algorithm to minimise the least squares distance and find the best fit. The solution to this optimization problem was unique because as we showed both mean and variance were monotonic functions of k_0 and δ (Extended Data Fig. 4e,f). Therefore, we could determine the global minimum of the least squares, which yielded the optimal parameters of the model.

This algorithm did not allow us to provide the confidence intervals for the estimated parameters in the original version. Therefore, in the revised manuscript, we determined the confidence intervals for the optimal parameter fits by Monte-Carlo sampling of the maximum likelihood function that minimised the distance between the experimental and theoretical means and variance. When we independently fit the head and hinge data with our model, we obtained the following parameters k_0 and δ and their confidence intervals:

Head data fit with 90% confidence intervals: $k_0 = 0.0027, (0.0023, 0.0036)$ $\delta = 1.23, (1.17, 1.29)$	Hinge data fit with 90% confidence intervals: $k_0 = 0.0025, (0.0016, 0.0034)$ $\delta = 1.61, (1.47, 1.71)$
--	---

These fits show that there is a significant difference between δ values, but no statistically significant difference in k_0 values between the two fits. Since the two data sets were fitted independently to the same model, these results strongly suggest that both datasets are best explained by the same value of k_0 , but different values of δ .

The mean and standard deviations of the experimental distributions as well as the values corresponding to the best fit were the following:

	Head		Hinge	
	Experiment	Theory	Experiment	Theory
Mean (pN)	24.3	24.7	18.7	18.4
STD (pN)	7.88	7.43	5.33	5.4

This shows that our model reproduces both the average and the width of the distributions in both cases as well as the increased variance for the head data in comparison with the hinge data. Thus, experimental distributions do not appear significantly boarder than the theoretical prediction.

In the revised version we also used bootstrap analysis and showed that the number of experimental samples obtained for the hinge and head data were sufficient to draw our conclusions. As shown in New Extended Data Fig. 4g, when the number of experimental trials increased, errors in determining parameters k_0 and δ decreased, as expected. Variances started to plateau after around about 10 measurements and the further increase in the sample size did not significantly improve the fits (New Extended Data Fig. 4h). Additionally, the p-value for testing the difference between δ values for the head and hinge constructs decreases, indicating that more experimental trials allow us to detect the difference in δ parameter between the head and the hinge constructs (New Extended Data Fig. 4i). At the same time, the p-value for testing the difference for k_0 does not change, further supporting the idea that the same parameter k_0 best explains both datasets for any size of the experimental sample. Similarly to the standard deviations, both p-values plateau after about 15 experimental measurements, demonstrating that our sample size was sufficient and that further increase in the number of experimental data points would not lead to a better goodness-of-fit.

I understand the author's logic that the opening of the cohesin ring should come from the weakest interface. However, this assumes that there is a significant difference in the force required for opening hinge vs kleisin/head, which is unknown. If the differences are not significant, how the force is applied at the molecular scale might determine which interface opens. Furthermore, without knowing k_0 for hinge opening and kleisin/head opening, the author cannot claim that the hinge should be the weakest point. Although hinge has a smaller contact area, it is also well-known that DNA passes through the opening of kleisin/head gate. Why do the authors discard the possibility that ring opening can occur at the head/kleisin interface?

We thank the reviewer for this comment. The observation that there is a significant difference in the force required for opening hinge versus the Smc3^{Psm3}-kleisin interface is not based on our assumption, but follows directly from our experimental data using the crosslinked hinge cohesin. In the revised manuscript, we have now performed additional experiments (New Fig. 3b) as well as simulations (New Extended Data Fig. 8) to test whether another interface might have a similar rupture force as the hinge interface. Furthermore, as the reviewer points out, how force is applied at the molecular scale might affect which interface opens. To control for this caveat, we made a like-for-like comparison of our hinge crosslinking data with the head cohesin dataset – in both cases the force is applied through the head domain with the exact same geometry.

Briefly, in the revised manuscript, we performed additional simulations that account for two interfaces that could break in a ring, described by two sets of constants: $k_0^{(1)}, \delta^{(1)}$ and $k_0^{(2)}, \delta^{(2)}$. We have tested all possible scenarios of rupture between two interfaces having weak or strong interactions and our new results show that if there is at least one weak interface in the molecule the histogram of rupture forces will always have only one peak at low forces (New Extended Data Fig. 8). Thus, our data showing high rupture forces of the hinge-crosslinked construct (Fig. 3d) rules out a

possibility that the Smc3^{Psm3}-kleisin interface may have rupture force similar to the non-crosslinked hinge interface.

Our experiments show that when both interfaces are allowed to disengage, the average rupture force is ~20 pN, which as reviewer points out can be explained by the molecule having either one weak and one strong interface, or both weak interfaces. However, when approximately half of the molecules had their hinge covalently closed, approximately half of the rupture forces had an average value of 70 pN. We show in the revised manuscript that the only model that can account for these data is that the other non-crosslinked interface (kleisin/hinge) must be significantly stronger and disengage at 70 pN (New Extended Data Fig. 8). Fitting our new model with two sets of parameters to the hinge-crosslinked data allowed us to determine the k_0 and δ parameters for both the hinge and kleisin/Smc3-head interfaces, confirming that the kleisin/Smc3-head is a much stronger interface (Extended Data Fig. 8). Thus, not only we do not “discard the possibility that ring opening can occur at the head/kleisin interface” but opening at the Smc3^{Psm3}-kleisin interface is an integral part of our model and is required to account for all the rupture force distributions that we observed in the study.

Finally, our model predicted that, if another interface was covalently closed, the histogram of the rupture forces should have only one peak at low forces and be indistinguishable from the histogram of the ruptures from non-crosslinked molecules (Extended Data Fig. 8). We have now tested this prediction in the revised manuscript and found an excellent match between the prediction and the data, further confirming both, the assumptions of our model, as well as our interpretation. We explained this more clearly in the revised version.

Regarding the experiments in Fig.2b, the authors use hinge-crosslinked cohesin and measure the rupture force to validate their hypothesis. They observe half of events occur at higher forces in this case. The authors specifically pinpoint that the 70 pN is from the cross-linked hinge opening, while this can also be the opening of other interfaces with higher interaction strength. Furthermore, the force required to break a covalent bond is typically an order of magnitude (>1000 pN), higher than 70 pN. In addition, the number of events is too low in Fig.2b. The extracted parameters from the fit of the data with a bimodal distribution likely have large uncertainties.

We apologise for the confusion, which was due to our poor explanation in the original version of the text. The reviewer is correct that once the hinge is covalently crosslinked it cannot be broken by the forces applied by optical tweezers, and our original interpretation was indeed that once the hinge is covalently closed it must be a different interface (e.g. Smc3^{Psm3}-kleisin) that breaks at 70 pN force. The crosslinking system used (SpyTag-SpyCatcher) forms a covalent bond, which would need forces above 1000 pN to be broken. As we also reasoned in our response to the point above, this allowed us to determine the parameters of the disengagement for both the hinge and the Smc3^{Psm3}-kleisin interfaces. We made this clearer in the revised version.

In the revised version, we fitted the hinge-crosslinked data from the original Fig. 3a (New Fig. 3d) with the updated model that had two interfaces that can break, and show that the two peaks can only be a result of the two molecular species of cohesin, one with covalently closed hinge and one without. Fitting the two peaks in the data to our model allowed us to determine the confidence intervals of the parameters $k_0^{(1)}$, $\delta^{(1)}$ and $k_0^{(2)}$, $\delta^{(2)}$ for both interfaces. As discussed in the previous point, these parameters determine the strength of the hinge and presumably Smc3^{Psm3}-kleisin interfaces, with the fits yielding the following values:

Fits with 90% confidence intervals:

$k_0^{(1)} = 0.0021, (0.001, 0.0034) \text{ s}^{-1}$ $\delta^{(1)} = 1.23, (1.1, 1.37) \text{ nm}$	$k_0^{(2)} = 2.7 \cdot 10^{-5}, (1.6 \cdot 10^{-5}, 3.9 \cdot 10^{-5}), \text{ s}^{-1}$ $\delta^{(2)} = 0.80, (0.77, 0.83), \text{ nm}$
---	--

The first set of parameters represents a weak interface and is statistically identical to the parameters inferred from the non-crosslinked (wild type cohesin) data. The second set of parameters corresponds to a much stronger interface, and it is only present in the dataset with the crosslinked hinge.

Therefore, our data indeed supports the reviewer's conclusion that "70 pN [...] can also be the opening of other interfaces with higher interaction strength". As our new Extended Data Fig. 8 shows, the two hinge-crosslink data can only be accounted for by the presence of two species of molecules in this experiment. One species must have only one strong interface. This species is not present in the non-crosslinked data and therefore represents molecules with a crosslinked hinge that rupture at the Smc3^{Psm3}-kleisin interface. The other species must have a weak interface present. This most likely represents molecules that failed to crosslink and have both interfaces that can rupture. However, the Smc3^{Psm3}-kleisin interface never has a chance to rupture in these molecules because the hinge ruptures at lower forces.

Most importantly, to support the claim of exclusive hinge opening, the authors should use kleisin/head cross-linked cohesin and measure the corresponding rupture force. If this leads to a value close to 20pN, this will pinpoint the opening to the hinge interface. Their modelling approach alone is not at all sufficient to support this claim.

We thank the reviewer for the suggestion, which we followed. We have measured rupture forces for cohesin complexes in which the kleisin/head interface was covalently crosslinked (*Smc3^{Psm3}-kleisin cohesin in our new manuscript*). As we explained above, it was not possible to use the suggested 6C model for this experiment. Therefore, we designed a new system in which the kleisin and Smc3-head subunits carried two SpyTags, that could be covalently closed by the addition of a distinct molecule consisting of two SpyCatcher proteins separated by a flexible and unstructured 50-nm long linker to crosslink the Smc3^{Psm3}-kleisin interface. With this system, the non-crosslinked complex with the two SpyTags loaded on DNA as efficiently as its wild-type construct and the Smc3-kleisin interface could be covalently closed after the molecule was topologically loaded on DNA. We verified that this was indeed the case and DNA could not anymore leave the complex via the Smc3-kleisin gate after the crosslinking (New Fig. 1g).

Then we applied forces to the topologically loaded cohesin with crosslinked Smc3-kleisin interface and found out that the distribution of the rupture forces in this case was indistinguishable from the wild type cohesin (New Fig. 3b). Since Smc3-kleisin interface could not rupture in this case, the ruptures must have been due to the disengagement at a different interface. Together with our hinge crosslinking data and new simulations and analysis presented in the revised version, this shows that the physically weakest interface within the cohesin complex – and the first to disengage under tension – is the hinge interface.

4.Regarding the topological loading of the second DNA, where does the entry of the second DNA occurs? Do you observe the entrapment of the second DNA from the hinge crosslinked cohesin?

We agree with the reviewer that understanding the entry points of the DNA into cohesin is an extremely interesting question. Our main finding is that two DNAs can be entrapped by a single cohesin ring, and that external mechanical force applied to a single cohesin disengages the cohesin ring and releases the DNA at forces in the order of 20 pN. Determining how the first and the second

DNA enter and exit the cohesin ring are fundamental next steps. What we can say is that, so far, we have failed to observe second DNA capture by cohesin with a covalently closed hinge. This negative result remains to be further explored - *e.g.* we know that second DNA capture strictly depends on the presence of the cohesin loader, and that the covalent hinge closures that have thus far been used in the literature, as well as by us, overlap with the hinge-cohesin loader interaction surface. To perform a conclusive analysis of the second DNA capture mechanism will therefore require several more experiments that go beyond the scope of our present work.

5.Can the authors measure k_0 (spontaneous release) values without applying force, at least for the non-crosslinked ones?

We thank the reviewer for this suggestion. In the revised manuscript, we clarified that the k_0 parameter in our model is the rate of the interface disengagement and not the rate of cohesin release from DNA. Thus, it cannot be visualised directly in our experiments. It is true that in our experiments these two parameters were coupled due to externally applied force – disengagement of the hinge always leads to the cohesin release under load because as soon as the hinge is disengaged DNA is physically removed from the cohesin ring by the algorithm that displaces the bead away from DNA. However, without force, cohesin may remain bound on DNA even with an open hinge. Thermodynamics also predicts that the hinge cannot remain open indefinitely and must close back again at some point in time, due to the significant affinity of the hinge domains and their close proximity in space even after dissociation. This makes the hinge reverse engagement very likely before the DNA has a chance to escape cohesin when there is no external force to give it direction.

Our model predicts that the rate of hinge disengagement without applying force is on the order of several minutes ($k_0 = 2 \cdot 10^{-3} \text{ s}^{-1}$). This prediction is supported by recent AFM data, which shows that the hinge may dissociate spontaneously on this timescale (Bauer *et al.*, Cell 2021 [video S1], Ryu *et al.*, NSMB 2020 [Fig. S2d], Kaur *et al.*, *bioRxiv* 2022). These AFM observations, which highlight the dynamicity and even spontaneous disengagement of the hinge domain, are in excellent agreement with our data that the hinge is the weakest point in the cohesin ring.

6.Line180-182. I don't follow the logic here. Even with the additional DNA, the stretching should reach >90% even at 1 pN, much lower than the 20 pN disruption force. I would imagine the DNA stretching to make a negligible difference. If the authors think the DNA stretching takes longer, wouldn't that increase the releasing force rather than reduce it?

We apologise for the confusion that stems from our poor explanation. We agree that the DNA spends most of the time at low forces and only a small fraction of time at the high forces that make difference for rupture. However, the longer the DNA, the proportionally more time it spends at all forces – *e.g.* a DNA which is 2 times longer will still spend most of its time stretching at low forces, as reviewer suggests, but it will also spend double the amount of time at any force, including high forces. In the revised manuscript New Extended Data Fig. 7f shows how force increases as a function of time as two DNAs are being stretched, one of which is twice the length of the other. As an example, we highlighted the times τ_1 and τ_2 that both molecules spend while the force is increasing from 10 to 15 pN. One can see that τ_2 is approximately two times larger than τ_1 , and therefore longer DNA spends more time while going from 10 to 15 pN. This gives cohesin more time to break at slightly lower forces before higher forces are reached. This explains why we see more ruptures at slightly lower forces ~ 15 pN for longer DNA (with the two DNA substrates). We have made this clearer in the revised manuscript.

7. Extended Figure2d – Here, the authors observe the FD curve go to zero after the release of one of the cohesins. Is this the majority of cases when the authors observe multiple peaks? The suggested

interpretation would be only true when the distance between two cohesins is large enough so that a large fraction of double-tethered DNA was stuck to the bead and not experiencing pulling forces till one of the proteins is released. If the two cohesins were located in close proximity, the force should not reduce to zero. Secondly, for two cohesins in proximity, the disruption force should be twice larger than the disruption force for one cohesin.

The reviewer is correct, and we observe both types of double rupture events: those in which the force drops back to zero between sequential ruptures (as in the original trace shown) and those in which the ruptures happen closer in time and the force does not go all the way to zero. In the revised manuscript we have included examples of both (New Extended Data Fig. 2c).

We also agree with the reviewer that, in theory, the force exerted by two closely positioned cohesins should be double of the force that can be exerted on a single complex. However, in practice, these events are expected to be vanishingly rare. Firstly, given the density of the streptavidin coating on the bead, two cohesin complexes that are very close to one another would have a low chance of both being able to bind the bead. Secondly, even if two cohesins bound to the bead, a force that is twice larger than for one single molecule would only be observed when the force exerted by the bead is balanced by equal forces exerted by the two cohesins. This would require perfect alignment with the optical trap positioned exactly between two cohesins, which can never be achieved in practice. Therefore, one cohesin will always carry most of the load due to this misalignment causing it to rupture first, which would then shift the load to the other cohesin. Somewhat similarly due to the geometry, two of the kinesin molecular motors can never generate the double amount of force compared to one kinesin when moving on microtubules (e.g. Jamison et al., 2010. Biophys J. 99:2967). Finally, the double amount of force could only be possibly observed if both cohesins resist forces for a sufficiently long time. However, rupture is a stochastic process which will result in one cohesin (likely experiencing a larger force) breaking earlier than the other. We made this clearer in the revised text.

8. Supplementary video 3 and 4. I see that the contour length of DNA just before the disruption is ~8 μm in both cases. At the force of ~20 pN, I expect the DNA to be fully stretched and close to its contour length (16 μm). This indicates there might be multiple attachment points on the DNA around the bead.

We thank the reviewer for spotting this error and apologise for the confusion caused by these movies. The reviewer is correct that DNA appears ~8 μm in length. This is because in a subset of preliminary experiments we used DNA approximately half the length of λ -DNA, resulting in the observed 8 μm length. Consistent with the above considerations of DNA lengths, forces measured using the shorter DNA were slightly higher and were not considered in the statistics in Fig. 2e. Inadvertently, this movie was included as an example. In the revised version of the manuscript we have replaced these movies by ones obtained with full length λ -DNA, showing what the stretching of 16 μm long DNA looks like.

Minor:

- Line 408 - “dichlorometylsilane” instead of “dychlorometylsilane”
- Line 571- Did the author mean “tan α ”

We thank the reviewer for pointing out these typos, which we have corrected.

REVIEWER #2

Remarks to the Author:

Review: Richeldi et al., "Mechanical disengagement of the cohesin ring", NSMB-A47027-T (2023)

Overview

One key role of cohesin complexes is to hold sister chromatids together until all chromosomes are aligned at the metaphase to anaphase transition, which requires outstanding mechanical properties of cohesin. Holding sister chromatids together is thought to involve the topological entrapment of two DNAs by one cohesin ring, which has been demonstrated using "CD" formation of circular minichromosomes before in yeast, but remains to be reconstituted in vitro and to be demonstrated properly in vivo using chromosomal DNAs. Richeldi et al now provide experiments that deal with the former.

The authors utilise previously-published protocols to bind cohesin tightly to DNA in an ATP-dependent manner and investigate using single molecule and optical tweezer experiments several important questions regarding the mechanical aspect of the cohesin-DNA interaction. How much force can a single cohesin complex withstand before opening and releasing the DNA? When releasing DNA upon mechanical pulling, which cohesin interface opens to release the DNA? Can a single cohesin complex hold two DNA molecules?

The main findings presented in this study show that: 1) A single cohesin complex can withstand about 20 pN before opening releasing DNA; b) A single cohesin complex can bind two DNA molecules at the same time (either ssDNA and dsDNA). Finally, the authors propose that mechanical disengagement of the DNA could play a role during cell division and interphase.

Major:

Most experiments are done well and the manuscript is concise and clear. My main concern is the lack of direct evidence of topological entrapment after the loading reaction. It would have been wonderful to see this sort of data on cohesin tested for topological entrapment through something akin the 6C assay developed in the Nasmyth group (<https://pubmed.ncbi.nlm.nih.gov/25414305/>). It would have been fairly easy to set this up for pombe cohesin and to repeat the 6C entrapment experiments before embarking on such a number of important single molecule experiments.

The 6C-crosslinks (and other C-C crosslinks) would also have provided a powerful way to create different FD curves since some of the interfaces/gates could have been covalently linked/closed. It is very difficult to be sure with the current data as presented (and previously published) what we are looking at: is the DNA inside the ring or not? Salt resistance I do not find very convincing.

We thank reviewer for these comments, and we agree that the topological nature of cohesin loading onto DNA was not sufficiently documented in our original manuscript.

The reviewer mentions 6C cohesin as powerful tool in which the topological outcome of cohesin-loader and ATP hydrolysis-dependent cohesin loading was previously characterised, involving covalent circularisation of the cohesin complex (e.g. Srinivasan *et al.* 2018 for budding yeast, or Kurokawa *et al.* 2020 for fission yeast). While this approach was designed to investigate the products of a bulk cohesin loading reaction, unfortunately, 6C cohesin crosslinking cannot easily be transferred to our single molecule setup. 6C crosslinking efficiency, following completion of a cohesin loading reaction, is typically in the 10% range. DNA retention following protein denaturation is thus indicative of topological entrapment. Conversely, 90% cohesin loss following crosslinking in a flow cell, as compared to 100% loss, would not constitute a robust readout for topological

entrapment. More importantly, the required conditions for protein denaturation in SDS at 50°C are not achievable in our microfluidic setting.

Although it was not possible in our setting to test topological loading using 6C cohesin, we designed a new system in which the kleisin and Smc3-head subunits carried two SpyTags, that could be covalently closed by the addition of a distinct molecule consisting of two SpyCatcher proteins separated by a flexible and unstructured 50-nm long linker to crosslink the Smc3^{Psm3}-kleisin interface. With this system, the non-crosslinked complex with the two SpyTags loaded on DNA as efficiently as its wild-type construct and the Smc3-kleisin interface could be covalently closed after the molecule was topologically loaded on DNA (New Fig. 1f). We verified that this was indeed the case and DNA could not anymore leave the complex via the Smc3-kleisin gate after the crosslinking (New Fig. 1g), consistent with unloading through opening of the kleisin N-gate (Higashi *et al.* 2020, Collier *et al.* 2022), and supporting the interpretation that DNA indeed enters the cohesin ring in our cohesin loading reactions.

Furthermore, cohesin loading onto DNA under our conditions strictly depended on the cohesin loader and on ATP hydrolysis, hallmarks of the topological loading reaction (New Fig. 1e). In the revised manuscript we also performed two additional experiments to confirm the topological loading of cohesin on DNA. First, we cleaved tethered DNA using the restriction enzyme XhoI to determine whether cohesin would slip off the broken DNA end – a mark of topological loading – or would remain bound. We observe that after DNA cleavage, cohesin slides off the DNA, even under low applied flow (10 μ l/min) and at physiological salt conditions of 130 mM NaCl (New Supplementary Video 1). Of the 25 cohesin molecules tested, all 25 left the DNA after cleavage. Secondly, we show that topologically loaded cohesin, once bound to the bead, can slide freely along the tethered λ -DNA, but strongly resists detachment when pulled perpendicularly to DNA consistent with topological entrapment (New Fig. 2a).

These two experiments with the confirmation that the topological loading in our case required DNA to pass the Smc3-kleisin gate (New Fig. 1g) confirmed that we applied force to cohesin molecules in which DNA was topologically entrapped inside the main cohesin ring. We have made this clearer in the revised manuscript.

Unrelated to the above issue, there may be quality problems with the hinge-crosslinked version of the protein (see below) and it would have also been easy to include constructs that covalently close gates through fusion constructs, as has been shown by James Collier (<https://elifesciences.org/articles/80310>). As a result, some of the data leaves me with the feeling that we are flying blind, measuring forces on something that could have been (much) better characterized.

We thank the reviewer for the comment and apologise that some of our presentation was unclear in the original manuscript. With regards to the reviewer's concerns of the quality of the crosslinked hinge, we would like to note that the amount of degradation products in our preparation is small, comparable to what was previously observed both with the SpyTag-SpyCatcher system (Fig. 1b, c in Collier & Nasmyth, 2022) or with cohesin used for cysteine crosslinking (Fig. 1b in Collier *et al.*, 2020). Moreover, degraded fragments of SMC complexes miss essential parts of their ATPase and would not be able to load onto DNA. Their presence therefore should not affect the results from the force measurements or their interpretation.

In the revised manuscript, to better characterise the forces of the opening interface within the cohesin ring, we have both performed additional experiments as well as developed additional simulations that supported our original conclusions and ruled out other possible interpretations.

Firstly, we measured rupture forces for cohesin complexes in which the kleisin/head interface was covalently crosslinked after loading on DNA (*Smc3^{Psm3}-kleisin cohesin in our new manuscript*). As we explained above, it was not possible to use the suggested 6C model for this experiment. Therefore, we designed a new system in which the kleisin and Smc3-head subunits carried two SpyTags, that could be covalently closed by the addition of a distinct molecule consisting of two SpyCatcher proteins separated by a flexible and unstructured 50-nm long linker to crosslink the Smc3^{Psm3}-kleisin interface after it was loaded on DNA. As we also already described in our response to one of the points above, closing the Smc3^{Psm3}-kleisin interface in this manner after DNA loading abolished spontaneous release of cohesin from the DNA through the Smc3-kleisin gate almost completely (New Fig. 1g).

Then we applied forces to the topologically loaded cohesin with crosslinked Smc3-kleisin interface and found that the distribution of the rupture forces in this case was indistinguishable from the wild type cohesin (New Fig. 3b). Since Smc3-kleisin interface could not rupture in this case, the ruptures must have been due to the disengagement interface.

In order to interpret these experiments together with the experiments in which the hinge was crosslinked, in the revised manuscript, we extended our simulations to account for the two interfaces that could break in cohesin ring, described by two sets of constants: $k_0^{(1)}$, $\delta^{(1)}$ and $k_0^{(2)}$, $\delta^{(2)}$. We have tested all possible scenarios of ruptures between two interfaces having weak (~20 pN) or strong (~70 pN) interactions (New Extended Data Fig. 8). These simulations show that the only model that can explain our data is that the hinge interface is weak (ruptures at ~20 pN force) and the Smc3^{Psm3}-kleisin interface is strong (ruptures at ~70 pN force) and that the topologically entrapped DNA escapes from the cohesin ring after the weakest interface breaks due to the applied force. We have made this clearer in the revised manuscript.

The finding of two DNAs potentially being entrapped in the same ring is exciting and goes somewhat along previous data from Luis Aragon (<https://www.science.org/doi/10.1126/sciadv.aay6804>). I found some of the data on this very important topic could have been presented better, for example including traces that show all components of the system coming in one by one. This is something the authors might be able to fix without too much trouble.

We thank the reviewer for this suggestion. We agree that it would be visually preferable to present real-time second DNA capture events, which however is experimentally challenging. Firstly, the reaction only occurs when incubating a large excess of the second DNA plasmid compared to the cohesin loaded onto DNA. For this reason, although the second DNA plasmid bears a fluorophore that is excited by a different laser wavelength than cohesin, the fluorescent oversaturation of the channels makes visualisation of individual cohesins and second DNA capture events during this incubation time impossible (even the best single fluorophore gives overall a relatively weak signal). Therefore, we can only observe second DNA capture events after having washed out any excess second DNA plasmid once the reaction occurred. Secondly, the capture of the second DNA by single cohesin complex is a rare event, we are often not able to observe the second DNA capture events in the same field of view. Instead, once unbound second DNA is washed away, we need to move around to find successful capture events. In our microscope one field of view (around 15 tethered DNAs) can be sampled during real-time observations. Despite these limitations, the cohesin and the second DNA fluorescent signals are spectrally completely separated. The unison movement of the two signals along the λ -DNA could only be the result of second DNA capture by cohesin.

We would also like to note that, unlike in the previous study by Gutierrez-Escribano *et al.*, 2019, our experiments visualise both single cohesin molecules and DNA. Capture of the DNA by single cohesins

in inefficient, but ability to visualize single cohesins and many DNA molecules some of which do not have any cohesin at all allows us to observe the behaviour and the properties of single cohesin – DNA interaction. The observation by Gutierrez-Escribano, that two DNAs incubated with cohesin can no longer be disjoined even by very high forces, suggests that likely unphysiological aggregates of multiple cohesins were responsible for the DNA-DNA contacts (an interpretation that those authors have confirmed in recent conference presentations).

My other major comment (6C assay) is difficult to rectify without significant additional experimentation. It would be good to see those performed, certainly if the certainty that publishing in NSMB is meant to convey is to be achieved. I do try to avoid suggesting additional experiments normally, but it seems important in this case and would have provided a number of additional controls for the FD curves.

As outlined above, a 6C assay is unfortunately not possible in our flow cell format. A 6C assay with fission yeast cohesin, however, was recently reported by Kurokawa *et al.* 2020, which has confirmed that cohesin loader- and ATP hydrolysis-dependent cohesin loading results in topological DNA interaction. Additionally, we have performed a series of experiments that have confirmed the topological nature of cohesin loading in our assay as we described above. These include measurements of resistance to movement along DNA that can only be explained by topological embrace and enzymatic cleavage of the tethered DNA. Furthermore, we have included an experiment in which we have covalently closed the Smc3^{Psm3}-kleisin interface after the topological loading of cohesin on DNA, which resulted in molecules that cannot unload through the Smc3^{Psm3}-kleisin gate. These experiments confirmed that cohesin complexes in our conditions were indeed topologically entrapping DNA and that this topological entrapment required DNA to pass through the Smc3^{Psm3}-kleisin interface. This shows that the topological entrapment occurs within the main cohesin ring. We have made this clearer in the revised manuscript.

Just to mention: we are not experts in single molecule methods or optical tweezer experiments. We are also not very familiar with the modelling procedures, other reviewers will need to provide critical input on these.

We thank the reviewer for this comment. Indeed, the interdisciplinary nature of this manuscript presents a special difficulty in clearly describing all the data and their interpretation from both a biological and a physical perspectives. However, we believe that the reviewer's comments helped us to significantly improve the quality of the manuscript and that it is now more interesting and accessible for a diverse audience of readers.

Minor:

1) Line 57: "...which in turn provides binding interfaces" might have to be rephrased. The way I read it, Scc3 mediates the binding of Scc2-Scc4, Pds5 and Wapl to the cohesin ring. However, to my knowledge that's not the role of Scc3 and these factors are directly binding the trimer.

We thank the reviewer for pointing out this issue, which we now corrected in the revised manuscript.

2) Line 78: See above comment about the loading reaction. How can we be sure this is loaded topologically? The 6C assay would have been extremely helpful.

Please see above our response to the major comments section for how we have confirmed the topological nature of our cohesin loading reaction.

3) Line 88: How do we know what ruptures? Do we need control experiments that show when the DNA detaches etc? Why not?

We thank the reviewer for the suggestion, which we followed. In the revised manuscript, we performed additional experiments in which we visualised cohesin before and after the force was applied (New Fig. 2d, left). We observe that cohesin's fluorescent signal, which at first colocalises with the bead, is lost after force application which results in a single-peak FD curve. Instead, in the control experiment where a smaller amount of force is applied (not sufficient for rupture), the fluorescent signal corresponding to the cohesin molecule remains on DNA (New Fig. 2d, right). This confirms that the disappearance of the fluorescent signal is due to the detachment of the complex and not photobleaching. These experiments also additionally confirmed that the single-peak rupture is indeed a signature of a single cohesin being detached from DNA and provided additional confirmation that we applied forces and measured the detachment of single cohesin molecules.

We also performed additional controls in which biotinylated cohesin was replaced with a biotin directly covalently attached to DNA. In the latter case, we did not observe ruptures at forces up to 80 pN (Extended Data Fig. 2d). These experiments confirmed that our data indeed shows ruptures of cohesin molecules.

4) Line 93: If 30% of FD curves showed one complex loaded onto one DNA molecule and 40% of the FD curves showed multiple complexes loaded onto a DNA molecule, maybe for completeness mention how the remaining 30% of the FD curves look like?

We thank the reviewer for this comment and apologise for the confusion caused by the insufficient explanation of the 30% number stated in the original manuscript.

The original 30% number was referred to the fraction of all DNAs that had only a single cohesin molecule on it, but not the fraction of beads attached via single cohesin.

In the revised manuscript we quantified directly the distribution of the number of cohesins per bead and show that the fraction of beads carrying only one cohesin matches the fraction of FD single-step rupture curves, which was $\sim 70\%$. We have calculated the distribution of the number cohesins per attached bead from the distribution of the number of cohesins per DNA and number of beads per DNA (New Extended Data Fig. 2a). This calculation showed that given the experimentally measured distributions of the number of cohesins per DNA and the number of beads per DNA, there is a $\sim 70\%$ probability that each bead on DNA is attached only via a single cohesin, whereas the rest of the beads ($\sim 30\%$) are attached via more than one cohesin (New Extended Data Fig. 2a).

This data is in excellent agreement with experimentally observed $\sim 70\%$ fraction of single-step ruptures and 30% of multiple step ruptures and suggests that all single-rupture events correspond to ruptures of single cohesin rings. We have made this clearer in the revised manuscript.

5) Line 115: Indeed, a smooth ring would be expected to always rupture at the weakest interface but the experimentation I think does not fully demonstrate that. Closing interfaces one by one would have been stronger evidence.

We thank the reviewer for the suggestion, which we followed. As we already discussed in our response to the major point above, in the revised manuscript we recorded rupture forces for both hinge-crosslinked as well as Smc3^{Psm3}-kleisin-crosslinked cohesins (New Fig. 3b,d). Our force application experiments show that high forces (~ 70 pN) are only observed for the hinge-crosslinked

cohesin, whereas disengagement events at high forces are never seen with the Smc3^{Psm3}-kleisin-crosslinked cohesin, and the rupture force distribution for the Smc3^{Psm3}-kleisin-crosslinked construct is indistinguishable from the wild type cohesin (New Fig. 3b). These new results further support our original conclusion that hinge is the weakest interface that breaks under mechanical load.

6) Lines 120-131: The authors report a hinge crosslinking efficiency of 70% and a bimodal distribution of the rupture forces, with half cohesin complexes detaching at 20 pN and half at 70 pN (lines 121-124). They explain this bimodal distribution by saying that 20 pN force is required to break complexes that failed to crosslink (therefore breaking at the hinge) and 70 pN force is required to break hinge crosslinked complexes. But if the crosslinking efficiency is 70%, shouldn't 70% of the complexes dissociate at 70 pN and 30% at 20 pN, in order to align the data with this hypothetical explanation, rather than 50-50?

We apologise for the confusion, due to our poor explanation in text. The reviewer is correct that if everything else is equal, 70% crosslinking efficiency should result in 70% of the molecules showing high rupture force, which is 20% more than in our experiments. However, this analysis assumes equal loading efficiency for the crosslinked and non-crosslinked molecules on DNA. Our data indicates that this is not the case – the hinge-crosslinked cohesin is less effective in loading onto DNA when compared to the “wild-type” complex (the same observation was made by Collier *et al.* 2022 using budding yeast cohesin harbouring hinge insertions at a corresponding position). More specifically, we observe that we need a concentration of 1.5 nM for the hinge-crosslinked complex to achieve the same amount of complexes loaded onto DNA as compared with the “wild-type” cohesin, which we load at 0.5 nM. In the mixture of 70% crosslinked and 30% non-crosslinked molecules, the latter will have a higher efficiency of loading which results in approximately equal amounts of crosslinked versus non-crosslinked molecules in DNA. This is in good agreement with what we observe experimentally. We have explained this more clearly in the revised manuscript.

7) Line 132: Covalent fusions or Spytag/catcher could have been used to interrogate other interfaces. Or cysteine/BMOE crosslinking.

We thank the reviewer for the suggestion, which we followed. In response to the major point above, we have already discussed our new system to interrogate the rupture of cohesin with a crosslinked kleisin/Smc3-head interface.

8) Line 143: Why did only 16 out of 21 cases lose the DNA after DNA cleavage?

We thank the reviewer for the opportunity to clarify this point. There are two main reasons why the second DNA dissociation efficiency was not 100%. Firstly, the efficiency of the restriction enzyme cleavage was likely not 100%. Importantly, the restriction enzyme was added to the buffer used for the second DNA capture reaction, likely decreasing its efficiency in cutting.

Secondly, in order to dissociate after enzyme digestion, the second DNA must diffuse out of the cohesin ring. Since the second DNA is 5 times larger than cohesin and diffusion at these salt concentrations is typically slow, even some of the successfully cleaved DNA molecules might have remained bound to cohesin due to steric hindrance of their diffusion outside of the cohesin ring. We made this clearer in the revised manuscript.

9) Line 160: Scc2/4 and Pds5/Wapl experiments produce very clear results, good. Again, interface closures/fusions would have been great additions here.

We thank the reviewer for the suggestion. The experiments referred to by the reviewer were performed to control for the activity of our cohesin complex after second DNA capture and demonstrated that indeed it could be removed by Pds5/Wapl as expected. As explained in detail above, we were able to covalently crosslink the Smc3^{Psm3}-kleisin interface whilst ensuring correct loading of the cohesin ring onto DNA.

10) Line 204: What is the evidence that cohesin ring breakage is the basis for centromere breathing?

We thank the reviewer for the opportunity to clarify this. During centromere breathing, cohesin is lost from centromeric regions presumably as a result of the physical force from spindle attachment. This physical force might have directly led to cohesin loss, or might have triggered a series of biochemical events that indirectly lead to cohesin unloading. In our revised manuscript, we make it clear that cohesin loss as the direct consequence of the physical force experienced from spindle attachment is only one possible explanation.

11) Line 208: "simple hinge closure". Do the authors observe events of hinge opening (unloading) and closing again (reloading) in their experiments?

In our experiments, as the hinge is disengaged, the continued movement of the stage physically displaces the bead with cohesin away from the DNA preventing it from any further possible interaction. Thus, in our experiments, ring disengagement is coupled to permanent cohesin removal. However, according to thermodynamics, we expect that once the cohesin hinge is disengaged it must be able to reversibly close again. While we could not see these events due to the nature of our single-molecule experimental setup (cohesin was dragged away before it had a chance to re-engage), some recent AFM data support the idea that the hinge make transiently open and then close again (Bauer et al., Cell 2021 [video S1], Ryu et al., NSMB 2020 [Fig. S2d], Kaur et al., bioRxiv 2022). These observations are consistent with our data as well as with the idea that hinge might be the weakest mechanical interface.

12) Lines 215-218: Therefore, would RNA Pol II unload cohesin at CTCF sites? If so, why wouldn't the helicase (line 221)?

We thank the reviewer for pointing out this logical inconsistency. What we meant to say is that both RNA Pol II and helicases can generate a sufficiently large mechanical force which could mechanically disengage the cohesin hinge interface. Whether this would lead to the cohesin removal or allow for the passage of RNA Pol II or helicase is as yet unknown. Exploring these questions will be exciting avenues for future experiments that our current study has opened up.

13) Lines: 223-224: "physical force is an alternative mechanism for disengagement of the cohesin ring". I find this sentence has little meaning phrased like this. "Physical force" is always an alternative because anything will come off if you pull hard enough. Maybe "physical force is a physiologically possible mechanism..." would be more appropriate. However, if on average there are 200 cohesin complexes loaded at the centromere, each withstanding 20 pN, the spindle would have to apply around 4,000 pN force, which is way higher than the 700 mentioned by the authors (lines 195-6). Therefore, in this case "physical force is a physiologically possible mechanism" would not be true.

We thank the reviewer for the excellent suggestion. Indeed "physiologically possible mechanism" is much more appropriate and closer in meaning to what we wanted to say. Given the geometry, it is unlikely that all 200 cohesin complexes are positioned so perfectly that all force would be equally distributed among them to balance the spindle force. It is much more likely that some cohesins,

possibly located close to kinetochores, will bear most of the load while others would experience much less force. We have changed the revised manuscript accordingly.

14) Discussion: The authors could expand. What questions does this study raise? For example, can the authors speculate further about and suggest approaches to investigate the relevance of cohesin mechanical disengagement in vivo?

Additionally, at the end of the Introduction and the Discussion, the authors highlight the importance of their finding in relation to the physical barriers that cohesin complexes find during loop extrusion, such as polymerases and helicases. I think in the Discussion the authors could expand on this, adding possible experimental approaches to validate the importance of cohesin mechanical disengagement in these situations.

We thank the reviewer for the excellent suggestion. In the revised manuscript we have discussed points raised by reviewer in more detail. In particular, it will be interesting to design cohesin variants in which interfaces can be conditionally closed/strengthened *in vivo*. Such variants might be built on the SpyCatcher-SpyTag system developed during our revisions. This will allow study of the consequences of altering mechanical cohesin responses during chromosome transactions, transcription, replication and chromosome biorientation.

15) Figure 1a: Both the coverslip and the bead bind to biotin, Is that a problem?

We apologise this was not clear enough in the original manuscript. We employ the biotin-avidin system for both DNA attachment to the coverslip as well as for the cohesin attachment to the bead. We chose this as the biotin-avidin interaction provides the strongest non-covalent binding system available, ensuring that detachment events represented cohesin disengagement and not breakage of the DNA ends or the cohesin-bead interaction. This was not a problem thanks to the additional biotin step in our washes. Once the biotinylated DNA was attached to the surface, we added copious amounts of free biotin molecules (5 mM) to block the free streptavidin on the surface of the coverslip. Free biotin was removed at the next stage before biotinylated cohesin was added to the flow cell and allowed to bind to DNA. After cohesin was incubated with DNA, all free biotin and biotinylated cohesins were removed from the flow cell and streptavidin coated beads were introduced. This allowed the beads to bind to cohesin without any interference from the surface. We have explained this process more clearly in the revised manuscript in the materials and methods section.

16) Figure 1b: If you add Scc2-4 in the loading assay, I'd include Scc2-4 in your graphical representation, or (maybe better) I'd mention in the legend that Scc2-4 isn't depicted for simplicity. The text reports that Scc2-4 is the "cohesin loader", but in the graphical representation the loaded complex is shown with Scc3 only, which might be misleading.

We thank the reviewer for this suggestion. We have amended our figure legend to explain that the loading was done in the presence of Scc2-4, which is not shown on the figure because it was later removed.

17) Figure 1c: Why does trace 3 (green) go to background level before the photobleaching step? Other traces go to background level before the photobleaching also in the other graphs ...

Bleaching is a stochastic process which occurs sooner for some molecules compared to others. Fig. 1c illustrates traces from three independent cohesin molecules and it shows that their intensity was the same before they photobleached, indicative of there being only one fluorophore on each complex. In Figure 1c, the green trace represents a molecule that happened to bleach first, and the

other two traces show molecules where the fluorophore stayed on longer before then bleaching at around the same time. This is the result of the natural stochasticity in the lifetime of fluorophores. We have changed the visual representation to make this clearer.

18) Figure 2 left & right: The distributions look bimodal?

We used the one-sample Kolmogorov-Smirnov test to check whether the distributions were indeed unimodal. This tests the null hypothesis that the sample data comes from a specified standard normal distribution (unimodal) or does not come from such distribution at the 5% significance level. Both head cohesin (p-value = 0.98) and hinge cohesin (p-value = 0.99) were described by the test as unimodal. Taking the two peaks from the hinge-crosslinked data as two separate samples, both also resulted unimodal (peak 1 = 0.14 and peak 2 = 0.49).

We also performed the two-sided Kolmogorov-Smirnov test to check the null hypothesis that the two sample datasets (the two peaks from the hinge crosslinked data) come from the same distribution or not. With a p-value of 1.4×10^{-6} the null hypothesis is rejected showing that the data comes from different distributions, and can therefore be classified as bimodal.

In the revised manuscript we now also show that means and variances of these distributions match the means and variances of the distributions predicted by our model of cohesin as a ring with an interface that can disengage due to applied force. We have now provided additional statistical tests and confidence intervals for all parameters determined from experimental data.

19) Figure 3a/b: As mentioned already, it would be really nice if more complete traces could be shown that show all of the components present and they appearance. This way we simply see coloured traces that are always the same. It is not terribly convincing this way, unfortunately.

We thank the reviewer for this suggestion. As we explained in our response to the major point above this was not technically possible because reactions of single molecules are rare and inefficient. As we showed above, we could only observe a handful of cohesin molecules in the field of view and the probability of any of them forming interaction with the second DNA was low. Therefore, we had to move to different fields of view to find single cohesins that established the interaction with the second DNA during the incubation period and after the DNA was washed out. We also could not increase the concentration of cohesin to see more events because in that case we would not be able to verify that the established interaction was indeed with a single cohesin molecule. Therefore, we had to rely on few cohesins to establish the interaction with DNA, which we could then find and study.

20) Figure 4a: Move the text "streptavidin-coated bead" on top of the red area.

We thank the reviewer for this suggestion. We have removed the red area completely to make the figure clearer.

21) Lines 349, 350 and 358: Which buffers were used?

We have expanded our Methods section to include the details of the buffers used.

22) Lines 351 and 359: What equipment and parameters were used?

We have expanded our Methods section to include the details of the equipment used.

23) Line 371: As mentioned already, it is unclear to me what topological state the complex/DNA are in. 6C assay?

We agree that the topological nature of cohesin loading onto DNA was not sufficiently documented in our original manuscript. As we explained above, we have added multiple additional experiments, which confirmed that interaction between cohesin and DNA was indeed topological and that it required that DNA to pass the Smc3-kleisin gate suggesting that in our force experiments DNA was inside the main cohesin ring for single cohesin molecules.

24) Line 411: Followed -> followed by

We thank reviewer for spotting this typo, which we corrected.

25) Suppl. Fig 1b: Why do Scc1 and Scc3 migrate at the same height in the gel even though they have very different MWs? What is the 2xSmc3-CLIP-biotin band?

We apologise for the confusion. Scc1 and Scc3 have very similar molecular weights in fission yeast and have been thoroughly characterised. Our results are consistent with several previous analyses of fission yeast cohesin complexes (Birkenbihl & Subramani, 1992, Tomonaga et al., 2000, Murayama & Uhlmann 2014). The additional 2xSmc3-CLIP-biotin band is the result of two separate Smc3-CLIP-biotin molecules binding to one single avidin molecule (detected in the gel). Indeed, avidin bears 4 binding sites for biotin and could therefore bind up to four Smc3-CLIP-biotin molecules. We have added these clarifications to the corresponding figure legend.

26) Suppl. FigS4b: With the Spytag/catcher crosslink, there should only be one 300 kDa band. There are many bands. Also, what is the band below the Smc1/Smc3 bands for WT proteins on the left? The WB in S4d is also not clean for this sample. I think this might indicate some serious quality issues with the crosslinked sample. 70% efficiency is low for Spytag/catcher, again highlighting that the choice of location and linker may not be as good as the one here for yeast: <https://elifesciences.org/articles/80310>.

We thank the reviewer for the comment and apologise that some of our presentation was unclear in the original manuscript. With regards to the reviewer's concerns of the quality of the crosslinked hinge, the amount of degradation products in our preparation is small, comparable to what was previously observed both with the SpyTag-SpyCatcher system (Fig. 1b, c in Collier & Nasmyth, 2022) or with cohesin used for cysteine crosslinking (Fig. 1b in Collier et al., 2020). Moreover, degraded fragments of SMC complexes miss essential parts of their ATPase and would not be able to load onto DNA. Their presence therefore should not affect the results from the force measurements or their interpretation.

In the revised manuscript, to better characterise the forces of the opening interface within the cohesin ring, we have both performed additional experiments as well as developed additional simulations that supported our original conclusions and ruled out other possible interpretations.

We measured rupture forces for cohesin complexes in which the kleisin/head interface was covalently crosslinked (Smc3^{Psm3}-kleisin cohesin in our new manuscript). As we explained above, it was not possible to use the suggested 6C model for this experiment. Therefore, we designed a new system in which the kleisin and Smc3-head subunits carried two SpyTags, that could be covalently closed by the addition of a distinct molecule consisting of two SpyCatcher proteins separated by a flexible and unstructured 50-nm long linker to crosslink the Smc3^{Psm3}-kleisin interface. As we also already described in our response to one of the points above, closing the Smc3^{Psm3}-kleisin interface in this manner after DNA loading abolished spontaneous release of cohesin from the DNA through the Smc3^{Psm3}-kleisin gate (New Fig. 1g).

The distribution of rupture forces for complexes with a covalently crosslinked Smc3^{Psm3}-kleisin interface was indistinguishable from the rupture forces of the non-crosslinked complexes. Together with our new simulations, we now show that the only scenario that can explain our data is that the hinge interface is weak (ruptures at ~20 pN force) and the Smc3^{Psm3}-kleisin interface is strong (ruptures at ~70 pN force) and that the topologically entrapped DNA escapes from the cohesin ring after the weakest interface breaks due to the applied force. We have made this clearer in the revised manuscript.

The insertion positions for our SpyTag and SpyCatcher corresponds closely to the ones chosen by Collier & Nasmyth. We cannot explain why the crosslinking efficiency in case of the fission yeast protein is somewhat lower, when compared to the budding yeast protein. However, we have explored an alternative location where we inserted crosslinks, which resulted in crosslinking efficiency of only around 50%. We are therefore satisfied that we have used the reasonably best possible reagents available to us to perform our experiments.

Decision Letter, first revision:

Message: Our ref: NSMB-A47027A-Z

3rd Aug 2023

Dear Dr. Molodtsov,

Thank you for submitting your revised manuscript "Mechanical disengagement of the cohesin ring" (NSMB-A47027A-Z). It has now been seen by the original referees and their comments are below. The reviewers find that the paper has improved in revision, and therefore we'll be happy to accept it in principle in Nature Structural & Molecular Biology, pending minor revisions to satisfy the referees' final requests and to comply with our editorial and formatting guidelines.

We are now performing detailed checks on your paper and will send you a checklist detailing our editorial and formatting requirements in about 7-10 days. Please do not upload the final materials and make any revisions until you receive this additional information from us.

To facilitate our work at this stage, it is important that we have a copy of the main text as a word file. If you could please send along a word version of this file as soon as possible, we would greatly appreciate it; please make sure to copy the NSMB account (cc'ed above).

Sincerely,

Dimitris Typas
Associate Editor
Nature Structural & Molecular Biology
ORCID: 0000-0002-8737-1319

Reviewer #1 (Remarks to the Author):

The revised manuscript by Richeldi and coworkers has improved greatly. The authors addressed most of the questions that I raised with adding new data and by extending their statistical and numerical analyses. They now clearly showed that the rupture force is from the detachment of single cohesins that are topologically loaded on DNA and the hinge is the mechanically weakest interface which is prone to disengage upon external force. I recommend this study for the publication in NSMB.

Reviewer #2 (Remarks to the Author):

Richeldi and co-workers have provided a substantially extended and edited version of work previously rejected by NSMB (A47027A-Z).

Previously, our major concern was that it had not been demonstrated by the authors, with sufficient certainty, that the cohesin complexes being pulled on are topologically loaded onto DNA, encircling it. While it is still not demonstrated with absolute certainty, the addition of a DNA cleavage experiment and the demonstration of cohesin sliding on the DNA provide good evidence that cohesin is not just bound to the DNA. Further, good evidence is provided by the inclusion of a new construct that can be closed at the Smc3-kleinin interface using two SpyTags and an innovative dimeric SpyCatcher construct and showing that it leads to the expected changes in behaviour when unloading DNA. I found this addition exciting and very helpful. It might even inspire others in the field to think about alternative gate closure methods based on the SpyCatcher system. And the new Smc3-kleisin crosslinked construct does not change behaviour when doing the pulling experiments, which supports the previous conclusion that the hinge is the weaker gate. The salt dependency of the loaded sample has been clarified well, too.

There are a number of improvements to the analysis and modelling of the pulling data obtained, which seem convincing to me and improve the manuscript. Reviewer #1 will need to comment on these more, since, as stated before, we are not experts in this area.

Concerns about protein quality, especially the hinge-crosslinked version have not been alleviated fully but we can live with it is because of the above added experiments that produce more confidence in the constructs being functional. The authors did try another crosslink but that was worse.

All our minor points have been addressed or at least acknowledged. For example: point 3 about whether it is rupture or bleaching and point 5, that more interfaces need to be closed (now two). The third interface (Smc1-kleisin) is thought by everyone in the field to be constitutive and strong (see C) below).

So, in summary, we feel much more positive about this work. The results make a lot of sense in the cellular context as is pointed out well in the Discussion. Knowing which interface breaks first is perhaps not transformative but being able to be sure about the forces cohesin can withstand and being sure that this behaviour is caused by it encircling DNA is important and will move the field and cell biology forward.

Some minor points in no particular order:

A) Is the SpyCatcher linker really 50 nm? That is very long. Does it have to be that long and why?

B) Line 101: why only 52 percent unloaded ?

C) The authors could have closed all interfaces for completeness. It would have removed the need to speculate what the second-weakest interface is. I am not suggesting to do this now, though.

D) Some very nice controls have been added. Another one could have been to cleave the SMC arms as reported previously. Again, not needed now.

E) Reference 1: earliest proposal of the ring model is in Haering 2002 (which also reports the first "open" hinge, as indicated currently only by References 26 & 27)

F) Line 210 ... because normally the DNA goes in through the hinge gate. One could have used the same SpyTag x2 / dimeric SpyCatcher approach at the hinge to close the gate AFTER loading instead.

G) Line 312 "singe"

H) Lines 332-334: As it is this sentence doesn't make much sense? How about "The build-up of tension generated by spindle forces could result in THE cohesin molecules experiencing the most strain, undergoING mechanical disengagement at the hinge."

I) Line 335: remove "through".

J) Lines 338-340: Is this assumption based on data? Are we sure that the hinge and the acetylated Smc3 head are far enough from each other, therefore preventing the acetylation to have an effect on the mechanical stability of the complex?

K) Line 471: what dilution of the primary antibody?

L) We like the last sentence of the Discussion!

Author Rebuttal, first revision:

Remark to all reviewers

We would like to thank the reviewers for their critical reading of our new manuscript. We greatly appreciate their renewed interest in our work and delighted by their positive and enthusiastic assessment of the revised manuscript.

Reviewer #1 (Remarks to the Author):

The revised manuscript by Richeldi and coworkers has improved greatly. The authors addressed most of the questions that I raised with adding new data and by extending their statistical and numerical analyses. They now clearly showed that the rupture force is from the detachment of single cohesins that are topologically loaded on DNA and the hinge is the mechanically weakest interface which is prone to disengage upon external force. I recommend this study for the publication in NSMB.

We want to express our thanks to the reviewer for acknowledging the improvement in the manuscript, with regards to both new experimental data and expanded statistical analyses. We are delighted that the reviewer is satisfied with our response and recommends this study for publication in NSMB.

Reviewer #2 (Remarks to the Author):

Richeldi and co-workers have provided a substantially extended and edited version of work previously rejected by NSMB (A47027A-Z).

Previously, our major concern was that it had not been demonstrated by the authors, with sufficient certainty, that the cohesin complexes being pulled on are topologically loaded onto DNA, encircling it. While it is still not demonstrated with absolute certainty, the addition of a DNA cleavage experiment and the demonstration of cohesin sliding on the DNA provide good evidence that cohesin is not just bound to the DNA. Further, good evidence is provided by the inclusion of a new construct that can be closed at the Smc3-kleinin interface using two SpyTags and an innovative dimeric SpyCatcher construct and showing that it leads to the expected changes in behaviour when unloading DNA. I found this addition exciting and very helpful. It might even inspire others in the field to think about alternative gate closure methods based on the SpyCatcher system. And the new Smc3-kleisin crosslinked construct does not change behaviour when doing the pulling experiments, which supports the previous conclusion that the hinge is the weaker gate. The salt dependency of the loaded sample has been clarified well, too.

There are a number of improvements to the analysis and modelling of the pulling data obtained, which seem convincing to me and improve the manuscript. Reviewer #1 will need to comment on these more, since, as stated before, we are not experts in this area.

Concerns about protein quality, especially the hinge-crosslinked version have not been alleviated fully but we can live with it is because of the above added experiments that produce more confidence in the constructs being functional. The authors did try another crosslink but that was worse.

All our minor points have been addressed or at least acknowledged. For example: point 3 about whether it is rupture or bleaching and point 5, that more interfaces need to be closed (now two). The third interface (Smc1-kleisin) is thought by everyone in the field to be constitutive and strong (see C) below).

So, in summary, we feel much more positive about this work. The results make a lot of sense in the cellular context as is pointed out well in the Discussion. Knowing which interface breaks first is perhaps not transformative but being able to be sure about the forces cohesin can withstand and being sure that this behaviour is caused by it encircling DNA is important and will move the field and cell biology forward.

We thank the reviewer for acknowledging the improvement in the manuscript, with regards to both new experimental data and expanded statistical analyses. We are delighted that the reviewer is satisfied with our response.

Some minor points in no particular order:

A) Is the SpyCatcher linker really 50 nm? That is very long. Does it have to be that long and why?

We thank the reviewer for this comment, and we agree that this was not sufficiently clearly explained. The linker that we used in the experiments consisted of 141 amino acid residues that separated the two SpyCatcher moieties. This linker is predicted to be a disordered polypeptide and assuming the average contour length of 1 amino acid residue to be ~ 0.4 nm, the expected length of the linker was calculated to be slightly over 50 nm. While the absolute length of the linker does not affect interpretation of the experiments and our conclusions, we reasoned that since both SpyCatchers are expressed as a single polypeptide in the same direction, a long linker length could facilitate the flexibility between the SpyCatchers and therefore improve chances of both proteins binding their respective SpyTags regardless of their orientation, which should improve efficiency of the crosslinking. Indeed, as indicated in the main text, with this construct we achieved remarkable efficiency of over 90%. We made this clearer in the supplementary methods included with the revised version of the manuscript.

B) Line 101: why only 52 percent unloaded ?

The reviewer correctly noted that only 52% of cohesin was released after a 60-minute incubation with ATP. Our interpretation is that spontaneous release of cohesin from DNA in the presence of ATP is not an efficient process. In previous bulk experiments (Murayama & Uhlmann, 2015), after an analogous protocol, 60-70% of the DNA entrapped by cohesin was released (see Figure 3a of the above mentioned article), which is similar, but slightly higher than in our experiments. In the bulk experiment the reaction was incubated at 32 °C, whereas the single-molecule reaction performed in this study was carried out under the microscope at room temperature. This is consistent with the idea that this reaction is inherently inefficient and accelerated by increasing temperature.

C) The authors could have closed all Interfaces for completeness. It would have removed the need to speculate what the second-weakest interface is. I am not suggesting to do this now, though.

We thank the reviewer for this comment. We agree that closing all interfaces would have allowed to conclusively determine the second weakest interface. However, this went beyond the scope of our study, which focused exclusively on determining the weakest interface. Our data unambiguously showed that the hinge domain is the first to break when cohesin is subjected to external force. We have made it clearer in the revised text. We agree that all interfaces should have been crosslinked to determine the second-weakest interface. Ongoing work in our group will hopefully shed light on this also. We clarified this in the revised text.

D) Some very nice controls have been added. Another one could have been to cleave the SMC arms as reported previously. Again, not needed now.

We thank the reviewer for this comment and we are delighted the reviewer finds our controls nice and sufficient. We agree, cleaving the SMC arms has been successfully used before (Stigler et al., 2016), but it requires insertion of an additional cleavable tag. In this study, we tried to reduce non-essential tags as the variants used here already bore several tags for purification, visualisation and force application. We would expect that cleavage of the cohesin would result in its release from DNA, just as cleavage of the nucleic acid has proven in the control performed.

E) Reference 1: earliest proposal of the ring model is in Haering 2002 (which also reports the first “open” hinge, as indicated currently only by References 26 & 27)

We thank the reviewer for spotting this. We have included Haering 2002 citation in our revised text.

F) Line 210 ... because normally the DNA goes in through the hinge gate. One could have used the same SpyTag x2 / dimeric SpyCatcher approach at the hinge to close the gate AFTER loading instead.

We thank the reviewer for the opportunity to expand on this comment. We agree that another way this could have been done is by crosslinking the hinge after the loading. Our preliminary experiments had shown that direct crosslinking of the Smc3^{Psm3}-kleisin interface hindered loading of the cohesin complex, and therefore we could only close this interface after the loading. However, we could close the hinge interface before the loading and the loading efficiency of the cohesin with the closed hinge domain was still sufficient to perform our experiments. Since these experiments were technically significantly easier than closing the complex afterwards, we approached the hinge crosslinking using this more straightforward protocol.

G) Line 312 “singe”

We thank the reviewer for spotting this typo, which has now been corrected.

H) Lines 332-334: As it is this sentence doesn't make much sense? How about “The build-up of tension generated by spindle forces could result in THE cohesin molecules experiencing the most strain, undergoING mechanical disengagement at the hinge.”

We thank the reviewer for pointing this out and for suggesting an alternative phrasing for the sentence. We have corrected the sentence in the manuscript to make it clearer.

I) Line 335: remove “through”.

We thank the reviewer for spotting this typo, which has now been corrected.

J) Lines 338-340: Is this assumption based on data? Are we sure that the hinge and the acetylated Smc3 head are far enough from each other, therefore preventing the acetylation to have an effect on the mechanical stability of the complex?

We thank the reviewer for giving us the opportunity to expand on this point. Although we do not have any data to conclusively show that head acetylation does not alter the mechanical stability of the cohesin ring, we also do not have any data pointing to the contrary. Our preliminary data showed that force application in the presence of ATP did not alter the rupture force observed to disengage

cohesin from the DNA. From this, we can speculate that movements involving the heads (such as head engagement/disengagement dependent upon ATP binding/hydrolysis) do not affect cohesin's mechanical stability and therefore acetylation is unlikely to change the mechanical stability of the cohesin ring. However, we agree that this statement is a speculation and we clearly indicated it as such in the revised manuscript.

K) Line 471: what dilution of the primary antibody?

The dilution of the mouse anti-PK primary antibody used was 1:10,000. We have now made this clearer in the Methods section.

L) We like the last sentence of the Discussion!

We thank the reviewer for the constructive feedback and the positive acknowledgement of the improvement of the overall manuscript!

Final Decision Letter:**Message** 11th Sep 2023

:

Dear Dr. Molodtsov,

We are now happy to accept your revised paper "Mechanical disengagement of the cohesin ring" for publication as an Article in Nature Structural & Molecular Biology.

Your paper will be published online soon after we receive proof corrections and will appear in print in the next available issue. You can find out your date of online publication by contacting the production team shortly after sending your proof corrections. Content is published online weekly on Mondays and Thursdays, and the embargo is set at 16:00 London time (GMT)/11:00 am US Eastern time (EST) on the day of publication. Now is the

time to inform your Public Relations or Press Office about your paper, as they might be interested in promoting its publication. This will allow them time to prepare an accurate and satisfactory press release. Include your manuscript tracking number (NSMB-A47027B) and our journal name, which they will need when they contact our press office.

About one week before your paper is published online, we shall be distributing a press release to news organizations worldwide, which may very well include details of your work. We are happy for your institution or funding agency to prepare its own press release, but it must mention the embargo date and Nature Structural & Molecular Biology. If you or your Press Office have any enquiries in the meantime, please contact press@nature.com.

Please note that *Nature Structural & Molecular Biology* is a Transformative Journal (TJ). Authors may publish their research with us through the traditional subscription access route or make their paper immediately open access through payment of an article-processing charge (APC). Authors will not be required to make a final decision about access to their article until it has been accepted. [Find out more about Transformative Journals](https://www.springernature.com/gp/open-research/transformative-journals)

Authors may need to take specific actions to achieve [compliance](https://www.springernature.com/gp/open-research/funding/policy-compliance-faqs) with funder and institutional open access mandates. If your research is supported by a funder that requires immediate open access (e.g. according to [Plan S principles](https://www.springernature.com/gp/open-research/plan-s-compliance)) then you should select the gold OA route, and we will direct you to the compliant route where possible. For authors selecting the subscription publication route, the journal's standard licensing terms will need to be accepted, including [self-archiving policies](https://www.springernature.com/gp/open-research/policies/journal-policies). Those licensing terms will supersede any other terms

that the author or any third party may assert apply to any version of the manuscript.

Sincerely,

Dimitris Typas
Associate Editor
Nature Structural & Molecular Biology
ORCID: 0000-0002-8737-1319
